# Pum2 and TDP-43 refine area-specific cytoarchitecture post-mitotically and modulate translation of *Sox5, Bcl11b*, and *Rorb* mRNAs in developing mouse neocortex

Kawssar Harb[1]*, Melanie Richter[2†], Nagammal Neelagandan[1†‡], Elia Magrinelli[3], Hend Harfoush[1§], Katrin Kuechler[1#], Melad Henis[2,4], Irm Hermanns-Borgmeyer[5], Froylan Calderon de Anda[2]*, Kent Duncan[1]*¶

[1]Neuronal Translational Control Group, Center for Molecular Neurobiology (ZMNH), University Medical Center Hamburg-Eppendorf (UKE), Hamburg, Germany; [2]Institute of Developmental Neurophysiology, Center for Molecular Neurobiology, University Medical Center Hamburg-Eppendorf, Hamburg, Germany; [3]Department of Basic Neuroscience, University of Geneva, Geneva, Switzerland; [4]Department of Anatomy and Histology, Faculty of Veterinary Medicine, New Valley University, New Valley, Egypt; [5]Transgenic Service Group, Center for Molecular Neurobiology (ZMNH), University Medical Center Hamburg-Eppendorf (UKE), Hamburg, Germany

*For correspondence:
kawssar.harb@gmail.com (KH);
froylan.calderon@zmnh.uni-hamburg.de (FCdA);
kent.duncan5@gmail.com (KD)

†These authors contributed equally to this work

Present address: ‡The Institute of Bioengineering (IBI), Ecole Polytechnique Federale de Lausanne (EPFL), Lausanne, Switzerland; §Faculty of Science, Alexandria University, Alexandria, Egypt; #University Children's Research @ Kinder-UKE (UCR), University Medical Center Hamburg-Eppendorf (UKE), Hamburg, Germany; ¶Evotec SE, Manfred Eigen Campus, Essener Bogen 7, Hamburg, Germany

Competing interest: The authors declare that no competing interests exist.

**Abstract** In the neocortex, functionally distinct areas process specific types of information. Area identity is established by morphogens and transcriptional master regulators, but downstream mechanisms driving area-specific neuronal specification remain unclear. Here, we reveal a role for RNA-binding proteins in defining area-specific cytoarchitecture. Mice lacking Pum2 or overexpressing human TDP-43 show apparent 'motorization' of layers IV and V of primary somatosensory cortex (S1), characterized by dramatic expansion of cells co-expressing Sox5 and Bcl11b/Ctip2, a hallmark of subcerebral projection neurons, at the expense of cells expressing the layer IV neuronal marker Rorβ. Moreover, retrograde labeling experiments with cholera toxin B in *Pum2; Emx1-Cre* and TDP43[A315T] mice revealed a corresponding increase in subcerebral connectivity of these neurons in S1. Intriguingly, other key features of somatosensory area identity are largely preserved, suggesting that Pum2 and TDP-43 may function in a downstream program, rather than controlling area identity per se. Transfection of primary neurons and in utero electroporation (IUE) suggest cell-autonomous and post-mitotic modulation of Sox5, Bcl11b/Ctip2, and Rorβ levels. Mechanistically, we find that Pum2 and TDP-43 directly interact with and affect the translation of mRNAs encoding Sox5, Bcl11b/Ctip2, and Rorβ. In contrast, effects on the levels of these mRNAs were not detectable in qRT-PCR or single-molecule fluorescent in situ hybridization assays, and we also did not detect effects on their splicing or polyadenylation patterns. Our results support the notion that post-transcriptional regulatory programs involving translational regulation and mediated by Pum2 and TDP-43 contribute to elaboration of area-specific neuronal identity and connectivity in the neocortex.

## Editor's evaluation

All of the reviewers and editors agree that your deep, rigorous, and multidimensional study addresses novel and significant questions, that it will be an important addition to the literature, and

that it presents experiments and data that will motivate further study in the field. You have convincingly demonstrated translational modulation of Sox5, Bcl11b, and *Rorb* by the RNA-binding proteins (RBPs) Pum2 and TDP-43, with thoughtful experiments supporting cell-autonomous effects of the RBPs on the translational regulation of Bcl11b, Sox5, and *Rorb*. Your in vivo gain and loss-of-function data in a range of genetically manipulated mouse lines, coupled with in vitro neuronal culture experiments, provide strong evidence for the control that Pum2 and TDP43 exert on these key regulators of neuronal diversification during corticogenesis. Thank you for completing such a deeply informative body of work and for discussing the complexities involved so thoughtfully within your paper.

## Introduction

The neocortex is the largest and the most complex structure in the mammalian brain and plays a crucial role in processing sensory information, controlling movement and higher-level cognition. Two prominent architectural features of the neocortex are its 'tangential areal' and 'radial laminar' organization. Neocortical areas, defined by Brodmann as the 'organs of the brain,' form the basis for sensory perception and mediate our behavior (*Rakic, 1988*; *Zilles and Amunts, 2010*). The basic plan of the neocortex comprises four primary areas, spatially organized into six horizontal layers, each containing a heterogeneous population of neurons, distinguished by their morphology, connectivity, molecular code, and function (*O'Leary and Nakagawa, 2002*; *Rash and Grove, 2006*; *Sur and Rubenstein, 2005*). Within each area, the relative number of neuronal subtypes appears to be tuned to correspond with area function. For instance, the primary motor area (M1) has a thick layer V with numerous subcerebral output neurons, but a very thin layer IV for receiving thalamic input. In contrast, the primary somatosensory area (S1) has exactly the opposite organization and is therefore adapted to receive input (*Dehay and Kennedy, 2007*; *Glickfeld et al., 2013*; *Yamawaki et al., 2014*).

Neocortical area patterning is controlled by a regulatory hierarchy: Morphogens establish differentially graded expression of transcription factors, which then determine the area identity of the neurons forming the cortical plate (*Alfano and Studer, 2013*; *O'Leary et al., 2007*; *O'Leary and Sahara, 2008*). This areal commitment of newly born projection neurons is followed by laminar fate determination. Opposing molecular programs direct their differentiation into one of the major neuronal subtype identities (*Greig et al., 2013*; *Jabaudon, 2017*; *Molyneaux et al., 2007*). Most work to date has addressed these two major regulatory schemes separately. Thus, how they interact and how neurons integrate area and subtype identities remains mysterious. Area-specific differences in layer-neuron identity imply exquisite molecular control over cell fate within specific areas. Nevertheless, downstream molecular mechanisms that define area-specific patterns of neuronal identity and connectivity remain poorly understood.

Historically, most analyses of how regulation of gene expression contributes to corticogenesis focused on transcriptional control by nuclear transcription factors, which clearly is a major driving force in control of neuronal fate (*Greig et al., 2013*; *O'Leary and Sahara, 2008*). In contrast, the role of post-transcriptional regulation in cortical development is still emerging. RNA-binding proteins (RBPs) are major mediators of post-transcriptional control and can influence different steps of mRNA metabolism, including splicing, stability, translation, and localization (*Pilaz and Silver, 2015*). Recent studies have revealed roles for RBPs in many aspects of cortical development that affect cortical cytoarchitecture and suggest potential connections between these effects and both neurodevelopmental and neurodegenerative diseases (*Jung and Lee, 2021*; *Kanemitsu et al., 2017*; *Kiebler et al., 2013*; *Kraushar et al., 2014*; *La Fata et al., 2014*; *Lee et al., 2019*; *Pilaz and Silver, 2015*; *Sena et al., 2021*; *Vessey et al., 2012*; *Zahr et al., 2018*). However, whether RBPs regulate cytoarchitecture area-specifically remains unknown.

Here, we examine the potential contribution of post-transcriptional regulation to area-specific neuronal identity and connectivity by focusing on two RBPs: Pumilio-2 (Pum2) and Tar-DNA binding protein 43 (TDP-43). We chose to focus on these specific proteins based on their known and distinct roles in post-transcriptional regulation in the nervous system and because of TDP-43's importance in neurodegenerative diseases that affect cortical neurons (*Buratti and Baralle, 2014*; *Goldstrohm et al., 2018*; *Lagier-Tourenne et al., 2010*; *Martínez et al., 2019*; *Vessey et al., 2012*; *Zahr et al., 2018*; *Zhang et al., 2017*). Pum2, a quintessential RBP enriched in the nervous system, is found exclusively in the cytoplasm and dendrites, where it controls post-transcriptional steps of gene expression

that take place in these subcellular compartments (*Goldstrohm et al., 2018*; *Vessey et al., 2012*; *Vessey et al., 2010*). As such, we consider it a particularly interesting neuronal RBP to investigate in control of area-specific cytoarchitecture. Previous work has implicated the combined action of Pum2 and its ortholog Pum1 in many aspects of brain development (*Zhang et al., 2017*), and Pum2 has recently been implicated in control of cortical axonogenesis (*Martínez et al., 2019*). However, area-specific phenotypes resulting from selective knockout of Pum2 in developing neocortical neurons have not previously been described.

Unlike Pum2, TDP-43 is a 'shuttling' RBP that moves back and forth between the nucleus and cytoplasm to regulate gene expression primarily at the post-transcriptional level in both compartments (reviewed in *Lee et al., 2011*). Strong overexpression of TDP-43 in developing neuronal progenitors leads to apoptosis with concomitant pleiotropic effects on cortical development (*Vogt et al., 2018*). However, the impact of lower-level, post-mitotic overexpression of TDP-43 on area-specific cytoarchitecture has not previously been examined. TDP-43 is heavily implicated as a key causal factor in the neurodegenerative diseases (amyotrophic lateral sclerosis [ALS]) and frontotemporal dementia (FTD), both of which show some degree of area-selective pathology in the neocortex of both patients and animal models (*Taylor et al., 2016*). While diseases are typically classified into either neurodevelopmental or neurodegenerative, there is long-standing interest in the idea that altered neuronal specification and wiring during development might ultimately contribute to degenerative disease later in life (*Greig et al., 2013*). Thus, one goal of our study was to see whether we could find evidence for area-specific effects on layer neuron identity in an established mouse model of ALS driven by a patient mutation in TDP-43.

By combining genetics with molecular imaging and in vivo biochemical approaches, we uncovered evidence for a role for RBPs in shaping the specialized layering pattern of S1. Our work highlights an apparent contribution of post-transcriptional repression of *Sox5* and *Bcl11b (Ctip2)* mRNAs and activation of *Rorß* mRNA as a downstream mechanism in area-specific control of neuronal identity and connectivity. Moreover, our data provide evidence that Pum2 and TDP-43 regulate neuronal identity post-mitotically in S1, and may do so at least partly through competing effects on translation of key regulators of neuronal identity.

## Results

### Contribution of RBPs Pum2 and TDP-43 to area-specific neuronal cytoarchitecture in the neocortex

We used a reverse-genetic approach to investigate whether RBPs might contribute to the establishment of neuronal identity in an area-specific manner. Specifically, we compared the expression of proteins that determine layer-specific neuronal subtypes in different cortical areas of mutant mice for the two RBPs, Pum2 and TDP-43. To this end, we generated *Pum2* mice with loxP sites flanking exons 6 and 7. Crossing these mice to a line expressing Cre recombinase under the control of the *Emx1* promoter (*Iwasato et al., 2000*) enabled selective inactivation of *Pum2* expression in the forebrain (*Gorski et al., 2002*; *Figure 1—figure supplement 1*). To examine a potential contribution of TDP-43 and a possible link to human disease, we used a previously described transgenic mouse line containing a mutant allele that causes the neurodegenerative disease, ALS, in human patients *Prnp-TARDBP $^{A315T}$* (TDP43$^{A315T}$) (*Wegorzewska et al., 2009*).

We analyzed the overall brain architecture in *Emx1$^{Cre}$; Pum2$^{fl/fl}$* (*Pum2* cKO) and TDP43$^{A315T}$ (TDP43$^{A315T}$) mice compared to their littermate controls (*Figure 1—figure supplements 2 and 3*). At postnatal day 0 (P0), brain size and cortical thickness were similar to littermate controls in both mutants (*Figure 1—figure supplement 2a and b*). Our Nissl staining showed no strong cortical morphological differences in coronal (P0) and sagittal (P7) sections of both mutants compared to their littermate controls (*Figure 1—figure supplement 3*). On a cellular level, nuclear size in S1 layer II–VI neurons was slightly larger in *Pum2* cKO compared to controls, while it was not significantly affected in TDP43$^{A315T}$ mutants (*Figure 1—figure supplement 2c*). Moreover, we did not observe any significant changes in the total number of DAPI cells in both mutants compared to their littermate controls (*Figures 1 and 2*, *Figure 1—figure supplement 5*). To check whether the neurogenesis to gliogenesis ratio might be affected in our mutants at P0, we performed staining for NeuN as a neuronal marker and GFAP as a glial marker (*Figure 1—figure supplement 2d*). Our staining showed that most

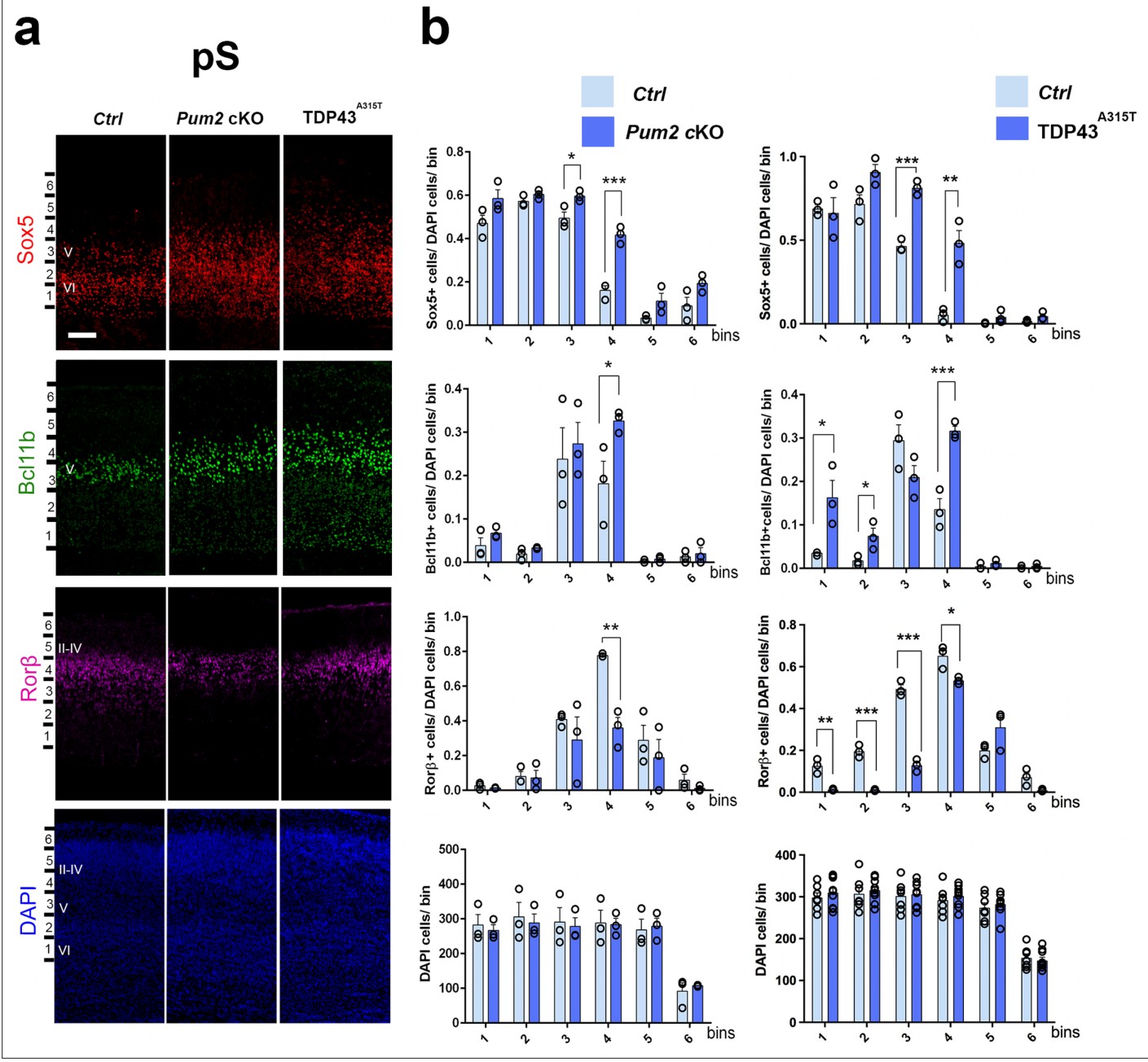

**Figure 1.** Neocortical neuronal identity of somatosensory cortex is altered in *Pum2* cKO and TDP43[A315T] mice. (**a**) Coronal sections from neonatal (P0) brains of controls (*Ctrl*), *Pum2* cKO, or *Prnp-TARDBP*[A315T] (TDP43[A315T]) mice were stained with antibodies recognizing Sox5, Bcl11b, or Rorβ or with DAPI to mark nuclei in the prospective somatosensory cortex (pS). (**b**) Quantification of results from n = 3 mice of each genotype is shown to the right of the relevant marker. Distribution of cells across six equal-sized bins is shown. For Bcl11b, only high-expressing neurons were counted. Data are shown as means ± standard error of the mean (SEM), n = 3 for each genotype. *p≤0.05, **p≤0.01, ***p≤0.001, two-tailed *t*-test. *Pum2* cKO: *Pum2*[fl/fl]; *Emx1*[Cre]; II–IV, V, VI: layers II–IV, V, and VI. Scale bar: 100 μm.

The online version of this article includes the following figure supplement(s) for figure 1:

**Figure supplement 1.** Generation of *Pum2* cKO mice.

**Figure supplement 2.** General cortical developmental features are unaltered in Pum2 and TDP-43 mutants.

**Figure supplement 3.** Cortical morphology of Pum2 and TDP-43 mutants is not affected.

**Figure supplement 4.** Specialized neocortical architecture of S1 and M1 is altered in *Pum2* cKO and TDP43[A315T] mutant mice.

*Figure 1 continued on next page*

*Figure 1 continued*

**Figure supplement 5.** Quantitative analysis of neocortical layer neuron identity in prospective somatosensory cortex (pS) vs. frontal/motor area (F/M) at P0.

**Figure supplement 6.** Normal layer VI and upper layers in *Pum2* cKO and TDP43[A315T] mice.

**Figure supplement 7.** Prospective somatosensory cortex (pS) layer IV/V phenotypes are also observed in *Pum2* KO, but not *Pum2[fl/+]*; *Emx1[Cre]* transgenic mouse line.

**Figure supplement 8.** Expression patterns and relative levels of TDP-43 and transgenic hTDP-43 proteins in developing mouse neocortex.

**Figure supplement 9.** Neuronal identity of layers IV and V is affected by TDP-43 gain of function.

cortical neurons co-expressed NeuN and DAPI, but GFAP was essentially absent from cortices of both mutants and controls. This is consistent with gliogenesis starting at E18.5-P0 in WT animals (*Miller and Gauthier, 2007*; *Sarnat, 1992*) and shows that this is not affected in either mutant. The same experiment showed significant hippocampal staining with both NeuN and GFAP, in accordance with earlier gliogenesis in the hippocampus (*Figure 1—figure supplement 2d*). Overall, these findings support use of DAPI as a normalization factor in our following analysis.

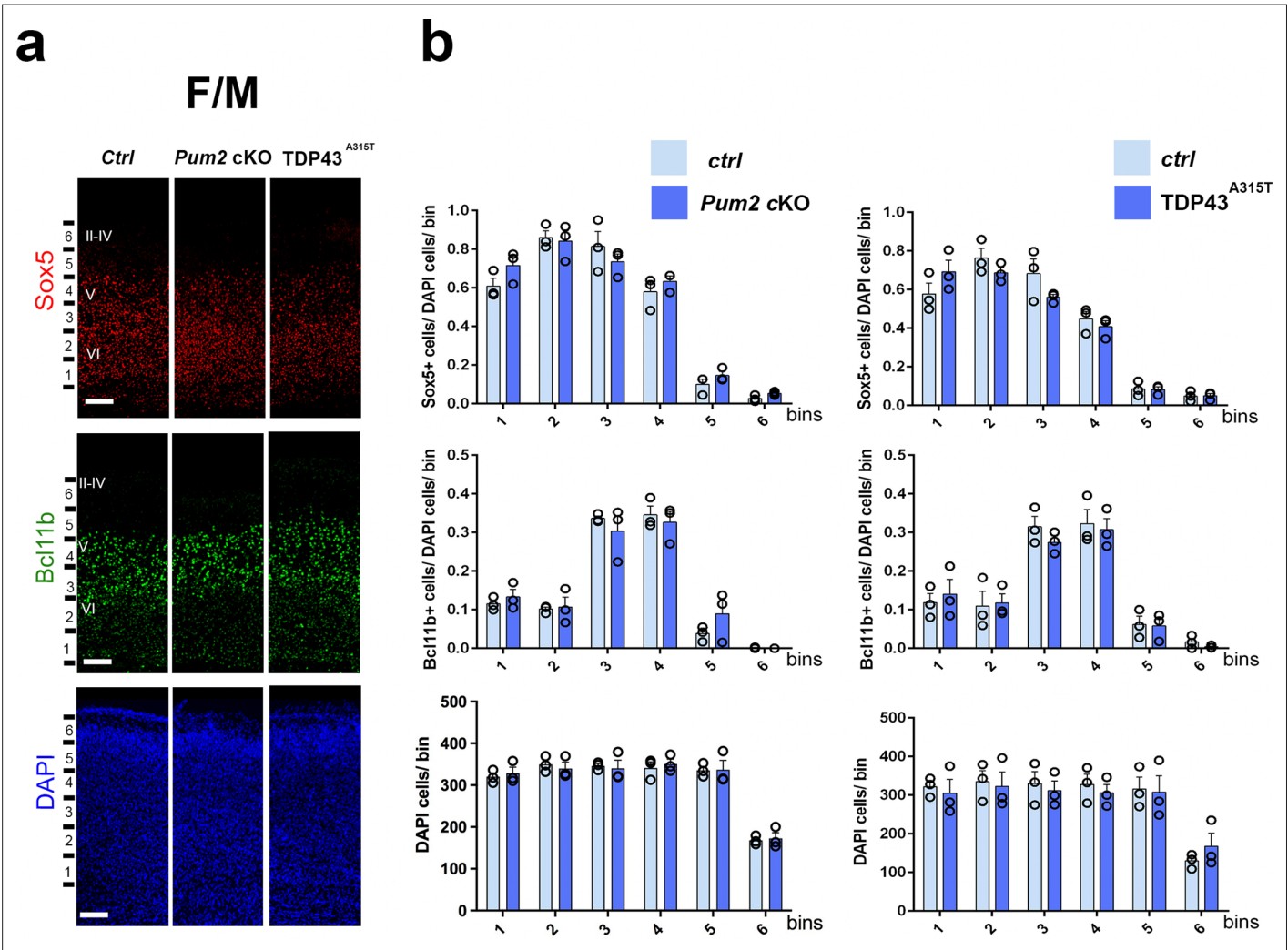

**Figure 2.** Neocortical neuronal identity remains unaffected in the motor cortex of Pum2 and TDP-43 mutants. (**a**) Coronal sections from neonatal (P0) brains of controls (*Ctrl*), *Pum2* cKO, or *Prnp-TARDBP[A315T]* (TDP43[A315T]) mice were stained with antibodies recognizing Sox5, Bcl11b, or with DAPI to mark nuclei in the frontal/motor area (F/M). (**b**) Quantification of results from n = 3 mice of each genotype is shown to the right of the relevant marker. Distribution of cells across six equal-sized bins is shown. For Bcl11b, only high-expressing neurons were counted. Data are shown as means ± standard error of the mean (SEM), n = 3 for each genotype, two-tailed *t*-test. *Pum2* cKO: *Pum2[fl/fl]*; *Emx1[Cre]*; II–IV, V, VI: layers II–IV, V, and VI. Scale bar: 100 µm.

We focused our analysis on the frontal/motor (F/M) and somatosensory cortices, which show characteristic differences in neuronal subtype ratios related to their specialized functions. The frontal motor area F/M is characterized by a thick layer V (*Dehay and Kennedy, 2007*; *Polleux et al., 2001*), in which many cells co-express the molecular determinants of subcerebral projection neurons (SCPNs): Sox5 and Bcl11b (*Chen et al., 2008*; *Kwan et al., 2008*; *Lai et al., 2008*). In contrast, the primary somatosensory area (S1) has a thick layer IV, in which most cells express Rorβ, a bona fide marker of layer IV stellate cells (*Jabaudon et al., 2012*; *Nakagawa and O'Leary, 2003 Figure 1—figure supplement 4a and b*). In P0 WT coronal sections, we observed dramatic differences in the prospective somatosensory area (pS) compared to the frontal motor area (F/M) when we analyzed key molecular identity determinants that define specific neuronal subtypes (*Figure 1—figure supplement 4b*; *Arlotta et al., 2005*; *Bedogni et al., 2010*; *Chen et al., 2008*; *Jabaudon et al., 2012*; *Kwan et al., 2008*; *Lai et al., 2008*; *McKenna et al., 2011*). In both mutants, the number of Sox5+ and Bcl11b+ neurons in the upper region of layer V was significantly increased and radially expanded in pS, accompanied by a corresponding decrease in the number of Rorβ+ neurons in layer IV (*Figure 1*, *Figure 1—figure supplements 4a and 5a*). DAPI staining revealed no significant differences (*Figure 1*, *Figure 1—figure supplement 5a*), consistent with a potential switch in neuronal identity specification, rather than effects on cell number or migration. Similar effects were not observed in F/M cortex, where neither the number nor the radial distribution of Sox5+, Bcl11b+, or Tbr1+ neurons in the mutant lines differed from controls and Rorβ was not expressed, as expected (*Figure 2*, *Figure 1—figure supplements 5b and 6b*). Unlike the dramatic effects on layer IV/V in pS, we detected no significant changes in the layer VI neuronal marker Tbr1 or the upper layer marker Cux1 (*Nieto et al., 2004*), implying normal neuronal specification in these layers (*Figure 1—figure supplement 6a*).

Importantly, we observed similar phenotypes in pS with a constitutive *Pum2* KO line, but not in *Emx1^Cre^*; *Pum2^fl/+^* heterozygotes, confirming that the phenotype is due to loss of Pum2 function, rather than the Cre line used or Cre expression per se (*Figure 1—figure supplement 7a and b*). Taken together, our results suggest that Pum2 functions within the forebrain to influence layer IV/V specification in the pS.

We next wanted to resolve the nature of regulation of layer neuron fate markers in pS by TDP43^A315T^. In particular, we wanted to determine whether regulation was due to a specific property of the mutant protein or reflected a gain of function due to overexpression. To this aim, we examined a transgenic line reported to overexpress human *Prnp-TARDBP* (TDP43) at relatively low levels in the brain, which does not develop symptoms (*Arnold et al., 2013*). We first compared expression of the respective transgenic proteins in the two lines. Both hTDP-43 and hTDP-43^A315T^ were broadly expressed in neocortical areas and layers, including layers IV and V of the pS, in a pattern qualitatively like endogenous TDP-43 (*Figure 1—figure supplement 8a*). Although the distribution of cells expressing transgenic hTDP-43 or hTDP-43^A315T^ was qualitatively similar across layers in both F/M and pS, the intensity of the expression of the protein variants was different in developing neocortex. Interestingly, layer V neurons expressed higher levels of hTDP-43^A315T^, which was confirmed using hTDP-43 and Flag antibodies (*Figure 1—figure supplement 8a and b*). In addition, using quantitative immunoblotting, we confirmed overexpression of TDP-43 in the cytoplasmic fraction of neocortical lysates of the TDP43 line and in both nuclear and cytoplasmic fractions of the TDP43^A315T^ line (*Figure 1—figure supplement 8c*). Consistent with higher intensity for mutant hTDP-43 in immunostaining, quantitative immunoblotting indicated that hTDP-43^A315T^ protein levels were significantly higher than the hTDP-43 protein levels in cytoplasmic-enriched fractions of neocortex (*Figure 1—figure supplement 8c*).

Analyzing effects on neuronal identity due to WT TDP43 overexpression in this line revealed clear effects like those seen with the TDP43^A315T^ line, with significant increases in the number of cells expressing Sox5 and Bcl11b protein and fewer cells expressing Ror*β* protein (*Figure 1—figure supplement 9a and b*). Although the magnitude of these phenotypic effects with TDP43 was not as strong as those observed with *hTDP-43*^A315T^, finding them with the TDP43 line demonstrates that they are not line- or mutation-specific. Moreover, since this line does not develop ALS-like symptoms, our observations further suggest that the underlying effect on altered neuronal fate marker expression in the pS is likely due to gain of WT TDP-43 function and that altered cortical architecture in S1 during development per se is not sufficient to result in ALS-like symptoms (see 'Discussion'). To simplify the experimental workflow and analysis for TDP-43, we focused in our following analyses on the

TDP43[A315T] line since it showed qualitatively identical, but quantitatively stronger, phenotypes relative to WT TDP43 in multiple assays.

In sum, our phenotypic analyses in the pS and F/M areas support a role for the RBPs Pum2 and TDP-43 in area-specific regulation of neuronal identity marker expression in layers IV and V of the developing somatosensory cortex. Moreover, they suggest that Pum2 promotes the normal pattern of S1 neuronal identity marker expression, whereas gain of TDP-43 function can act in an apparently opposite manner to repress it.

## Increased subcerebral connectivity for S1 neurons in *Pum2* cKO and TDP43[A315T] mice

Co-expression of Bcl11b and Sox5 is a hallmark of SCPNs (*Chen et al., 2008*; *Kwan et al., 2008*; *Lai et al., 2008*), and ectopic Bcl11b overexpression in upper-layer progenitors is sufficient to redirect their axons from corticocortical projections into projections to subcerebral targets (*Chen et al., 2008*). We therefore wondered whether the increase and radial expansion of neurons expressing molecular determinants of SCPNs (Bcl11b and Sox5) would be accompanied by increased SCPN connectivity (*Arlotta et al., 2005*; *Chen et al., 2008*; *Kwan et al., 2008*; *Lai et al., 2008*). To examine this directly in *Pum2* cKO and TDP43[A315T] mice, we injected fluorophore-labeled cholera toxin B (CTB) into the pons for retrograde labeling of SCPNs (*Conte et al., 2009*; *Figure 3a*). This revealed significantly increased labeling in layer V and a striking radial expansion in both *Pum2* cKO and TDP43[A315T] vs. their respective littermate controls (*Figure 3b and c*).

To understand whether the increase in Sox5 corresponds with the increase in Bcl11b, we co-immunostained Sox5 and Bcl11b in coronal sections of *Pum2* cKO and TDP43[A315T] and their control littermates. Our analysis showed an increase in the number of Sox5+/Bcl11b+ neurons in both mutants, suggesting that ectopic expression of Sox5 corresponds with that of Bcl11b (*Figure 4a*). We next combined retrograde labeling of SCPN with staining for either Sox5 or Bcl11b 2 to test whether the increased number of SCPN in S1 directly corresponds with the increased number of Sox5+/Bcl11b+ neurons. Our co-immunostaining showed that all retrogradely labeled neurons in controls and mutants co-expressed both Sox5 and Bcl11b (*Figure 4b and c*). Thus, the typical area-specific neuronal connectivity of S1 is dramatically altered in both *Pum2* cKO and TDP43[A315T], with more SCPNs in layer V and ectopic SCPNs in the position normally occupied by layer IV in S1. This pattern is reminiscent of motor cortex (*Armentano et al., 2007*; *Harb et al., 2016*; *Tomassy et al., 2010*), and thus, reflects apparent "motorization" of layer IV/V in S1.

## Most aspects of somatosensory area identity appear to be properly determined in *Pum2* cKO and TDP43[A315T] mice, despite layers IV and V being 'motorized'

Previously described mutants with a motorized layer IV/V in S1 affect multiple aspects of pS area identity (*Armentano et al., 2007*; *Harb et al., 2016*; *Tomassy et al., 2010*). Thus, we envisaged two hypotheses to explain apparent motorization of layer IV/V in *Pum2* cKO and TDP43[A315T] mutant mice. On the one hand, Pum2 and TDP-43 might control area identity, like previously described transcriptional regulators mentioned above. Alternatively, they could control layer IV/V specification and connectivity without affecting area identity per se. To test these hypotheses, we examined two hallmarks of area identity in S1 of *Pum2* cKO and TDP43[A315T] mice. We first checked the expression pattern of two standard molecular markers of neocortical area identity: Lmo4 (motor) (*Huang et al., 2009*) and Bhlhb5 (sensory) (*Joshi et al., 2008*). Both *Pum2* cKO and TDP43[A315T] mice showed overall a wild-type pattern of Lmo4 and Bhlhb5 expression (*Figure 5a*). Quantitative analysis of the total number of Lmo4 and Bhlhb5 cells normalized to DAPI showed major differences between motor and somatosensory cortex in all genotypes, as expected, suggesting that the pS maintains its areal identity and does not show an F/M identity. On a laminar level, comparison of Lmo4 and Bhlhb5 analysis between controls and mutants (source data related to *Figure 5a*) showed no significant changes in Lmo4 and Bhlhb5 in the pS of *Pum2* cKO compared to controls, but a significant increase in Lmo4 in bin1 and decrease in Bhlhb5 in bins 3 and 4 in TDP43[A315T]. In the case of the motor cortex, an increase in Bhlhb5 was observed in bin 6 of TDP43[A315T] while Lmo4 was unaltered in all bins. *Pum2* cKO did not show any change for Bhlhb5, but Lmo4 expression was decreased in bins 1 and 4. These differences are not surprising since both Lmo4 and Bhlhb5 regulate area-specific laminar identity (*Cederquist*

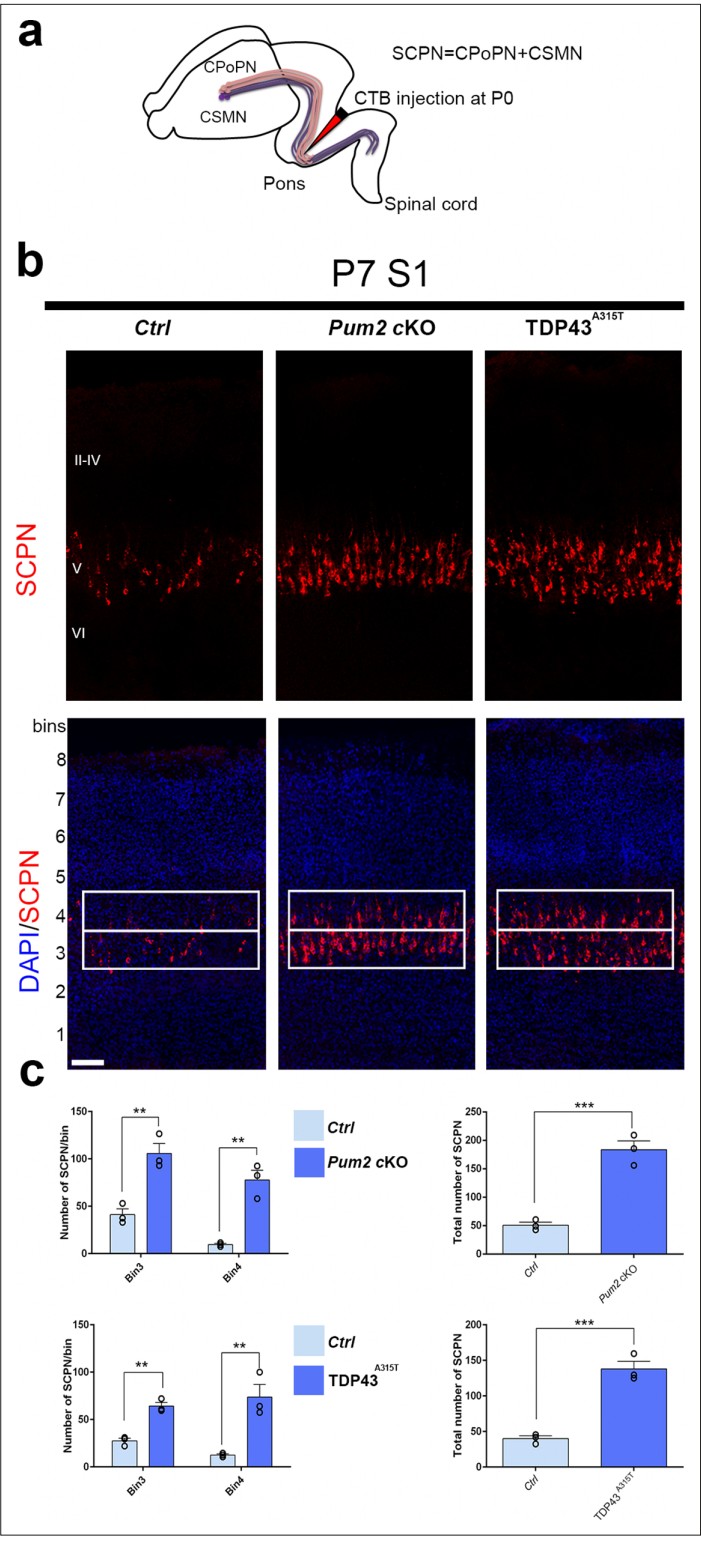

**Figure 3.** Increased subcerebral connectivity in somatosensory cortex of *Pum2* cKO and TDP43[A315T] mice. (**a**) Schematic representation of cholera toxin subunit B (CTB) injections at the midbrain/hindbrain junction (pons) for retrograde labeling of subcerebral projection neurons (SCPNs), including corticospinal PNs (CSMN) and corticopontine PNs (CPoPN). (**b**) Coronal sections from primary somatosensory cortex (S1) of controls, *Pum2* cKO, and TDP43[A315T] mice at P7 traced for SCPNs without (top) or with DAPI (bottom) staining. S1 columns merged with DAPI are divided into eight equal bins. White rectangles indicate bins 3 and 4. (**c**) Quantification of retrogradely

*Figure 3 continued on next page*

*Figure 3 continued*

labeled SCPNs in equal-sized bins for the three genotypes. Analysis of bins 3 and 4 is shown separately in the left panel and combined in the right panel. Data are shown as means ± standard error of the mean (SEM), n = 3 for each genotype. **p≤0.01, ***p≤0.001, two-tailed *t*-test. *Pum2* cKO: *Pum2*$^{fl/fl}$; *Emx1*$^{Cre}$; II–IV, V, VI: layers II–IV, V and VI, respectively. Scale bars: 100 µm.

---

*et al., 2013*; *Greig et al., 2013*; *Harb et al., 2016*; *Joshi et al., 2008*) and do not affect our conclusion regarding the unchanged areal identity of pS or F/M. Next, we examined another major hallmark of area identity in S1: the specialized 'barrels' in layer IV. These clusters of glutamatergic interneurons receive somatosensory input from the whiskers via the thalamus and can be visualized by serotonin (5HT) staining of the thalamic presynaptic terminals. In contrast to previously described mutants with motorized somatosensory areas (*Armentano et al., 2007*; *Tomassy et al., 2010*), barrels formed efficiently at the same tangential position and their numbers are not significantly different from controls in S1 of either *Pum2* cKO and TDP43$^{A315T}$ mice (*Figure 5b*), supporting a lack of major changes to thalamocortical axonal targeting and patterning in S1. Taken together, these experiments suggest that Pum2 and TDP-43 contribute to elaboration of area-specific cytoarchitecture of layers IV and V without strongly affecting area identity per se.

## Cell-autonomous and post-mitotic effects of TDP-43 gain of function and Pum2 loss of function on regulation of Sox5, Bcl11b, and Rorβ in pS neurons

We next asked whether ectopic expression of hTDP-43 or the patient mutant hTDP-43$^{A315T}$ would be sufficient cell-autonomously to drive a switch in expression of Sox5, Bcl11b, and Rorβ. For these experiments, we prepared primary neuronal cultures from pS-enriched neocortices at E18.5 and transfected them with plasmids containing either WT TDP43, TDP43$^{A315T}$, or EGFP as a control via electroporation before plating. After 2 days in culture, we fixed and stained the cells for both the transfected protein and for endogenous Sox5, Bcl11b, or Rorβ protein. We identified transfected cells by immunostaining for one of the epitope tags on hTDP-43 or EGFP for control transfections (*Figure 6a–c*). Subsequently, we quantified the number of transfected neurons that were also positive for Sox5, Bcl11b, or Rorβ protein in the three different transfections. This revealed that expression of either WT hTDP-43 or the hTDP-43$^{A315T}$ mutant in transfected cortical neurons could strongly induce Sox5 and Bcl11b proteins to a similar extent (*Figure 6a and b*). Expression of either protein also significantly reduced Rorβ protein expression (*Figure 6c*). These results are consistent with our observations with transgenic lines and suggest that increased levels of TDP-43, rather than a mutant-specific activity, contribute to altered layer neuron identity determinant expression through a gain-of-function mechanism. In addition, they further imply that TDP-43 overexpression can act cell-autonomously to control layer IV/V identity determinant expression in developing cortical neurons in the pS.

To determine whether similar effects could be observed with TDP-43 in vivo and would extend to Pum2 loss of function, we performed IUEs. All electroporated plasmids used the pNeuroD promoter, which drives expression post-mitotically in newly born neurons and is not expressed in progenitors (*Guerrier et al., 2009*). We confirmed the efficiency of Pum2 deletion and hTDP-43 overexpression by double staining for GFP to label electroporated neurons and either for Pum2 or hTDP-43 (*Figure 7—figure supplement 1*). As shown in *Figure 7*, electroporating plasmids at E13.5, encoding either pNeuroD-Cre in *Pum2*$^{fl/fl}$ mice or WT TDP43 or the ALS-derived mutant TDP43$^{A315T}$ in WT mice, respectively, was sufficient to cell-autonomously drive a switch in expression of Sox5, Bcl11b, and Rorβ in pS. Post-mitotic deletion of Pum2 (*Figure 7a*) and expression of either WT hTDP-43 or the hTDP-43$^{A315T}$ mutant (*Figure 7b*) in newly born deep-layer cortical neurons led to robust induction of Sox5 and Bcl11b proteins and reduction of Rorβ protein expression; accordingly, mutant TDP-43 showed a slightly stronger effect than WT TDP-43. These in vivo results obtained after IUE are strikingly reminiscent of those seen in *Pum2* cKO or TDP43$^{A315T}$ mice (*Figure 1*, *Figure 1—figure supplement 5a*) or after transfection of pS-enriched primary neurons (*Figure 6*) and provide another line of experimental evidence that loss of Pum2 and gain of TDP-43 function, respectively, yield these phenotypes.

Electroporation of the same plasmids at E14.5 to target upper layer neurons revealed that the fate of these neurons could not be altered by either loss of Pum2 function or gain of TDP-43 function (*Figure 7—figure supplement 2*), consistent with our earlier data with mutant and transgenic

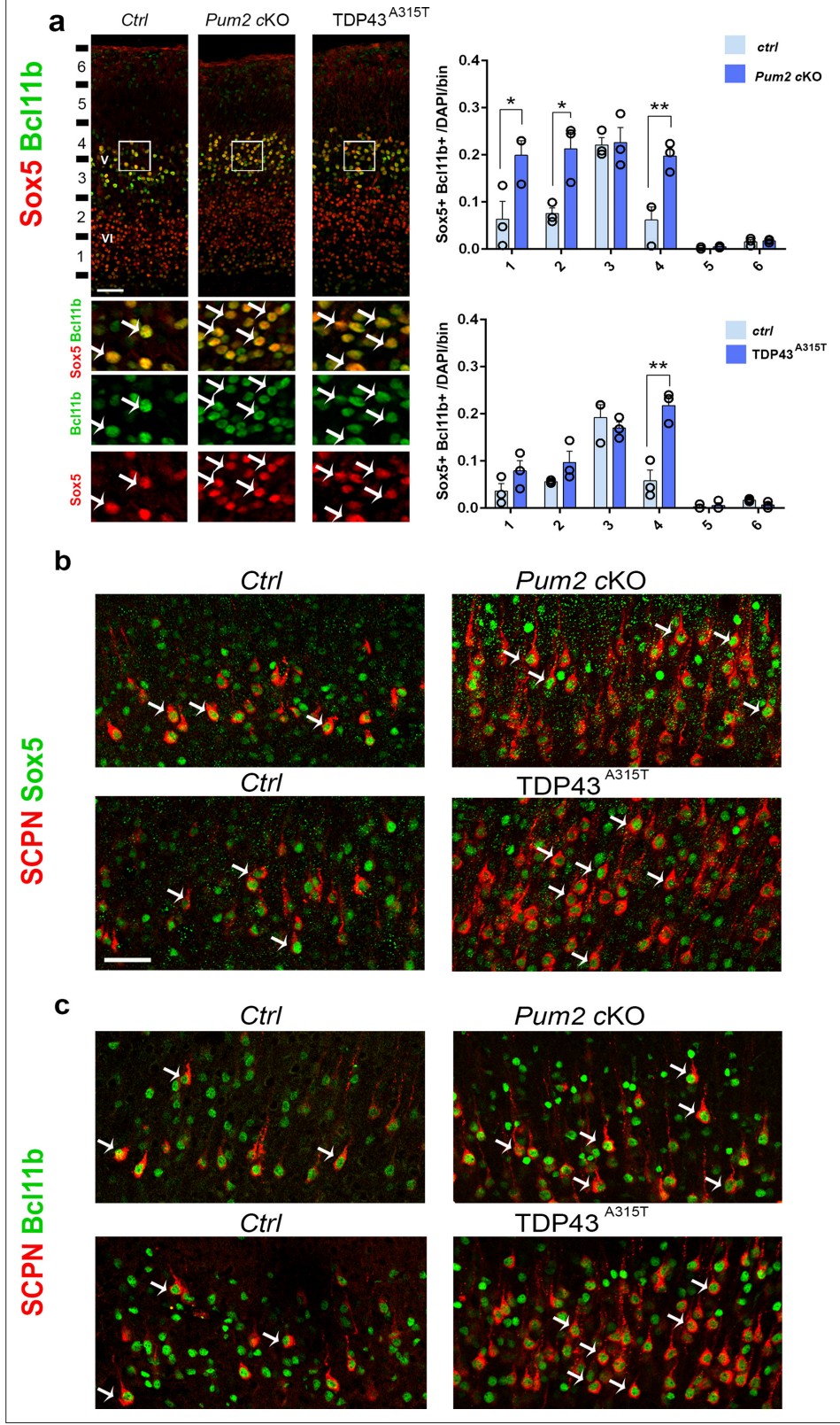

**Figure 4.** Subcerebral projection neurons' (SCPNs') increase colocalizes with Sox5 and Bcl11b in Pum2 and TDP-43 mutants. (**a**) Coronal sections from neonatal (P0) brains of controls (*Ctrl*), *Pum2* cKO, or *Prnp-TARDBP^A315T* (TDP43^A315T) mice were stained with antibodies recognizing Sox5 and Bcl11b in the prospective somatosensory area (pS). Quantification of Sox5 and Bcl11b colocalization from n = 3 mice of each genotype is shown to the right

*Figure 4 continued on next page*

*Figure 4 continued*

across six equal-sized bins. (**b, c**) Coronal sections from primary somatosensory cortex (S1) of controls, *Pum2* cKO, and TDP43[A315T] mice at P7 traced for SCPNs combined with Sox5 (**b**) or Bcl11b (**c**) staining. White arrows in (**a–c**) indicate colocalization. Data are shown as means ± standard error of the mean (SEM), n = 3 for each genotype. *p≤0.05, **p≤0.01, two-tailed *t*-test. *Pum2* cKO: *Pum2[fl/fl]*; *Emx1[Cre]*; V, VI: layers V and VI. Scale bars: 100 µm.

lines. Electroporated upper-layer neurons with either pNeuroD-Cre in *Pum2[fl/fl]* mice or WT TDP43 or TDP43[A315T] in WT mice did not show ectopic Sox5 or Bcl11b expression, exactly like neurons electroporated with the control pNeuroD-IRES GFP. Collectively, our results with transfection of primary neurons from the pS and IUE of developing pS in vivo support the idea that TDP-43 gain of function and Pum2 loss of function can cell-autonomously and post-mitotically change the relative expression of known molecular determinants of layer IV/V neuronal identity in the pS of developing neocortex.

## Evidence that Pum2 and TDP-43 probably use post-transcriptional mechanisms to regulate layer IV/V neuronal identity determinants

We next investigated the molecular mechanisms used by Pum2 and TDP-43 to control area-specific neuronal identity and connectivity in S1. For these studies, we again focused for simplicity on the TDP43[A315T] line since it showed quantitatively stronger phenotypes relative to WT TDP43. Because both RBPs are known to post-transcriptionally regulate their target mRNAs, we first analyzed mRNA levels for the previously characterized layer IV/V molecular determinants *Sox5*, *Bcl11b*, and *Rorb*, which we showed in *Figure 1* and *Figure 1—figure supplement 9* to have an increased or decreased number of cells positive for these proteins in the neocortex of *Pum2* cKO mice and in mice overexpressing either hTDP-43[A315T] or WT hTDP-43 protein transgenically. In parallel, we also analyzed mRNA levels for *Fezf2*, a master regulator of subcerebral identity, which functions upstream of Bcl11b to specify the fate of layer V subcerebral neurons (*Chen et al., 2005*; *Chen et al., 2008*; *McKenna et al., 2011*; *Molyneaux et al., 2005*; *Rouaux and Arlotta, 2013*). qRT-PCR with RNA obtained from dissected pS-enriched neocortex at P0 indicated no significant differences in steady-state mRNA levels of any of these mRNAs in either *Pum2* cKO or TDP43[A315T] relative to littermate controls (*Figure 8a and b*; *Table 1*). This suggests that altered mRNA levels are not likely to be the basis for altered levels of Sox5, Bcl11b, and Rorβ proteins in S1, although effects within specific cell types might potentially be missed in a pS-wide assay.

Next, we sought to confirm the results from our qRT-PCR assays using an independent method with higher spatial resolution. To this end, we performed RNA-specific single-molecule fluorescent in situ hybridization (smFISH) to enable quantification of mRNA levels within newly born neurons in specific layers of the pS. To enable direct comparison, these experiments were also performed at P0, the time when protein levels and cell fate were strongly altered in the *Pum2* cKO and TDP43 transgenic lines. In order to determine whether a change in mRNA levels within specific layer neurons might explain the protein level changes observed by antibody staining in *Figure 1*, we hybridized specific antisense probes to *Sox5*, *Bcl11b*, *Rorb*, and *Fezf2* mRNAs and analyzed mRNA levels in pS using the same binning approach, but now for the mRNAs (*Figure 8c*). Specifically, we counted the number of smFISH dots, which correspond to single mRNAs, in specific regions of the pS for confocal images obtained from each genotype. As shown in *Figure 8d*, this revealed no significant differences in the levels of *Sox5*, *Bcl11b*, or *Fezf2* mRNA levels in either mutant relative to controls. In contrast, we observed a paradoxical increase in *Rorb* mRNA in both genotypes (*Figure 8d*), even though Rorβ protein levels were decreased (*Figure 1*). Taken together, our qRT-PCR and smFISH data do not reveal evidence for transcriptional or stability effects on mRNA levels. Accordingly, such changes may therefore not be the reason for altered Sox5, Bcl11b or Rorβ protein levels in the pS of *Pum2* cKO or TDP43[A315T] mutants. .

It is important to note that we cannot exclude potential effects on transcription and/or mRNA stability in neuronal subpopulations that might be missed in our bulk assays. Moreover, smFISH may not be sufficiently quantitative to detect these effects in situ. Potential caveats notwithstanding, these orthogonal assays provide reasonable evidence that the protein-level phenotypes may result from post-transcriptional effects impinging on mRNA translation and/or protein stability.

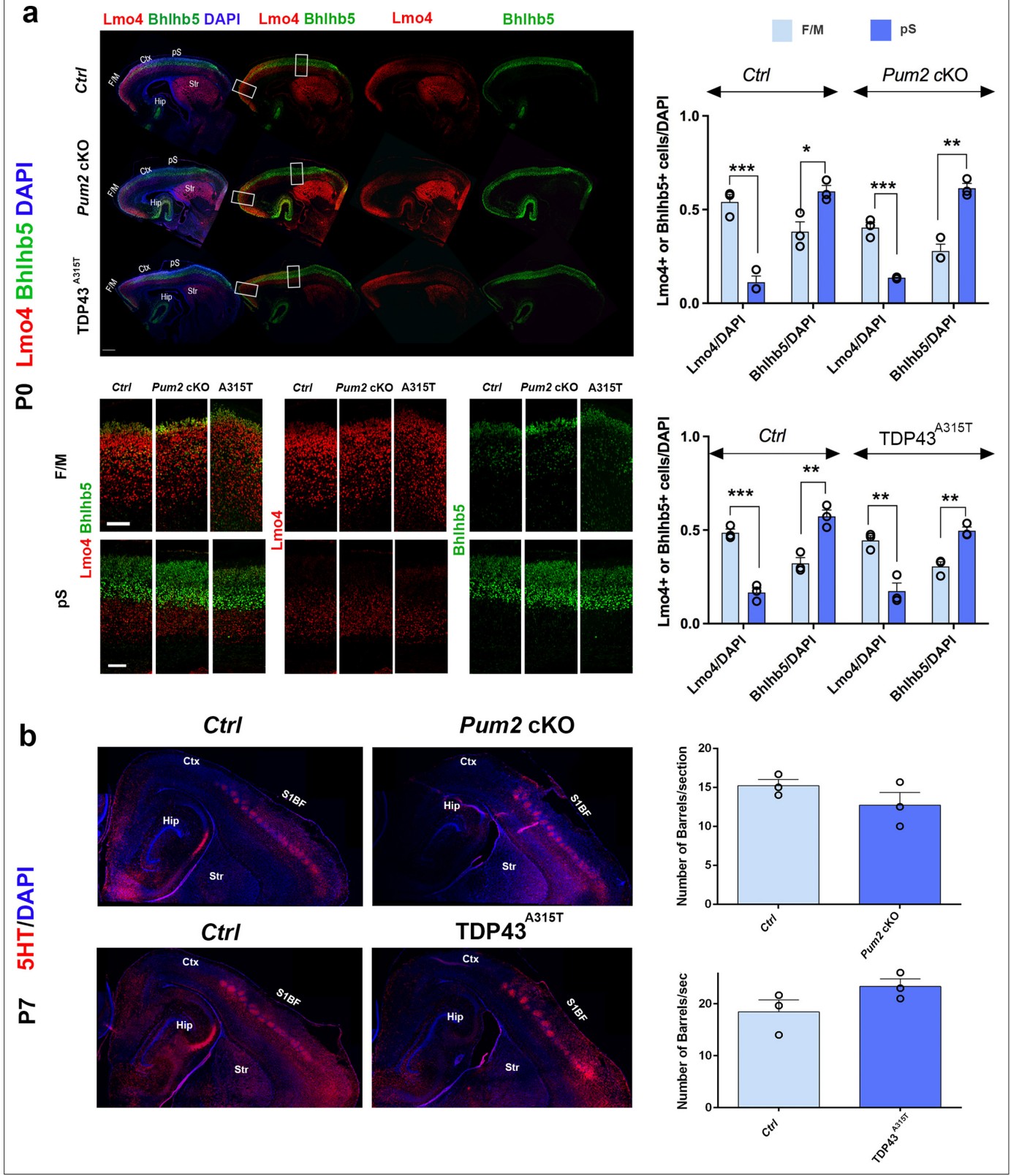

**Figure 5.** Somatosensory area identity is properly determined in *Pum2* cKO and TDP43[A315T] mutants despite layers IV and V being 'motorized.'. (**a**) Coronal sections of one brain hemisphere from controls (*Ctrl*), *Pum2* cKO, and TDP43[A315T] brains at P0 co-immunostained for Lmo4 and Bhlhb5. Selected regions are marked by white rectangles in the upper panel, and high-magnification views of frontal motor (F/M) and prospective somatosensory (pS) areas are shown below. Scale bars: 400 μm and 100 μm, respectively. Quantification of results is shown to the right. (**b**) Sagittal sections from controls,

*Figure 5 continued on next page*

Figure 5 continued

Pum2 cKO (top), and TDP43[A315T] (bottom) at P7 immunolabeled for serotonin (5HT). Quantification of the number of barrels per section is shown to the right. Scale bar: 100 μm. Data are shown as means ± standard error of the mean (SEM), n = 3 for each genotype. *p≤0.05, **p≤0.01, ***p≤0.001, two-tailed t-test. Pum2 cKO: Pum2[fl/fl]; Emx1[Cre]; Ctx: cortex; Hip: hippocampus; Str: striatum; S1BF: barrel field region of S1. Scale bar: 100 μm.

## No detectable tissue-wide effects of Pum2 or TDP-43 on splicing or polyadenylation site usage of *Sox5*, *Bcl11b*, or *Rorb* mRNAs in developing neocortex

Our qRT-PCR and smFISH data did not provide evidence for mRNA-level changes as the basis for the observed protein-level changes in layer IV/V of pS. We therefore considered whether other RNA regulatory mechanisms might underlie changes in Sox5, Bcl11b, and Rorβ protein levels. Thus, we next examined the potential effects on alternative pre-mRNA splicing and alternative 3′ end formation/polyadenylation (APA), two post-transcriptional regulatory mechanisms that can indirectly affect translation and/or protein stability and are implicated in the control of brain development (*Furlanis and Scheiffele, 2018*; *Hermey et al., 2017*; *Nguyen et al., 2016*; *Zheng and Black, 2013*). Consistent with its cytoplasmic localization, Pum2 has not been implicated in either of these nuclear pre-mRNA processing events. However, numerous studies have demonstrated alternative splicing regulation by TDP-43, including an analysis of the transgenic TDP43 line that we examined here (*Arnold et al., 2013*; *Lagier-Tourenne et al., 2012*; *Polymenidou et al., 2011*; *Tollervey et al., 2011*). Moreover, TDP-43 knockdown in cultured cell lines has also been shown to affect APA site usage (*Rot et al., 2017*), suggesting that this might also potentially occur with hTDP-43 overexpression in the intact developing brain. We therefore examined the potential effects on splicing and APA in pS-enriched neocortex of the *Pum2* cKO and TDP43[A315T] lines.

Focusing initially on pre-mRNA splicing, we designed primers to specific splice variants of *Sox5*, *Bcl11b*, and *Rorb* (*Table 2*) annotated in the Ensembl release 98 database for mouse (GRCm38.p6) (*Zerbino et al., 2018*). As shown in *Figure 9—figure supplement 1a and b*, we detected expression of these mRNA variants at different levels in pS at P0 using this approach, consistent with alternative splicing occurring in this tissue. However, we did not observe any significant changes in their levels relative to littermate controls in tissue from either *Pum2* cKO or TDP43[A315T] mice. To further probe the potential effects on alternative splicing of *Sox5* mRNA with an independent approach, we used previously described RT-PCR primer sets (*Table 3*) that produce different amplicon sizes resolvable by agarose gel electrophoresis depending on alternative splicing (*Edwards et al., 2014*). Consistent with qRT-PCR, this approach revealed that mRNA variants previously characterized in non-neuronal tissues are also generated by alternative splicing in developing neocortical pS. However, these splicing patterns were not altered significantly in either *Pum2* cKO or TDP43[A315T] mice (*Figure 9—figure supplement 1c*). These data suggest that there are no significant tissue-wide effects on splicing of *Sox5*, *Bcl11b*, or *Rorb* mRNAs in the neocortical pS of either *Pum2* cKO or TDP43[A315T] mice.

After not finding any significant tissue-wide effects on alternative splicing of key determinants of layer IV/V neuronal identity, we next examined the potential effects on mRNA 3′ end formation via APA. Transcript isoforms with different 3′ ends were annotated in the Ensembl release 98 database for mouse (GRCm38.p6) (*Zerbino et al., 2018*) for *Sox5*, *Bcl11b*, and *Rorb* (*Table 4*), and we confirmed expression of these isoforms in the pS at P0 by qRT-PCR using specific primer sets (*Figure 9—figure supplement 2a*). This revealed clear differences in the relative expression of the isoforms in the developing pS at baseline, but no significant changes in the relative levels of the mRNA isoforms in the mutant lines relative to their respective littermate controls (*Figure 9—figure supplement 2b*). We conclude that APA of *Sox5*, *Bcl11b*, and *Rorb* mRNAs is not generally affected in the pS area of developing neocortex in either *Pum2* cKO or TDP43[A315T] mice.

While we cannot rule out subtle effects in neuronal subpopulations that might be missed in our tissue-wide assay, these results with pS-enriched RNA and isoform-specific primers do not support either alternative splicing or APA of *Sox5*, *Bcl11b*, and *Rorb* mRNAs as a likely basis for effects on the corresponding proteins observed in *Pum2* cKO or hTPD-43 transgenic mice.

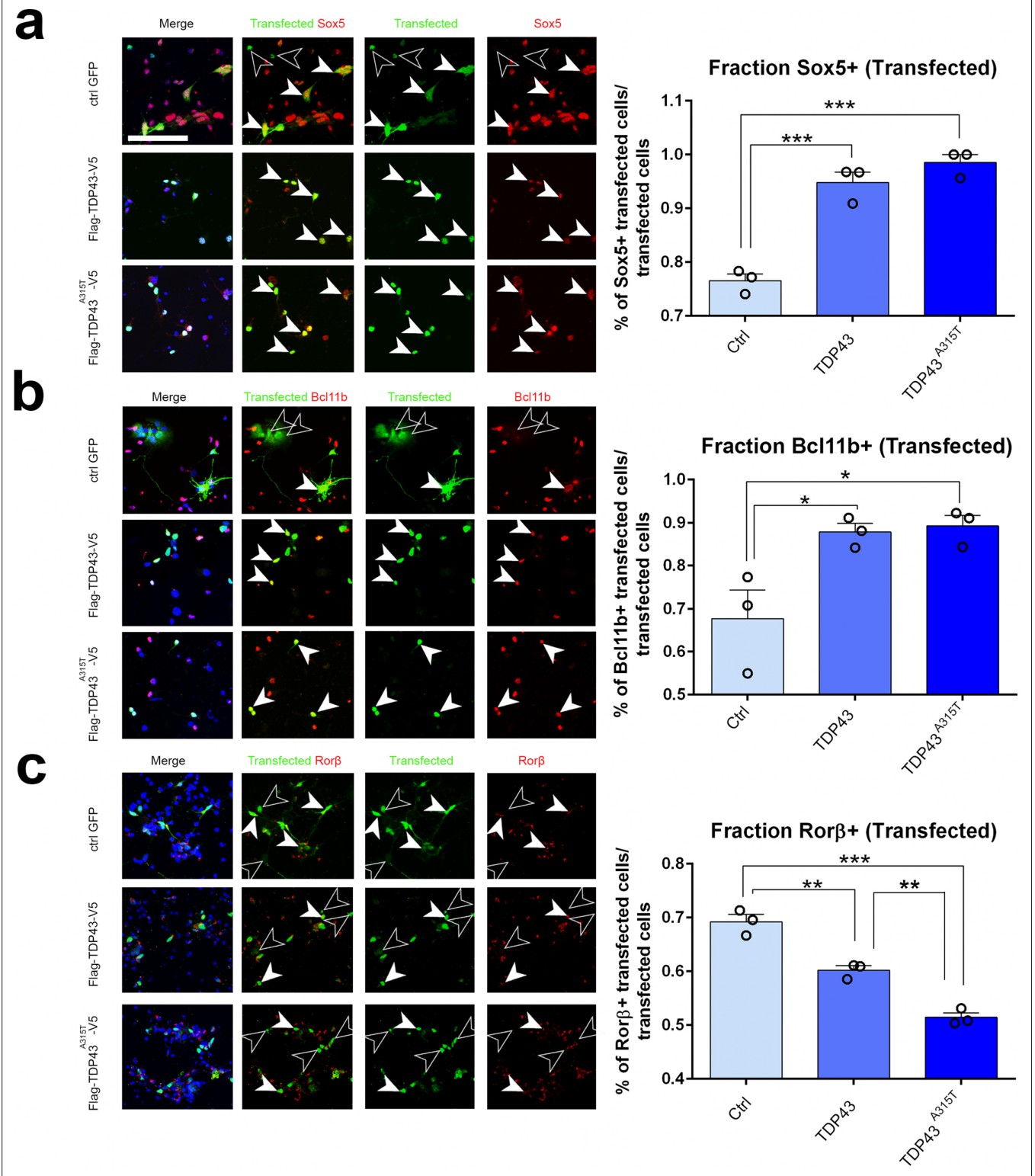

**Figure 6.** TDP-43 gain of function cell-autonomously regulates layer IV and V molecular determinants in vitro. (**a–c**) Primary neurons harvested from WT cortical lysates enriched for somatosensory cortex at E18.5 were transfected before plating with plasmids encoding either control GFP, TDP43, or TDP43[A315T]. After 48 hr in culture, neurons were fixed and stained with antibodies recognizing GFP to label control transfected neurons or recognizing either the Flag (**a, b**) or V5 (**c**) epitope tag to label neurons transfected with either TDP43 or TDP43[A315T]. All transfected neurons were co-immunolabeled with antibodies recognizing Sox5 (**a**), Bcl11b (**b**), or Rorβ (**c**) and with DAPI. Quantification of the fraction of Sox5[+], Bcl11b[+], or Rorβ[+] neurons among all transfected neurons is shown to the right of the representative images. At least 50 cells were counted for each replicate of every transfection. Data are shown as means ± standard error of the mean (SEM), n = 3 for each transfection. *p≤0.05, **p≤0.01, ***p≤0.001, two-tailed *t*-test. Scale bar: 100 µm.

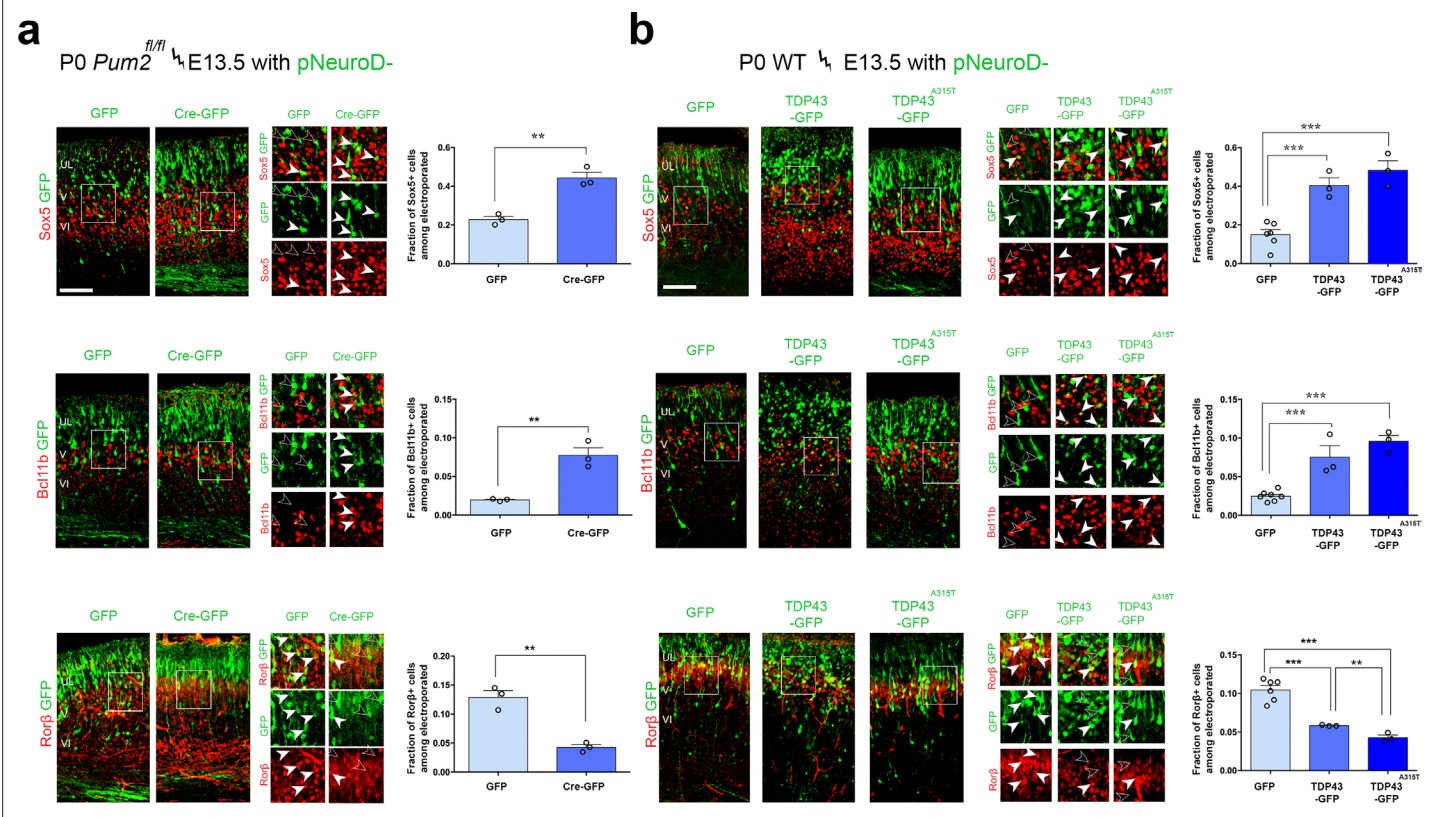

**Figure 7.** RNA-binding proteins Pum2 and TDP-43 regulate layer IV and V molecular determinants post-mitotically and cell-autonomously in vivo. (**a, b**) Coronal sections from *Pum2*$^{fl/fl}$ (**a**) or WT (**b**) brains at P0 electroporated at E13,5 with pNeuroD-IRES-GFP as control, or with p-NeuroD-IRES-Cre-GFP to ablate Pum2 expression (**a**) or p-NeuroD-TDP43-IRES-GFP or p-NeuroD-TDP43$^{A315T}$-IRES-GFP to overexpress hTDP-43 alleles only in post-mitotic neurons. Sections are co-stained with antibodies recognizing GFP to label electroporated neurons and antibodies recognizing Sox5, Bcl11b, or Rorβ. High-magnification views are shown to the right. White arrowheads indicate examples of electroporated neurons expressing Sox5-, Bcl11b-, and Rorβ-positive neurons while empty arrowheads indicate electroporated neurons not expressing these proteins. Quantification of the fraction of Sox5$^+$, Bcl11b$^+$, or Rorβ$^+$ neurons among all electroporated cells is shown to the right of the representative images. Data are shown as means ± standard error of the mean (SEM), n = 3 for each electroporation. Both p-NeuroD-IRES-Cre-GFP and hTDP-43 alleles were co-electroporated with T-dimer (red) to distinguish them from littermate control brains electroporated only with pNeuroD-IRES-GFP. For both hTDP-43 alleles, the respective control littermates for each variant were combined to a total of n = 6 for pNeuroD-IRES-GFP electroporations. **p≤0.01, ***p≤0.001, two-tailed *t*-test. UL, V, VI: upper layers, layers V and VI. Scale bar: 100 μm.

The online version of this article includes the following figure supplement(s) for figure 7:

**Figure supplement 1.** Validation of electroporation efficiency.

**Figure supplement 2.** Upper-layer neuronal identity is not affected in Pum2 and TDP-43 mutants.

## Evidence for both translational activation and repression of *Sox5*, *Bcl11b*, and *Rorb* mRNAs by Pum2 and TDP-43 in developing neocortex

We next considered whether there might be specific effects on the translation of *Sox5*, *Bcl11b*, and *Rorb* mRNAs. To examine the potential effects of Pum2 and hTDP-43$^{A315T}$ on translation in developing neocortex, we used sucrose density gradient fractionation-based polysome profiling of neocortical lysates from mutants and littermate controls. This classic biochemical fractionation method can reveal changes in the relative number of ribosomes engaged with cellular mRNAs on a global and mRNA-specific level (*Figure 9a and b*). Importantly, because the percentage of total RNA signal in the different fractions is plotted, changes in an mRNA's translational status in this assay are unrelated to the mRNA levels themselves.

Unlike mice lacking the RBP HuR, which show strong defects in brain development that correlated with effects on both general and mRNA-specific translation (*Kraushar et al., 2014*), we observed no

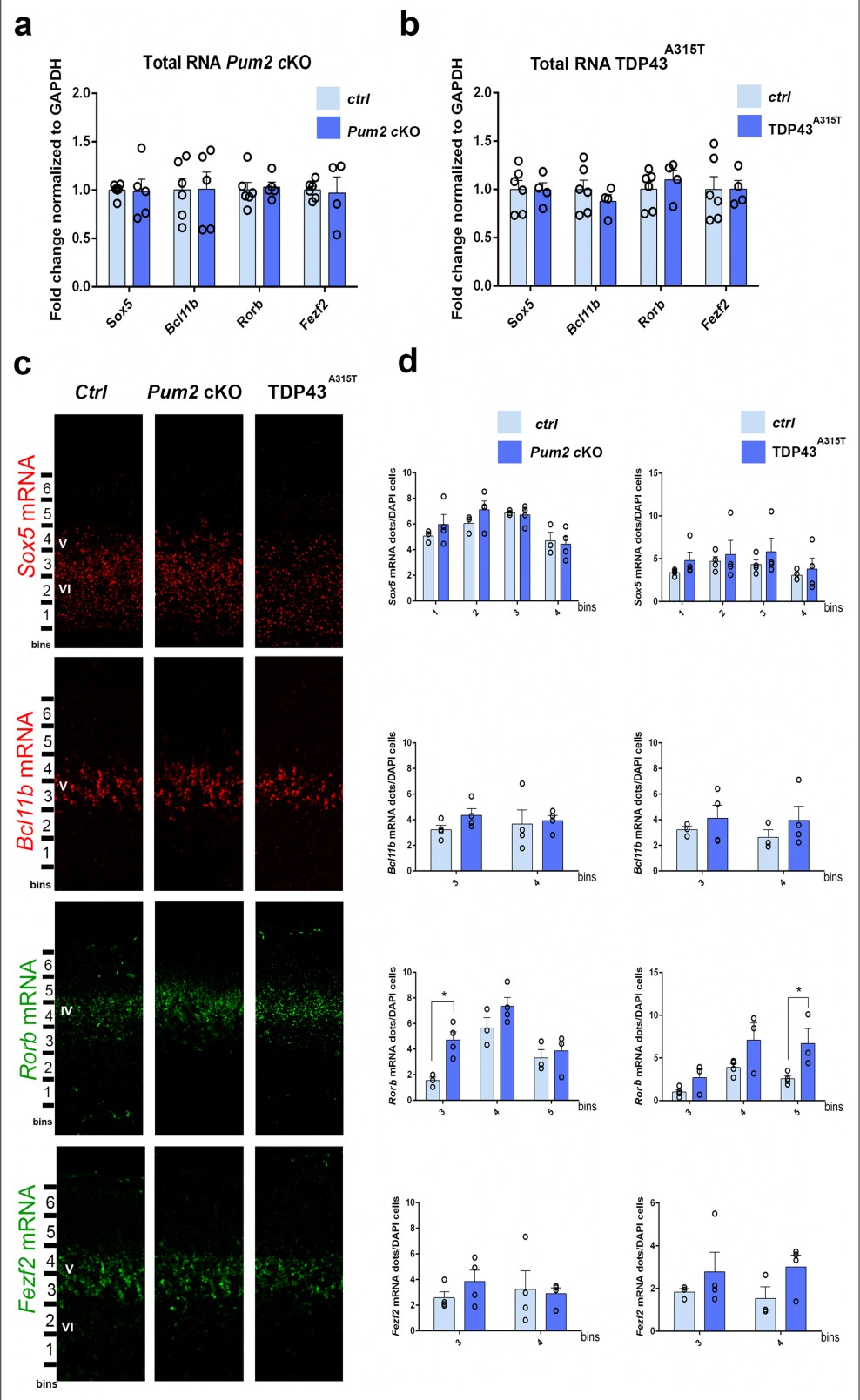

**Figure 8.** mRNA levels of layer IV/V neuronal identity determinants remain unchanged in *Pum2* cKO or TDP43[A315T] mutants. qRT-PCR of RNA derived from P0 somatosensory area-enriched cortical lysates for *Pum2* cKO (**a**) or TDP43[A315T] (**b**). The fold change for *Sox5, Bcl11b, Rorb,* and *Fezf2* mRNAs normalized to *GAPDH* mRNA is shown for mutants relative to respective control samples (*Ctrl*). Data are displayed as means ± standard error of the

*Figure 8 continued on next page*

*Figure 8 continued*

mean (SEM) for at least n = 4 of each genotype. (**c**) Single-molecule fluorescent in situ hybridization (smFISH) for *Sox5*, *Bcl11b*, *Rorb*, and *Fezf2* mRNAs on coronal sections from the prospective somatosensory area (pS) of controls (*Ctrl*), *Pum2* cKO, and TDP43[A315T] mice at P0. Distribution of cells across six equal-sized bins is shown. (**d**) Quantification from (**c**). The number of RNA dots in the bins where they are mostly expressed is normalized to the total number of cell nuclei (DAPI) within that bin. Data are shown as means ± SEM, at least n = 3 for each genotype. *p≤0.05 by two-tailed *t*-test. *Pum2* cKO: *Pum2*[fl/fl]; *Emx1*[Cre]; IV, V, VI: layers IV, V and VI, respectively. Scale bar: 100 μm.

differences in the overall polysome profiles in *Pum2* cKO or TDP43[A315T] neocortices (***Figure 9—figure supplement 3a***), suggesting that general translation is not strongly affected in this tissue in these lines. We next used qRT-PCR from the polysome fractions to investigate mRNA-specific translational regulation, normalizing to an in vitro transcribed *Renilla* luciferase (RLuc) 'spike-in' standard that we added to the fractions prior to RNA purification (***Figure 9c and d***). As noted above, multiple 3′UTR isoforms of *Sox5*, *Bcl11b*, and *Rorb* mRNAs with different numbers of predicted binding sites for Pum2 and/or TDP-43 are annotated in the Ensembl release 98 database for mouse (GRCm38.p6) (***Zerbino et al., 2018***), and we found that a mixed population of transcripts appears to be expressed in developing neocortex (***Figure 9—figure supplement 2a and b***). In our sucrose density gradient polysome profile analyses, we focused on primer sets (***Tables 1 and 4***) recognizing mRNA isoforms with predicted binding sites for Pum2 and/or TDP-43 (***Figure 9—figure supplement 2a***) wherever possible because material was limited and we reasoned that this would improve sensitivity in this bulk tissue assay.

Results for polysome profiling for TDP43[A315T] from whole neocortices at E14.5, the peak time of birth for layer IV neurons, are shown in ***Figure 9c***. For the specific *Sox5* and *Bcl11b* mRNA isoforms examined, we observed a significant shift in the percentage distribution to heavier polysome fractions, consistent with an increased number of ribosomes translating these mRNAs in the mutant transgenic line. Strikingly, *Rorb* mRNA was regulated in exactly the opposite manner, showing a significant shift to a lighter gradient fraction corresponding to approximately one ribosome/mRNA (i.e., monosomes) in TDP43[A315T] compared to the percentage of mRNA signal present in a heavier fraction (corresponding to greater than approximately seven ribosomes/mRNA). This pattern is consistent with a reduced number of ribosomes engaged with this mRNA in mutant neocortex. In contrast, no significant differences were found with *Fezf2* mRNA, highlighting apparent specificity of the effects on the other layer IV/V identity determinants.

**Table 1.** qRT-PCR primers.

| mRNA | Forward primer (5′–3′) | Reverse primer (5′–3′) |
| --- | --- | --- |
| *Sox5* | CCAGGACTTGTCTTTCCAG | CCCTGAAGCAGAGGAAGATG |
| *Bcl11b* | AAGCCATGTGTGTTCTGTGC | AAAGGCATCTGTCCAAGCAG |
| *Rorb* | ATGCCAGCTGATGGAGTTCT | TAGCTCCCGGGGATAACAATG |
| *Fezf2* | GTGGCTCCCACCTTTGTACATTCA | TCACGGTGACAGGCTGGGATTAAA |
| *Cux1* | CCTGCAGAGTGAGCTGGAC | GCTTGCTGAAGGAGGAGAAC |
| *Gapdh* | TTGATGGCAACAATCTCCAC | CGTCCCGTAGACAAAATGGT |
| *Pum2* | CCCCGAGATTCTAATGCAAG | CTGGAAGAAGCACGGTGAAT |
| *Pum2 exons 6&7* | ATTGGGCCCTCTTCCTAATC | CCAACTTGGTCCATTGCAT |
| *Tardbp* | CGTGTCTCAGTGTATGAGAGGAGTC | CTGCAGAGGAAGCATCTGTCTCATCC |
| *Emx1* | ACCATAGAGTCCTTGGTGGC | TGGGGTGAGGATAGTTGAGC |
| *Sox6* | GCATAAGTGACCGTTTTGGCAGG | GGCATCTTTGCTCCAGGTGACA |
| *Unc5C* | ACTCAATGGCGGCTTTCAGCCT | GGTCCAGAATTGGAGAGTTGGTC |
| *18s rRNA* | CTTAGAGGGACAAGTGGCG | ACGCTGAGCCAGTCAGTGTA |
| *Rluc* | TGGTAACGCGGCCTCTTCT | GCCTGATTTGCCCATACCAA |

**Table 2.** qRT-PCR splicing isoforms primers.

| mRNA | Forward primer (5′–3′) | Reverse primer (5′–3′) |
| --- | --- | --- |
| Sox5 204 | CGTACATGATACGTCCTCCC | CCAGCCCCACTGTTTATTC |
| Sox5 206 | CTTGAGGTTTGTTCTCCTCTG | GCCATAGTGGTTGGGATCAG |
| Sox5 211 | GTACATGATACGTCCTCCCC | TCTTGTCTGTGTGAATGCTG |
| Sox5 diff | ATGCTTACTGACCCTGATTTAC | TCTCACTCTCCTCCTCTTCC |
| Bcl11b 201 | CAGTGTGAGTTGTCAGGTAAAG | GCTCCAGGTAGATTCGGAAG |
| Bcl11b 202 | TCCCAGAGGGAACTCATCAC | GCTCCAGGTAGATTCGGAAG |
| Bcl11b 203 | CCTACTGTCACCCACGAAAG | GCTCCAGGTAGATTCGGAAG |
| Rorb 201 | CTGCACAAATTGAAGTGATACC | AAACAGTTTCTCTGCCTTGG |
| Rorb 202 | AAGCATAGCACGCAGCACTC | ATCCCGGAGGATTTATCGCCAC |
| Rorb 203 | AGCGGAATTTTTGGGTTCTC | ACGTGATGACTCCGTAGTG |

Unlike with TDP43[A315T], for *Pum2* cKO neocortices, we did not observe significant effects on mRNA translational status in the polysome assay at E14.5 (*Figure 9—figure supplement 3b*). Therefore, we performed additional polysome analyses at other stages to gain insight into when during development translational control by Pum2 might be detectable using this assay. At P0, when neurogenesis is complete (*Buratti and Baralle, 2014*; *Chen et al., 2008*) and it is technically possible to enrich for the pS by dissection (*Figure 9b*), we found evidence for increased ribosome engagement with *Sox5* and *Bcl11b* mRNAs and reduced ribosome density on *Rorb* mRNA (*Figure 9d*). Conversely, we did not observe an effect on *Sox5*, *Bcl11b, or Rorb* mRNA translation at E13.5, the peak birth time for layer V neurons (*Greig et al., 2013*; *Molyneaux et al., 2007*), or at E18.5 (*Figure 9—figure supplement 3b*). Taken together, these data support the idea that translational regulation of the mRNAs encoding Sox5, Bcl11b, and Rorβ proteins by Pum2 may begin after birth and is therefore more likely to occur in post-mitotic neurons, rather than in neuronal progenitors. Consistent with this idea, when we directly examined nascent neurons of *Pum2* cKO mice at E13.5, we did not observe increased protein expression of regulators of layer VI and V neuronal identity Sox5, Bcl11b, or Tbr1 (*Figure 9—figure supplement 3c*), providing further evidence that regulation might be post-mitotic, rather than at progenitor level. Because our polysome gradient assay detects changes in translational status independently of mRNA levels and because we did not find any evidence of corresponding effects on mRNA levels, splicing, or polyadenylation of these mRNAs, we conclude that increased translation of *Sox5* and *Bcl11b* mRNAs, together with decreased translation of *Rorb* mRNA, is likely to be at least one molecular mechanism contributing to the corresponding changes detected at the protein level in developing pS of the *Pum2* cKO and TDP43[A315T] lines. Taken together, our genetic and

**Table 3.** Sox5 isoforms PCR primers.
Each forward primer has its reverse primer below. F: forward; R: reverse.

| Allele | Primer (5′–3′) | Predicted size (bp) |
| --- | --- | --- |
| mSox5-346F | CCT TTC ACC TTC CCT TAC ATG | 833 |
| mSox5-1178R | AGC AGC TGC CAT AGT GGT TG | |
| mSox5-512F | CAA CTC ATC TAC CTC ACC TCA G | 457 |
| mSox5-968R | CAG AAG CTG CTG CTG TTG | |
| mSox5-899F | ACA GCG TCA GCA GAT GGA G | 637 |
| mSox5-1535R | GCT AAC TCT TGC AGA AGG AC | |
| mSox5-1426F | CTG CAT CAC CCA CCT CTC | 535 |
| mSox5-1960R | CTG ATG TTG GAA TTG TGC ATG | |

**Table 4.** qRT-PCR 3′UTR isoforms primers.

| mRNA | Forward primer (5′–3′) | Reverse primer (5′–3′) |
| --- | --- | --- |
| *Sox5 S1* | GCCGTTCTCAGGTGAAAAGA | GCCTGACATTATTCCCCAAT |
| *Sox5 S2* | CAGACAACTGCAGCCACTTC | TTGGCAACATGAGAGGACTG |
| *Sox5 S3* | TAGGTCACTTGGGGGAAAGC | GCAAGGGCATTGTGTTGTTA |
| *Sox5 S4* | TGCAAACTACCATCTCACTTG AA | TGGCATGAATGATAACATAAAA CC |
| *Bcl11b B1* | GGACGGGAAAATGCCATAAG | AAGTCACCTCCACTCCATATC |
| *Bcl11b B2* | TACCCTGCCCTTTTGACACC | TTGACAGAGACACACAAGTCC |
| *Rorb R1* | GGAAAACAGGGTAATGGAAGG | GGGAACATCAAGTAGACACAG |
| *Rorb R2* | AAATATGTACTCGCTCCCTTTC | AGCCCTGTCCCTTTCTTAG |

biochemical data establish a correlation between effects on the translational status of *Sox5*, *Bcl11b*, and *Rorb* mRNAs and area-specific effects on levels of the encoded proteins in S1.

## Cytoplasmic Pum2 and TDP-43 localize with and directly bind to mRNAs encoding key regulators of layer IV/V neuronal identity in developing neocortex

We next asked whether apparent effects on translation by Pum2 and TDP-43 in developing neocortex could potentially be mediated by direct interaction of these proteins with the regulated mRNAs. Endogenous mouse TDP-43 is present in cytoplasmic lysates from neocortex at P0 (*Figure 1—figure supplement 8*), implying that it could conceivably function in the cytoplasm to regulate post-transcriptional processes such as translation at this stage. Moreover, immunostaining revealed that both Pum2 and endogenous mouse TDP-43 were detectable in the cytoplasm of both progenitors and post-mitotic neurons in the pS during early neurogenesis and postnatally (*Figure 10—figure supplement 1*). We also performed high-resolution imaging of post-mitotic neurons in layer IV/V of the developing pS using combined immunostaining/smFISH. *Bcl11b* and *Rorb* mRNAs were observed as discrete foci primarily in the cytoplasm, whereas *Sox5* mRNA foci were detected in both the nucleus and cytoplasm (*Figure 10a and b*). Cytoplasmic mRNA foci overlapped with the diffuse staining seen throughout the cytoplasm for both Pum2 and TDP-43 (*Figure 10a and b*). These data demonstrate that both these mRNAs and the RBPs that regulate them are present in the cytoplasm of post-mitotic neurons in developing pS, suggesting that they could potentially interact there in neuronal messenger ribonucleoprotein (mRNP) complexes. In addition, our immunoblot analyses in *Figure 1—figure supplement 8* demonstrated increased levels of both hTDP-43 and hTDP-43$^{A315T}$ proteins in the cytoplasm at P0. This observation is consistent with a possible gain-of-function effect of hTDP-43 in this cellular compartment.

To assess whether Pum2 and TDP-43 might directly interact with *Sox5*, *Bcl11b*, or *Rorb* mRNAs, we examined several published genome-wide binding studies for Pum2 (*Hafner et al., 2010*; *Sternburg et al., 2018*; *Uyhazi et al., 2020*) and TDP-43 (*Colombrita et al., 2012*; *Herzog et al., 2020*; *Kapeli et al., 2016*; *Narayanan et al., 2012*; *Polymenidou et al., 2011*; *Tollervey et al., 2011*). However, as far as we could tell, these mRNAs were not detected in these studies. Presumably, this is because they show a relatively specific temporal and spatial expression pattern in the developing neocortex, whereas most published studies examined cultured cell lines or whole brain/adult material from patients or mice. Interestingly, *Bcl11b* was detected in a RIP study of TDP-43 targets on E18.5 rat cortical neurons after 14 days in culture (*Sephton et al., 2011*). Moreover, iCLIP of Pum2 from neonatal mouse brain revealed interaction of Pum2 with *Sox5* mRNA and the same study found that *Bcl11b* mRNA was deregulated in brains of *Pum1/Pum2* double knockouts (*Zhang et al., 2017*). Encouraged by these positive observations, but recognizant of the inherent potential for false positives and negatives in genome-wide studies (*Williams et al., 2017*), we decided to assess potential interactions ourselves in developing neocortex using a directed approach. To this end, we adapted a directed UV-cross-link immunoprecipitation (UV-CLIP) protocol for neocortex that we used previously with cultured motor neuron-like cells (*Neelagandan et al., 2019*).

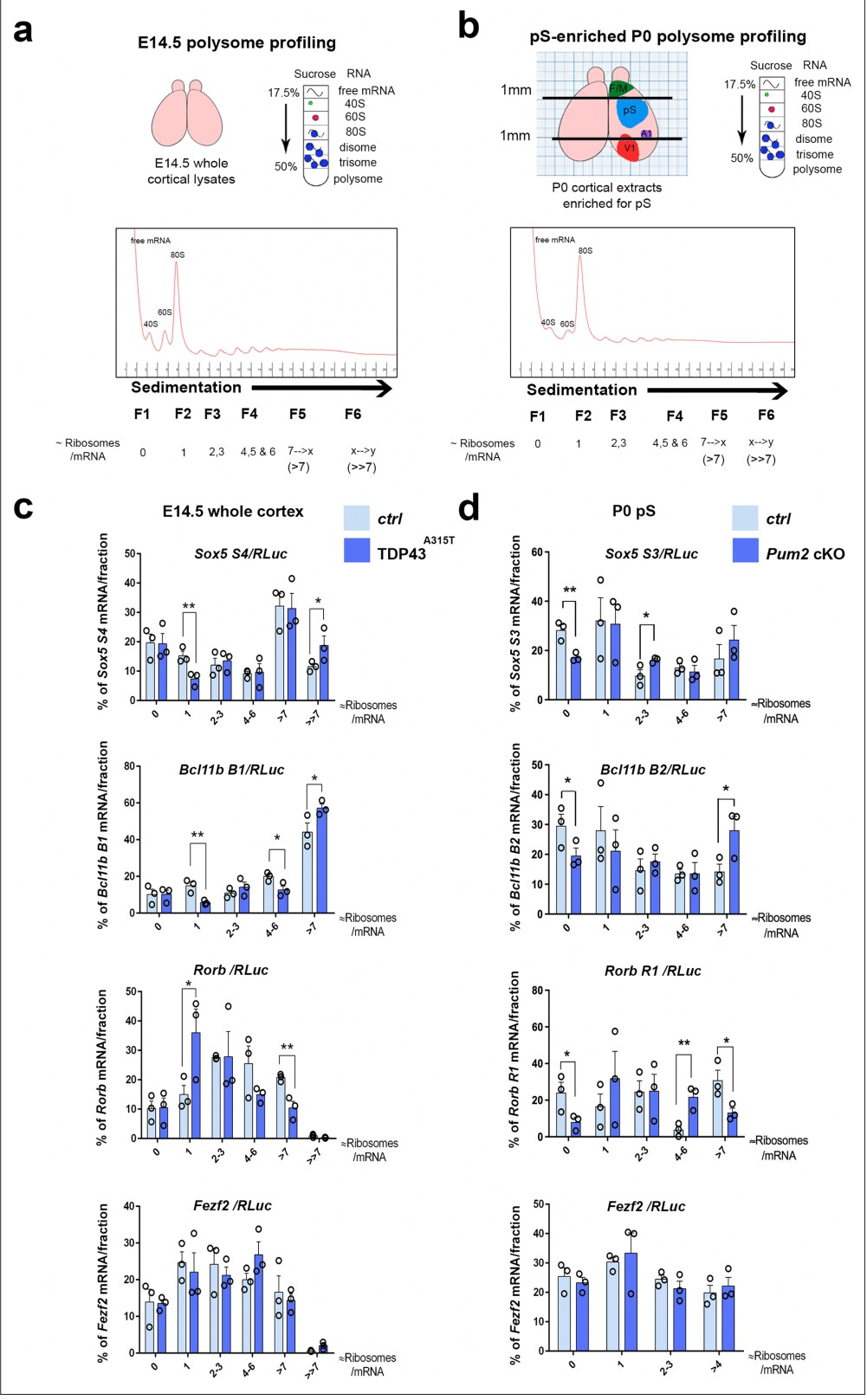

**Figure 9.** Translational control of layer IV/V neuronal identity determinants by Pum2 and TDP-43 in developing neocortex. (**a**) Schematic overview of polysome profiling for developing neocortices. Lysates from dissected E14.5 cortices were separated on polysome gradients, and RNA was prepared from fractions (F1–6) corresponding to the indicated ribosomal densities. (**b**) A schematic representation showing dissection of an enriched prospective

*Figure 9 continued on next page*

*Figure 9 continued*

somatosensory region from P0 brains using millimeter paper to eliminate 1 mm from the rostral end and 1 mm from the caudal end of cortices. Lysates for polysome profiling were made from the remaining part. F/M: frontal/motor area; pS: prospective somatosensory cortex; A1: primary auditory cortex; V1: primary visual cortex. (**c, d**) Histograms depict the distribution of the *Sox5*, *Bcl11b*, *Rorb*, and *Fezf2* mRNAs across the gradient fractions for TDP43[A315T] (**c**) and *Pum2* cKO (**d**), relative to corresponding controls (*Ctrl*). Samples in heavier gradient fractions were virtually pooled at analysis to simplify visualization in (**d**) and in the case of the *Bcl11b B1* primer in (**c**). Levels of specific mRNAs in each fraction were analyzed by qRT-PCR with normalization to an RLuc mRNA spike-in control, which was added in an equal amount to the fractions prior to RNA preparation. Data are shown as means ± standard error of the mean (SEM), n = 3 for each genotype. *p≤0.05, **p≤0.01, one-tailed *t*-test. *Pum2* cKO: *Pum2*[fl/fl]; *Emx1*[Cre].

The online version of this article includes the following figure supplement(s) for figure 9:

**Figure supplement 1.** *Sox5*, *Bcl11b*, and *Rorb* splicing is unaffected in Pum2 and TDP-43 mutant neocortices.

**Figure supplement 2.** 3'UTR isoforms with predicted binding sites for Pum2 and TDP-43 are expressed in developing neocortex, and alternative polyadenylation remains unaltered in Pum2 and TDP-43 mutants.

**Figure supplement 3.** Pum2 represses *Sox5* and *Bcl11b* mRNA translation in post-mitotic neurons.

To determine whether Pum2 and endogenous mouse TDP-43 can bind directly to *Sox5*, *Bcl11b*, and *Rorb* mRNAs in developing neocortex, we performed UV-CLIP assays with cytoplasmic lysates from wild-type C57Bl/6J mouse neocortex followed by qRT-PCR. After first verifying enrichment relative to rabbit IgG control immunoprecipitations (IPs), we next measured the percent of input RNA in the IPs, comparing this to 18S rRNA to assess biologically relevant interactions (*Figure 10c*). As a control for cross-linking dependence of detected interactions, we also included an IP from non-UV-treated lysates. Enrichment in the IP that is UV-dependent implies that direct physical interaction between the protein and the RNA tested was occurring in vivo in the neocortex prior to lysis. As expected, we found strong UV-dependent interaction of each RBP with its own mRNA, consistent with previous reports (*Ayala et al., 2011*; *Galgano et al., 2008*; *Hafner et al., 2010*; *Polymenidou et al., 2011*; *Tollervey et al., 2011*). In contrast, Pum2 interacted with *TDP-43* mRNA to a much lesser extent, and we did not detect significant interaction of endogenous mouse TDP-43 with *Pum2* mRNA, suggesting minimal cross-regulation. Both Pum2 and mouse TDP-43 showed significant interaction with *Sox5* and *Rorb* mRNAs in UV-CLIPs. Pum2 also showed significant cross-linking to *Bcl11b* mRNA, whereas for TDP-43 this was just above conventional thresholds for statistical significance. We did not see significant interaction of *Fezf2* mRNA with Pum2 or *Cux1* mRNA with either protein relative to 18S rRNA, consistent with our finding that neither these mRNAs nor the encoded proteins showed altered regulation in developing neocortex in the *Pum2* cKO or TDP43[A315T] lines. Together with our imaging assays, these directed UV-CLIP experiments support the idea that both Pum2 and TDP-43 can directly interact with specific mRNAs encoding layer IV/V neuronal identity determinants in vivo in the cytoplasm of cells in the developing neocortex. This suggests that direct interaction of Pum2 and TDP-43 with these mRNAs could potentially mediate the post-transcriptional regulatory effects described above (*Figure 9*).

## Discussion

In the neocortex, functionally related neuronal ensembles are grouped into areas specialized for processing certain types of information. Within areas, neuronal subtypes with similar projection patterns and connectivity are grouped into characteristic layers (*Rakic, 1988*; *Rash and Grove, 2006*; *Zilles and Amunts, 2010*). Although all neocortical areas have a similar six-layer architecture, layer identity and connectivity are sculpted in an area-specific manner to serve its specialized functions (*Dehay and Kennedy, 2007*). Genetic approaches in the mouse have identified many proteins that determine neocortical area identity and other proteins that control neuronal sub-specification across the cortex to give rise to the layers (*Greig et al., 2013*; *Jabaudon, 2017*; *Molyneaux et al., 2007*; *O'Leary et al., 2007*; *O'Leary and Nakagawa, 2002*; *O'Leary and Sahara, 2008*). However, a fundamental, unresolved issue is the nature of the downstream molecular mechanisms that control neuronal subtype specification in an area-specific manner. Previous work addressing this issue has highlighted roles for transcriptional regulators, such as Bcl11a/Ctip1 and Lmo4, in sculpting area-specific cytoarchitecture

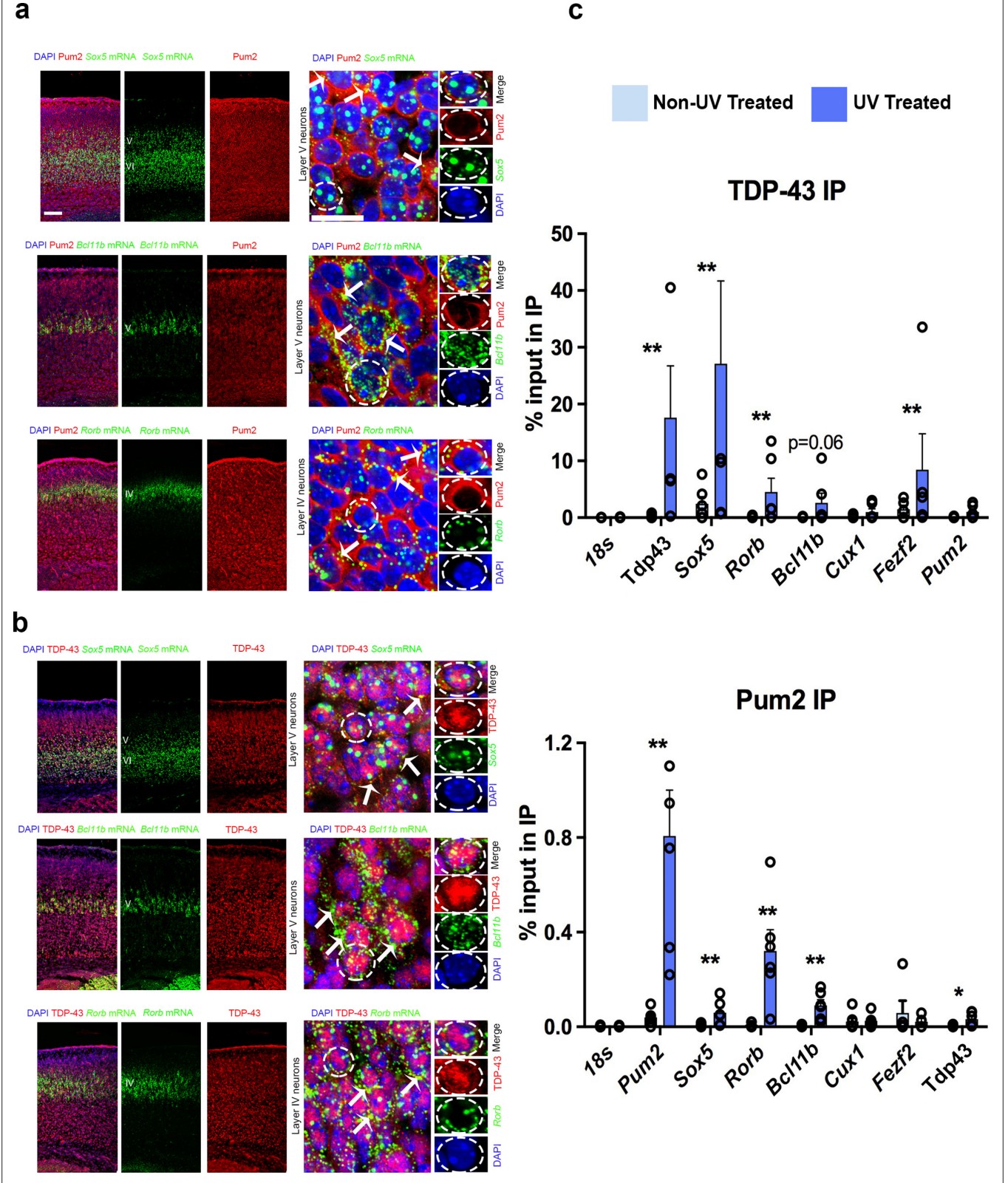

**Figure 10.** Pum2 and TDP-43 interact directly with mRNAs encoding key regulators of layer IV/V neuronal identity in developing neocortex. (**a, b**) Single-molecule fluorescent in situ hybridization (smFISH) for *Sox5, Bcl11b,* and *Rorb* mRNAs coupled with immunofluorescence for Pum2 (**a**) or TDP-43 (**b**) on coronal sections from the prospective somatosensory area (pS) of WT mice. High-magnification views taken in layer V for *Sox5* and *Bcl11b* or layer IV for *Rorb* are shown to the right. White arrows indicate examples of *Sox5, Bcl11b,* and *Rorb* mRNAs that overlap with Pum2 or TDP-43 protein

*Figure 10 continued*

immunofluorescence signal. Individual channels for a representative cell (delineated with dashed lines) are shown to the very right of each respective image. Scale bars: 25 µm. (**c**) UV Cross-linking immunoprecipitation (UV-CLIP) results from E18.5 cortices are shown. Dissociated cells were either cross-linked with UV light or left untreated as a control. Lysates were used for immunoprecipitations with antibodies against TDP-43 (top), Pum2 (bottom), or control nonspecific IgG (not shown). RNA in the input and immunoprecipitated (IP) eluate were analyzed by qRT-PCR for the indicated mRNAs. After verifying enrichment relative to IgG controls for UV-treated samples, histograms were generated that represent the fraction of input mRNA co-immunoprecipitated with either Pum2 or TDP-43 in the presence or absence of UV cross-linking. Statistically significant enrichment was evaluated relative to 18S rRNA, which is not known to interact significantly with either protein. Reduced signal in the absence of UV cross-linking implies an interaction is cross-linking-dependent, that is, direct. Data are represented as means ± standard error of the mean (SEM) from n = 3–6 samples. *p≤0.05, **p≤0.01 Mann–Whitney *U* test.

The online version of this article includes the following figure supplement(s) for figure 10:

**Figure supplement 1.** Pum2 and TDP-43 are expressed in progenitors and post-mitotic neurons in developing neocortex.

in sensory/visual or rostral motor cortex, respectively (*Cederquist et al., 2013*; *Glickfeld et al., 2013*; *Greig et al., 2016*; *Woodworth et al., 2016*). Here, we combined genetic approaches with molecular imaging and in vivo biochemical assays and generated evidence supporting a new role for post-transcriptional regulation by RBPs in elaboration of area-specific cytoarchitecture. Specifically our results reveal cell-autonomous and post-mitotic roles for the RBPs Pum2 and TDP-43 in shaping the specialized neuronal cytoarchitecture of layer IV/V that is a hallmark of the sensory cortical area, S1. Moreover, our biochemical analyses support the possibility that these RBPs achieve this regulation, at least in part, through effects on the translational status of mRNAs encoding key molecular determinants of layer IV/V neuronal identity.

The similar neurodevelopmental phenotypes in S1 and common effects on downstream molecular targets (Sox5, Bcl11b, and Rorβ) that we observed upon Pum2 loss of function or hTDP-43/hTDP-43$^{A315T}$ overexpression suggest mechanistic overlap. Collectively, our data support the notion that these two RBPs directly interact with mRNAs encoding key regulators of layer IV/V neuronal fate to regulate them post-transcriptionally, at least in part through effects on translation.

To gain insight into the molecular mechanisms through which Pum2 and TDP-43 affect the expression of layer IV/V molecular determinants, we examined many different steps of gene expression, including transcription/mRNA stability, isoform diversity generated by splicing and alternative 3′ end processing, as well as translation. However, we only detected significant effects on the distribution of *Sox5*, *Bcl11b*, and *Rorb* mRNAs in sucrose density gradients from pS (*Figure 9*), providing evidence that translation is affected. How strong is the case for translational regulation based on our sucrose density gradient polysome profiling assays? Two big advantages of this assay vs. tagged-ribosome alternatives (*Heiman et al., 2008*; *Sanz et al., 2009*) are that it is independent of mRNA levels and can reveal shifts of an mRNA between gradient fractions. The latter reflects translational regulation driven by changes in ribosome number/mRNA, rather than just ribosome access. For example, in the case of *Rorb*, there is a shift of almost half the mRNA from a fraction with approximately seven ribosomes per mRNA to the fraction with approximately one ribosome/mRNA. In our view, this is a fairly strong effect on ribosome density that would be predicted to lead to a significant reduction in protein output from this mRNA, in perfect agreement with and offering a reasonable explanation for the protein-level phenotypes in the pS. The effects on *Sox5* and *Bcl11b* mRNAs are arguably more subtle, but this might be expected in a bulk tissue assay. Importantly, although our sucrose gradient assays lack cellular resolution, we see no reason why this should lead to false-positive effects. One caveat is that the shifts we observe may not reflect altered ribosome association since we do not purify ribosomes directly or demonstrate that the complexes are disrupted by puromycin treatment of neocortices prior to cell lysis. However, we think the clear congruence between the effects on mRNAs in the gradients and at the protein level favors the simple interpretation of effects on ribosome density on the mRNAs. On balance, we think our positive results in the gradient polysome profiling assays indicate that translational regulation of these mRNAs by Pum2 and TDP-43 is occurring and could therefore contribute to layer IV/V cytoarchitecture in S1. Future experimental approaches with higher cellular resolution will help to determine whether important contributions from transcriptional or other post-transcriptional mechanisms might have escaped detection in the assays that we performed here.

It is important to understand that even though the *Pum2* cKO and TDP43 overexpression phenotypes are highly similar, both at the neurodevelopmental and post-transcriptional/translational levels,

our genetic strategy implies opposite modes of action for these RBPs. Pum2 loss-of-function phenotypes indicate that Pum2 *promotes* normal layer IV/V cytoarchitecture in S1, whereas phenocopy by TDP-43 gain of function suggests that TDP-43 can *oppose* this process. While we cannot yet say the relative contribution of translational control to the overall process, this competing regulation is reflected in our polysome gradient data in *Figure 9*, which imply that Pum2 normally represses translation of the mRNAs for layer V fate determinants, *Sox5* and *Bcl11b*, whereas TDP-43 activates them. Conversely, Pum2 activates translation of a molecular determinant that can drive layer IV fate: Rorβ, whereas TDP-43 appears to repress *Rorb* mRNA translation. Importantly, the predicted binding sites for each RBP in the 3′ UTRs of these mRNAs do not overlap for the most part, suggesting that simultaneous binding and competition on the same mRNA molecule would be possible. An interesting line of future experimentation would be to delineate the exact binding sites on the regulated mRNAs for both proteins and dissect the relative contribution they make to regulation in the context of newly born layer IV/V neurons in the pS.

Many other RBPs presumably bind to the mRNAs affected here and may also thereby contribute to post-transcriptional regulation as co-factors, competitors, or independent regulators. Bearing this in mind, it would also be interesting to focus on specific *cis*-elements in the 3′ UTRs of *Sox5*, *Bcl11b*, and *Rorb* mRNAs and their relative contributions to regulation. This would be conceptually similar to work pioneered in *Caenorhabditis elegans* to dissect the regulatory logic underlying terminal differentiation of specific neuronal classes (*Hobert, 2008*; *Hobert and Kratsios, 2019*), but at a post-transcriptional level. Similar approaches in other systems have provided major insights into the molecular regulatory logic underlying post-transcriptional regulation during oocyte development (*Piqué et al., 2008*). Given that all of these mRNAs show diversity in their 3′ UTRs which is likely to impact on stability and translation, it will also be important to examine the relative amounts of specific isoforms in developing layer IV/V neurons and incorporate this information into models of post-transcriptional regulation of layer IV/V neuronal specification in S1. Autoregulation and cross-regulation should also be examined, and interplay with transcriptional regulation will clearly be a key aspect to understand.

Our data also provide further support for the idea that RBPs can function in a 'dual' translational regulatory mode, acting either as activators or repressors depending on mRNA context. Most previous studies examining mRNA-specific translational regulation by Pum2 and TDP-43 have characterized them exclusively as repressors (*Cao et al., 2010*; *Coyne et al., 2014*; *Majumder et al., 2012*; *Vessey et al., 2010*; *Wickens et al., 2002*; *Zahr et al., 2018*). However, a recent study from our group revealed a translational enhancer function for both hTDP-43 and hTDP-43[A315T] in cultured neuronal cells (*Neelagandan et al., 2019*). Pumilio was reported to function as a translational repressor in the context where it was originally identified (*Lehmann and Nüsslein-Volhard, 1991*; *Murata and Wharton, 1995*), and this function is clearly conserved among Pumilio family (Puf) proteins (*Wickens et al., 2002*). Nevertheless, there is also precedent for translation activation of specific mRNAs by Puf proteins in both *Xenopus* oocytes (*Piqué et al., 2008*) and *C. elegans* (*Kaye et al., 2009*). Recent work with shRNA knockdowns in cultured cortical neurons also reported a translational enhancer function for Pum2, although this appeared to be more general (*Schieweck et al., 2021*). Other studies have focused on other post-transcriptional effects. For example, simultaneous knockdown of Pum1 and Pum2 in cultured non-neuronal cells affected stability of hundreds of mRNAs (*Bohn et al., 2018*), although potential effects on translation were not analyzed in this study. Our results with sucrose density gradient polysome profiling provide in vivo evidence for mRNA-specific translational activator roles for both Pum2 and TDP-43 in the context of mammalian brain development. Moreover, they suggest the possibility of dynamic switching between repressor and activator capabilities during development via mechanisms that remain to be defined.

A critical issue raised by our studies is the enigma of area-specific regulation by Pum2 and TDP-43, given that both are ubiquitously expressed RBPs. One possibility is that area-specific signaling mechanisms might converge on post-translational modifications of Pum2 and TDP-43. In addition, RBPs often work together in co-factor complexes (e.g., *Vessey et al., 2012*; *Zahr et al., 2018*) and an unidentified RBP co-factor for area-specific post-transcriptional regulation might be expressed in an area-specific manner. There is also evidence that thalamic innervation can affect the molecular composition of the ribosome itself and that this differentially impacts translation of specific mRNAs in a spatial and temporal manner (*Kraushar et al., 2015*). Thus, one can also imagine that area-specific effects on ribosome composition and function might also play a role in RBP regulatory capacity within

specific cortical areas. Clearly, future work will be needed to resolve the important issue of how spatial control arises through ubiquitously expressed proteins.

Our results raise the possibility that post-transcriptional regulation by Pum2 and TDP-43 might reflect a 'downstream module' for area-specific neuronal subtype specification. An unusual feature of the S1 layer IV/V 'motorization' phenotype, which we show in *Figures 1 and 3*, is its selective effect on this aspect of area identity. As shown in *Figure 5*, other molecular and cytoarchitectural aspects of S1 area identity, such as expression of specific molecular markers and formation of characteristic barrels, appear largely preserved in both *Pum2* cKO and TDP43$^{A315T}$ mutants. In contrast, other mutants, identified to date that lead to a motorized S1, appear to affect all of these aspects of area identity (*Alfano and Studer, 2013*; *Armentano et al., 2007*; *O'Leary and Nakagawa, 2002*; *O'Leary and Sahara, 2008*; *Tomassy et al., 2010*). We interpret the selectivity in *Pum2* cKO and TDP43$^{A315T}$ mutants as evidence that they might function as components of a downstream regulatory module for elaboration of specific aspects of area identity, rather than controlling identity per se. Future work will be necessary to resolve whether Pum2 and TDP-43 function directly downstream of previously described area identity determinants or comprise a parallel pathway. Regardless, our results raise the intriguing possibility that neocortical arealization involves at least two genetically separable components: initial 'area definition' and subsequent 'area elaboration.' This observation suggests a general genetic strategy for identifying downstream elaboration modules of area-specific architectural elements: identifying mutants that selectively affect specific elements of area identity while leaving others intact. Systematic screening for such 'area elaboration mutants' might be one fruitful strategy to elucidate the downstream molecular programs that elaborate area-specific subtype specification and connectivity. While transcriptional regulation will certainly play a crucial role here, our results also support casting a broader 'genetic net' to include potential contributions of post-transcriptional regulation.

The findings we report here also shed light on a fundamental issue in molecular control of cortical development: Which regulatory mechanisms are established in neuronal precursors, and which take place in post-mitotic neurons? A previous study with *Pum2*-targeting shRNAs delivered by IUE observed translational de-repression of a lower-layer marker, TLE4, in neuronal progenitors (*Zahr et al., 2018*). However, several lines of evidence imply that the regulation we observe here with *Pum2* cKO mice occurs in post-mitotic neurons. First, regulation is observed at P0, when cortical neurogenesis is complete (*Figure 9d*) and binding to these mRNAs is also strong at E18.5 (*Figure 10c*). Second, we did not observe any effect on *Sox5* or *Bcl11b* mRNA translation at E13.5, E14.5, or E18.5 (*Figure 9—figure supplement 3b*), the peak birth time for layer V and IV neurons and even prenatally, but at P0 when neurogenesis is completed, and neurons are already post-mitotic (*Figure 9d*). Third, our examination of nascent neurons of *Pum2* cKO mice at E13.5 did not show increased protein expression of Sox5, Bcl11b, or Tbr1 (*Figure 9—figure supplement 3c*). Fourth, we saw apparent neuronal fate changes in pS1 when we performed IUE with either Cre or TDP43 in the pNeuroD context, which is believed to be exclusively expressed in post-mitotic neurons (*Guerrier et al., 2009*). It will be important to verify that conclusions based on Pum2 loss-of-function phenotypes can be rescued by restoring Pum2 protein levels. Nevertheless, our results support the notion that regulation of Sox5, Bcl11b, and Rorβ protein levels can occur post-mitotically.

One developmental mechanism that seems to be implied by our data is that some newly born S1 neurons that are normally fated to become layer IV neurons might conceivably be re-specified if the levels or activity of Pum2 or TDP-43 would be sufficiently reduced or increased, respectively. Assuming this model is correct, two issues are raised. (1) What might be the underlying molecular basis for this hypothetical and apparently highly selective re-specification capacity? (2) What might be its biological value as a regulatory mechanism? With respect to the first point, we can speculate that these S1 neurons in layer IV, and no other layers, might be inherently predisposed to re-specification by virtue of having related genetic programs to the recently derived layer V neurons. Shared molecular expression patterns in these populations have been described at both the transcriptomic and proteomic levels (*Ayoub et al., 2011*; *Poulopoulos et al., 2019*; *Sadegh et al., 2021*). Interestingly, shared molecular expression signatures extend to the noncoding genome and include microRNAs (miRNAs) miR-128, miR-9, and let-7, which are functionally distinct, yet commonly involved in specifying neurons of layers VI and V and layers IV, III, and II, respectively: they can transiently alter their relative levels of expression to change from stem-cell competence towards a neurogenic stage-specific pattern. Furthermore, these shared miRNAs are able to shift neuron production between earlier-born and later-born fates

to generate laminar identity (*Shu et al., 2019*; *Zolboot et al., 2021*). Future work could validate the predictions of this intriguing model and explore potential interplay between transcriptional and post/transcriptional regulation in this context.

Regarding our data, the most obvious molecular determinants to be relevant here would be *Sox5* and *Bcl11b*, which are expressed in at least a subset of newly born layer IV neurons at a low level. In this model, regulation by Pum2 and TDP-43 can tune production of these transcription factors, with Pum2 normally putting a brake on their synthesis, and TDP-43 having the capacity to amplify the output from lower mRNA levels in these newly born neurons. However, we also see reciprocal effects on *Rorb* mRNA in the polysome gradients. This could be a consequence or epiphenomenon, but this observation does suggest that it might also contribute to effects on expression. Under normal conditions, Pum2 might activate translation of this mRNA, amplifying the switch to a layer IV fate, whereas increased TDP-43 levels appear able to downregulate *Rorb* mRNA's translation. In the most extreme version of this model, translational regulation is the key element, with Pum2 and TDP-43 governing a 'translational switch' controlling neuronal fate. A more likely scenario is that other yet-to-be-defined post-transcriptional mechanisms (e.g., regulated protein turnover) may play equally or even more important roles. Determining whether this model is correct and defining the relative contribution of translational control vs. other regulatory mechanisms will require new approaches with much higher spatial and temporal resolution to correlate the fate of these specific neuronal populations with specific molecular changes within them, including assays specifically measuring translational effects. From this perspective, we find it extremely encouraging that regulation seems strikingly similar in our IUE assays in vivo and in our pS-enriched primary neuron transfection assays in vitro (compare results in *Figures 6 and 7*). To us, this suggests strong potential to recapitulate the core regulatory effects on expression of proteins affecting cell fate and gene expression in an ex vivo system (e.g., slice cultures) that would be amenable to live imaging and enable more rapid experimental manipulations with a wider variety of readouts.

Considering the second point regarding biological value, our work raises the intriguing – albeit currently speculative – possibility that altering the relative activity of Pum2 and TDP-43 within the cytoplasm of developing pS neurons might potentially provide a mechanism to dynamically tune the fate of layer IV/V neurons in response to environmental inputs. According to this view, neuronal identity in S1 is not fully hardwired, but somewhat plastic. We can further speculate that optimal setting of network-level parameters in the developing brain might require fine-tuning of neuronal identity between SCPNs in response to evolving input from the hypothalamus and presumably intracortical signaling as well. In other words, neuronal fate for these populations might not yet be locked in, but rather remain plastic until particular later critical periods in cortical development have been completed. In this regard, we find it interesting that the effects we observe are manifested post-mitotically and that altered translational regulation is observed at least as late as P0 in the *Pum2* cKO, a time when neurogenesis per se is complete, and activity-dependent, wiring-driven effects will play an increasingly important role. One can hypothesize that there is still capacity to tune layer-neuron cytoarchitecture in S1 at this stage in response to network activity and that competing regulation by Pum2 and TDP-43 might play a role in re-specification. Experiments to directly examine this possibility can be envisaged. Specifically, we think it would be interesting to integrate electrophysiological approaches with detailed cellular-level analyses of post-transcriptional regulation by Pum2 and TDP-43 and its interplay with transcriptional regulation in this specific developmental context.

What might be the broader impact on brain function and implications for human health of altered S1 cytoarchitecture resulting from loss of Pum2 or increased levels of TDP-43? Reduced Pum2 function has been implicated in epilepsy in both rodents and humans (*Follwaczny et al., 2017*; *Siemen et al., 2011*; *Wu et al., 2015*), and altered cortical wiring in S1 during development might conceivably contribute to seizures due to perturbations of excitation/inhibition balance that propagate through the network (*Guerrini and Dobyns, 2014*). However, the contribution of altered sensory system function to epilepsy remains unclear and seizures reported by others in *Pum2* KO mice might very well have a completely different origin. Other behavioral phenotypes associated with loss of Pum protein function in the brain have also been described (*Siemen et al., 2011*; *Zhang et al., 2017*). Although technically challenging, it would clearly be of great interest to examine whether altered wiring of S1 contributes to these behavioral effects and, if so, the underlying physiological basis.

In contrast to Pum2, TDP-43 deregulation is mainly implicated in neurodegenerative diseases, particularly ALS and FTD, both of which strike layer V neurons in multiple cortical areas late in life (*Geser et al., 2010*; *Taylor et al., 2016*). We found that modest overexpression of a patient-derived mutant allele of TDP-43 during cortical development significantly increases the number of layer V neurons and dramatically alters connectivity of S1, significantly enhancing the number of subcerebral projections (*Figures 1 and 3*). Whether these alterations contribute causally to disease remains to be determined; however, they are not sufficient for disease since *Pum2* cKO mice show a similar phenotype, but do not develop ALS-like symptoms. Effects on laminar identity in a wild-type TDP43 transgenic line that does not develop ALS symptoms (*Figure 1—figure supplement 9*) also seem to favor the idea that altered specification in S1 is unrelated to ALS/FTD. However, the effects in this asymptomatic line were weaker than those observed in the patient-derived mutant line that develops symptoms (compare *Figure 1—figure supplement 9* to *Figure 1*). The weaker effect on S1 specification in this line might conceivably be below a threshold needed to contribute to disease, and this might also explain the absence of ALS-like phenotypes in mice lacking Pum2. Future work should therefore examine whether altered connectivity is a general phenomenon of loss of Pum2 or gain of TDP-43 function and whether there might be a correlation between the level of TDP-43 expression and altered wiring. Assuming this proves true, it would then be of great interest to examine how developmental alterations in area-specific connectivity seen in these mice affect signaling in cortical networks and whether this ultimately contributes to degeneration of layer V cortical neurons and their subcerebral targets in spinal cord.

## Materials and methods

### Animal welfare and approvals

All animal care and experimental procedures were performed according to the institutional guidelines of the UKE or University of Geneva and relevant national law. In Hamburg, guidelines were those of the UKE Animal Research Facility (FTH) and conformed to the requirements of the German Animal Welfare Act. Ethical approvals were obtained from the State Authority of Hamburg, Germany (G10/107_Pumilio, G14/003_Zucht Neuro, N086/2020_Pum2/TDP43 IUEs, ORG_520 and ORG_765).

### Generation and use of *Pum2* cKO mice

ES cell lines targeting cassette for exons 6 and 7 of Pum2 were obtained from KOMP (link: CSD45770; parental ES cell line: JM8A1.N3) and expanded for injection according to their protocol. Cells were injected into morulae derived from BALB/C mice using the PiezoXpert (Eppendorf).

Germline transmission was verified by the long PCR procedure recommended by KOMP, as well as by Southern blotting. Founder lines were mated to a line expressing Flp recombinase in the germline (*Rodríguez et al., 2000*) to excise the targeting cassette and generate the 'floxed' conditional allele. The constitutive *Pum2* KO line was generated by mating this line with mice expressing Cre recombinase in the germline (*Schwenk et al., 1995*). As the original KOMP ES cell lines were on a C57Bl/6N background, the floxed *Pum2* and *Pum2* KO lines were backcrossed more than 10 times to C57Bl/6J prior to use and were also maintained by routine backcrossing to C57Bl/6J.

**Table 5.** Genotyping primers.

| Allele | Forward primer (5'–3') | Reverse primer (5'–3') |
| --- | --- | --- |
| Pum2 KO | GCTGCTACTCCCTTTCTTGC | GAGCACATGTGGAGGTCAGA |
| Pum2 WT and floxed | GCTGCTACTCCCTTTCTTGC | CCAAGGCGCTCAACTACTTC |
| Cre | TAACATTCTCCCACCGCTAGTACG | AAACGTTGATGCCGGTGAACGTGC |
| Actin | CAATAGTGATGACCTGGCCGT | AGAGGGAAATCGTGCGTGAC |
| TDP43$^{A315T}$ | GGATGAGCTGCGGGAGTTCT | TGCCCATCATACCCCAACTG |
| TDP43 | GGATGAGCTGCGGGAGTTCT | TGCCCATCATACCCCAACTG |
| Control for TDP43 | CAAATGTTGCTTGTCTGGTG | GTCAGTCGAGTGCACAGTTT |

## Mouse housing and genetics

Mice were housed in a barrier facility and maintained under standard housing conditions with a 12 hr light/dark cycle and ad libitum access to water and chow.

*Pum2^{fl/fl}* mice were crossed to *Emx1^{Cre}* to inactivate Pum2 in forebrain principal neurons and glia (*Iwasato et al., 2000*). *Pum2^{fl/fl}* littermates were taken as controls. For experiments characterizing conditional heterozygotes, *Emx1^{Cre}; Pum2^{+/fl}* mice were mated to *Pum2^{+/fl}*.

Mice expressing either *hTARDBP* (*Arnold et al., 2013*) or *hTARDBP^{A315T}* (*Wegorzewska et al., 2009*) were obtained from the Jackson Laboratory (Bar Harbor, Maine, USA; stocks 017907 and 010700, respectively) on a congenic C57Bl/6J background. Non-transgenic littermates were taken as controls. At least three independent litters were used for each analysis. All mouse lines used for experiments were congenic on C57Bl/6J and maintained by backcrossing to this wild-type background. Mouse lines were genotyped using primers in *Table 5*. Early morning of the day of the vaginal plug was considered as embryonic day 0.5 (E0.5).

## Postmortem tissue collection

Embryonic and postnatal brain samples were fixed either for 2 hr (for immunohistochemistry [IHC]) or overnight (for fluorescent in situ hybridization) at 4°C in PFA 4%. Samples were then embedded in optimal cutting temperature (OCT) medium (JUNG) after being equilibrated progressively in 10, 20, and 30% sucrose, and cut on a Leica cryostat. No samples were excluded in this work. For each experiment, a minimum of three animals from different litters were used.

## Nissl staining

20 µm coronal and sagittal sections were rinsed for 2 min in distilled water and incubated for 30 min in cresyl violet and washed twice in distilled water. Additional sequential incubation for 2 min in 20, 50, and 75% ethanol, 96% ethanol/acetic acid, and 100% ethanol followed. Sections were then incubated twice for 5 min in 100% ethanol first and then with xylol. After drying, sections were finally mounted with Eukitt and stored at room temperature (RT).

## Immunofluorescent staining and imaging

Immunofluorescent imaging was performed on cryosections. Briefly, slides were boiled in an unmasking buffer (sodium citrate 0.1 M, pH 6). After three PBS washes, cryosections were blocked with 10% goat serum and 0.3% Triton X-100 for 1 hr at RT. Primary antibody incubations were carried out overnight at 4°C. Secondary antibodies were added for 2 hr at RT. The following primary antibodies were used: rat anti-Ctip2/Bcl11b (dil 1:300, Abcam ab18465), rabbit anti-Sox5 (1:300, Abcam ab94396), mouse anti-Rorβ (1:200, Perseus Proteomics PP-N7927-00), rabbit anti-Cux1 (1:100, Millipore ABE217), rabbit anti-Tbr1 (1/300, Abcam ab31940), rat anti-Lmo4 (1:500, gift from J. Valsvader), guinea pig anti-Bhlhb5 (1:500, gift from B. Novitch), rabbit anti 5-HT (1/10000, Immunostar 20080), rabbit anti-Pum2 (1:100, Bethyl A300-202A), rabbit anti-Tdp43 (1:300, Abcam AB41881), guinea pig anti-NeuN (1: 300, Synaptic Systems 266004), and mouse anti-GFAP (1:300 Synaptic Systems 173211). The following Alexa-conjugated secondary antibodies from Life Technologies were used: goat anti-rabbit FC (488, 594), goat anti-rat FC (488, 594), goat anti-mouse FC (488, 594, 633), and goat anti-guinea pig FC (488) (dil 1:300). Slides were incubated for 10 min in PBS with DAPI (1:1000, Thermo Fisher) and mounted with ROTI Mount FluorCare (Roth).

## Retrograde labeling with cholera toxin B

For retrograde labeling, anesthetized P0 pups were injected with Alexa Fluor 555-conjugated CTB (1 mg/ml; Invitrogen, volume injected: 300 µl) under ultrasound guidance using a Vevo 770 ultrasound backscatter microscopy system (Visual Sonics). Subcerebral injections were performed at the midbrain–hindbrain junction using a nanojector (Nanoject II Auto-Nanoliter Injector, Drummond Scientific Company 3-000-204) to label all SCPNs, including corticopontine projection neurons and corticospinal motor neurons. Injected pups were perfused at P7, and brains were collected and 40-µm-thick sections were cut at the cryostat and either directly incubated for 10 min with DAPI and mounted with ROTI Mount FluorCare (Roth) or treated for immunostaining. In the last case, slides were incubated with the blocking solution (10% goat serum and 0.3% Triton X-100) for 1 hr at RT and were then incubated with primary antibodies overnight at 4°C. The primary antibodies used were

rat anti-Ctip2 (Bcl11b) (dil 1:300, Abcam ab18465), and rabbit anti-Sox5 (1:200, Abcam ab94396). Subsequent to three washes with PBS, the slides were incubated with corresponding Alexa Fluor 488 secondary antibody (1:300; Life Technologies) for 2 hr at RT. Sections were washed with PBS three times and incubated for 10 min in PBS with DAPI (1:1000) (Thermo Fisher) and mounted with ROTI Mount FluorCare (Roth).

## smFISH and combined IHC/smFISH

Collected brains were fixed for 24 hr in PFA 4% at 4°C. Samples were then embedded in OCT medium (JUNG) after being equilibrated progressively in 10, 20, and 30% sucrose, and cut on a Leica cryostat (thickness: 16 µm). Cryosections were left 1 hr to dry at –20°C and then stored at –80°C. RNAscope in situ hybridization assays were performed according to the manufacturer's instructions (Advanced Cell Diagnostics [ACD]). Briefly, cryosections were gradually dehydrated in 50%, 70%, and twice in 100% ethanol for 5 min each at RT. Slides were left to dry for 30 min at RT. In between all pretreatment steps, tissue sections were briefly washed into a Tissue-Tek Slide Rack submerged in a Tissue-Tek Staining dish with distilled water. Incubations were performed on the HybEz II hybridization system (ACD). The pretreat solution 1 (hydrogen peroxide reagent) was applied for 10 min at RT, and then the tissue sections were boiled in pretreat solution 2 (target retrieval reagent) for 5 min. Slides were treated with pretreat solution 3 (protease III reagent) for 30 min at 40°C for FISH while with pretreat solution 4 (protease IV reagent) for 20 min for FISH combined with IHC. Custom mouse *Sox5* (413291), *Bcl11b* (413271-C2), *Fezf2* (313301-C3), *Cux1* (442931), and *Rorb* (444271-C3) RNAscope probes were designed and purchased from ACD. In addition, the negative (Cat# 310043, ACD) and positive (Cat# 313911, ACD) control probes were applied and allowed to hybridize for 2 hr at 40°C. The amplification steps were performed according to the manufacturer's instructions. In between every amplification step, sections were washed with 1× wash buffer. Detection was performed using TSA Plus fluorophore (fluorescein, cyanine 3, or cyanine 5) (1:1500-1:3000) from PerkinElmer for 30 min at 40°. Slides were rinsed twice in 1× wash buffer, incubated for 10 min in distilled water with DAPI (ACD), and then mounted with ROTI Mount FluorCare (Roth). For combined IHC/smFISH, following the amplification step, sections were processed for IHC. Briefly, slides were incubated with the blocking solution (10% goat serum and 0.3% Triton X-100) for 1 hr at RT and were then incubated with primary antibodies ON at 4°C. The primary antibodies used were rabbit anti-Pum2 (1:100, Millipore 03-241) and rabbit anti-Tdp43 (1:300, Abcam AB41881). Subsequent to three washes with PBS, the slides were incubated with corresponding Alexa Fluor 555 secondary antibody (1:300; Life Technologies) for 2 hr at RT. Brain sections were rinsed with PBS three times and incubated for 10 min in PBS with DAPI (ACD) and mounted with ROTI Mount FluorCare (Roth).

## Polysome profiling and total RNA preparation

Animals were collected at either E13.5, E14.5, E18.5, or P0, and cortices were dissected in a dissection buffer containing 2.5 mM HEPES-KOH (pH 7.4), 35 mM glucose, 4 mM NaHCO₃, and 100 µg/ml cycloheximide and flash frozen in liquid nitrogen and stored at –80°C. For P0 cortices, a somatosensory area-enriched region has been dissected by using a millimeter paper and taking out with a blade 1 mm from the rostral and 1 mm from the caudal regions of the cortex (*Figure 9b*). After genotyping, three replicates for controls and either *Pum2* or *TDP-43* mutants were processed. Each replicate consists of one, two, three, or four pooled cortices for P0, E18.5, E14.5, and E13.5, respectively.

Cortices were homogenized using a glass dounce in 400 µl of lysis buffer containing 20 mM HEPES KOH (pH 7.4), 150 mM KCl, 5 mM MgCl₂, 0.5 mM DTT, 100 µg/ml cycloheximide (Sigma-Aldrich), 1X cOmplete mini EDTA-free Protease Cocktail (Roche), 40 units/ml RNaseIn (Promega), and 20 units/ml SUPERaseIn (Thermo Fisher Scientific). Cortical lysates were centrifuged for 10 min at 2000 × *g* at 4°C, and supernatants were supplemented with 1% NP-40 and 1% Triton X-100 and incubated on ice for 5 min. After centrifugation for 10 min at 20,000 × *g* at 4°C, the debris-free supernatants were collected. 20 µl of the input lysates were saved as a reference for qRT-PCR. The OD₂₆₀ of each lysate was measured on a NanoDrop spectrophotometer, and volumes were adjusted to ensure equal OD unit loading. 400 µl of each sample was loaded onto 14 × 95 mm Polyclear centrifuge tubes (Seton Scientific) containing 17.5–50% sucrose gradients (in Gradient Buffer containing 20 mM Tris-HCl, pH 7.4, 5 mM MgCl₂, 150 mM NaCl, 1 mM DTT, 100 µg/ml cycloheximide); the sucrose gradients were generated using the Gradient Master 108 programmable gradient pourer (BioComp). The sucrose

gradients containing the cortical lysates were then centrifuged for 2 hr and 15 min at 35,000 rpm in a SW40Ti rotor in a Beckman L7 ultracentrifuge (Beckman Coulter). After centrifugation, gradients were fractionated and measured for RNA content using a Piston Gradient Fractionator (BioComp) attached to a UV monitor (Bio-Rad).

For puromycin treatment, both control lysates and the puromycin-treated lysates (2 mM in lysate) were incubated on ice for 15 min followed by another 15 min at 37°C prior to loading on the 17.5–50% sucrose gradient.

For polysome to monosome (P/M) ratio analysis, areas under the curves representing the monosome and polysome peaks in gradient profiles were quantified using ImageJ (*Schneider et al., 2012*), and the P/M ratio was calculated by dividing the area under the curve of polysome peaks by area under the curve of the monosome peak.

## qPCR and PCR with polysome gradient fractions and total RNA

Prior to RNA purification, individual gradient fractions were aligned with corresponding profiles and pooled according to the scheme presented in *Figure 9*. Pool 1 contains the non-ribosome-bound portion of the gradient, pool 2 contains 80S monosomes, pool 3 contains disomes and trisomes, pool 4 contains mRNAs with ~4–6 ribosomes bound, and fractions from the deeper fractions corresponding to roughly seven or more ribosomes per mRNA were divided into two equal pools, 5 and 6, respectively. 1 ng of an in vitro-transcribed RLuc spike-in mRNA was added to each of the six pools as a recovery control and for normalization of the samples.

Total RNA was prepared from the six gradient fraction pools and the corresponding input lysate samples using Trizol in a ratio of 3:1 and the PureLink RNA mini kit (Thermo Fisher Scientific) according to the manufacturer's specifications. The purified RNA was concentrated by ammonium acetate precipitation using GlycoBlue carrier (Thermo Fisher Scientific). Pellets were washed with 70% ethanol, air-dried, and resuspended in nuclease-free water. RNA concentrations were determined using a NanoDrop spectrophotometer, and 250 ng of RNA was reverse transcribed with random primers using the SuperScript II cDNA Synthesis Kit (Thermo Fisher Scientific) according to the manufacturer's instructions. qPCR was performed using FastStart Universal SYBR Green Master (ROX) (Roche). All reagents and kits were used according to the manufacturer's instructions. The $\Delta C_t$ method was used for relative quantification of qPCR data. For polysome fractions, levels of spike-in RLuc RNA were measured first and their relative distribution across the fraction pools was calculated and normalized to the non-ribosome-bound pool. The same procedure was used for all other RNAs analyzed, and their distribution was additionally normalized to the one obtained for RLuc RNA and expressed as a percentage of cumulative signal. For input lysate RNA samples, values for specific mRNAs were normalized to GAPDH and represented as fold change of mutants to controls. Primers used for all qPCR analyses are described in *Table 1*; *Table 4*; *Table 2*. Similar cDNA from input lysate RNA of P0 somatosensory area-enriched cortical lysates was also used for RT-PCR to detect Sox5 splicing isoforms (*Table 3*) as in *Edwards et al., 2014*.

## UV-CLIP with qRT-PCR as readout

Embryos were collected at E18.5 and cortices from 10 embryos were dissected and harvested in 5 ml ice-cold 1× PBS, in which they were resuspended by pipetting using P1000 then P10 tips. Dissociated cortices that were enough for 10 immunoprecipitations were then divided equally into two 10 cm dishes on ice. One half was UV irradiated (4 * 100 mJ/cm²) using a Stratalinker. The other half was used as control non-UV-treated sample. Cross-linked cells and non-cross-linked cells were divided into 500 µl samples and were centrifuged at top speed for 10 s at 4°C. Pellets were lysed in 1 ml of lysis buffer (50 mM Tris–HCl, pH 7.4, 100 mM NaCl, 1% NP-40, 0.1% SDS, 0.5% Na-deoxycholate, 1× cOmplete Protease Inhibitor Cocktail [Roche]). A fraction of the lysate corresponding to 5% of the input material (50 µl) was retained to use as a reference for calculating the fraction of input material in the IP pellet. The remaining lysate was added to Protein G Dynabeads pre-bound with either 4 µg rabbit polyclonal TDP-43 antibody (Abcam ab41881) or 5 µg of Pum2 antibody (Millipore #03-241) or 5 µg rabbit IgG (Millipore #03-241) as a control and rotated at 4°C overnight. Beads were subsequently washed twice for 2 min in high salt buffer (50 mM Tris–HCl, pH 7.4, 1 M NaCl, 1 mM EDTA, pH 8.0, 1% NP-40, 0.1% SDS, 0.5% Na-deoxycholate), followed by washing twice for 2 min in wash buffer (20 mM Tris–HCl, pH 7.4, 10 mM NaCl, 0.2% Tween-20) and a final washing step for 2 min in

NT2 buffer (50 mM Tris–HCl, pH 7.4, 150 mM NaCl, 1 mM $MgCl_2$, 0.05% NP-40). RNA was eluted by incubation with 30 µg Proteinase K (Carl Roth) in NT2 buffer for 30 min at 55°C. RNA extraction was carried out from the eluate and input sample using Trizol reagent in a ratio of 3:1 followed by the addition of chloroform and subsequent purification by PureLink kit (Ambion). All RNA obtained from each sample (input or IP) were used to generate cDNA libraries using random hexamers (Thermo Scientific) and the RevertAid RT reverse transcription kit (Thermo Scientific), following the manufacturer's protocol.

We first verified that we had lower Cts for a given mRNA in the specific IPs relative to IgG control in the UV-cross-linked samples, implying specific signal over background. To calculate target mRNA enrichments, we first calculated the $\Delta C_t$ for TDP-43 or Pum2 IP versus input and converted this to a linear 'fold change' value. These were then corrected for the reduced amount of input analyzed (i.e., divided by 20), and then multiplied by 100 to obtain '% of input mRNA in IP.' Statistical comparisons were performed relative to 18S rRNA as neither protein has been shown to functionally regulate this RNA.

## Immunoblotting

Cerebral cortices from P0 controls, TDP43, and TDP43[A315T] pups were dissected after genotyping, and nuclear and cytoplasmic proteins were separated using the NE-PER kit from Pierce (Thermo Scientific) according to the manufacturer's instructions. Samples were loaded on a 10% SDS polyacrylamide gel and subjected to standard SDS-PAGE electrophoresis on Mini-Protean tetra cell (Bio-Rad).

Immunoblotting to nitrocellulose or PVDF was performed using an iBlot rapid transfer device (Life Technologies) according to the manufacturer's guidelines. Blots were blocked in 5% milk/TBS-T solution and probed with antibodies diluted as indicated. Signals were visualized using fluorescent secondary antibodies and imaged on a LI-COR Odyssey CLx (LI-COR). Antibodies used in this study were mouse anti-human monoclonal TDP-43 (Novus Biologicals, H000023435-M01) (1:500), rabbit polyclonal TDP-43 (G400) (CST-3448) (1:1000), rabbit anti-Emx1 (Abcam, ab136102) (1:500). goat anti-rabbit IRDye 680LT (1:15000, LI-COR), goat anti-mouse IRDye 680LT (1:15000, LI-COR), goat anti-rabbit IRDye 800CW (1:15000, LI-COR), and goat anti-mouse IRDye 800CW (1:15000, LI-COR).

## Primary neuron transfections

Primary neuronal cultures were prepared from somatosensory area-enriched cortices (*Figure 9b*) of E18.5 C57BL/6J mice. After Hanks' Balanced Salt Solution (HBSS) washes, neurons were incubated for 10 min at 37°C with Papain and DNase I (Worthington). Tissue was then triturated with a Pasteur pipette, and supernatant was separated from cell debris into a new tube. Cells were resuspended in Dulbecco's Modified Eagle Medium (DMEM)/fetal calf serum (FCS) after centrifugation for 10 min at $1000 \times g$ and were immediately transfected. Transfections were performed using the Amaxa nucleooporation system using Mouse Neuron Nucleofector Kit (VPG-1001) following the manufacturer's manual. $5 * 10^6$ cells were used for each transfection. Briefly, neurons were resuspended in 100 µl of nucleofector solution containing 3 µg of DNA and transferred into the special electroporation cuvette. The transfection program used was O-005. Cells were collected after electroporation in DMEM/FCS and left for 1 hr at 37°C in the incubator. $0.5 * 10^6$ cells were grown on glass coverslips (12 mm diameter, Carl Roth) coated with poly-L-lysine in 12-well plates (Sarstedt) in Primary Neuro Basal Medium (Lonza) supplemented with NSF-1, penicillin/streptomycin antibiotics to 1% (v/v) and L-glutamine to 0.5 µM. Neurons were cultured for 2 days at 37°C in a 5% $CO_2$ environment prior to immunofluorescent staining. Neurons were fixed for 2 min with 4% PFA and 3 min with ice-cold methanol and then washed three times in PBS for 10 min each. Neurons were incubated with the blocking solution (10% goat serum and 0.3% Triton X-100) for 1 hr at RT, and were then incubated with primary antibodies overnight at 4°C. The primary antibodies used were rabbit anti-Flag (dil 1:200, Sigma-Aldrich F7425), mouse anti-V5 (dil 1:300, Invitrogen P/N 46-1157), chicken anti-GFP (dil 1:400, Abcam 13970) rat anti-Ctip2 (Bcl11b) (dil 1:300, Abcam ab18465), rabbit anti-Sox5 (1:200, Abcam ab94396), and mouse anti-Rorβ (1:100, Perseus Proteomics PP-N7927-00). After three washes with PBS, the slides were incubated with corresponding Alexa Fluor (488, 594, 633) secondary antibodies (1:300; Life Technologies) for 2 hr at RT. Neurons were rinsed with PBS three times and incubated for 10 min in PBS with DAPI (1:1000) (Thermo Fisher) and mounted with ROTI Mount FluorCare (Roth).

## Plasmids used for transfection of cortical neurons

The human TDP-43 plasmids (WT TDP43 and A315T mutant) and the control plasmid pEGFP-C1 have been previously described (*Neelagandan et al., 2019*). Briefly, the human TDP43 plasmids were generated in a pCMV Sport6 vector backbone with an N-terminal FLAG tag and C-terminal V5 tag. pKM29 contains the WT TDP43 coding sequence (CDS) in this context and pKM36 has the A315T mutant. The full-length ORF of human TDP43 was amplified from human TDP43 (TDP43) clone ID30389805 (Open Biosystems) without a stop codon (to allow the addition of 3′ tags to the protein product) and cloned using SalI and NotI into pCMV Sport 6.1. A FLAG-tag-encoding sequence for the 5′ end and a V5-tag-encoding sequence for the 3′ end were made by oligo annealing and cloned using KpnI/SalI (FLAG) and XbaI/HindIII (V5) into the human TDP-43-containing plasmid. The A315T mutation was introduced into the human TDP-43-containing plasmid using the QuikChange Site-Directed Mutagenesis Kit (Agilent Technologies, Cat# 200519).

## Plasmids used for in utero electroporation

The pNeuroD-IRES-GFP (*Guerrier et al., 2009*) plasmid was obtained from Addgene (plasmid number 61403). The pNeuroD-Cre-IRES-GFP (*Vitali et al., 2018*) was obtained from Dr. Denis Jabaudon Laboratory (Geneva, Switzerland).

For human TDP-43 and TDP-43$^{A315T}$ mutant plasmids, the full-length ORF were generated as described above for primary neurons transfection. Briefly TDP43 V5 and TDP43$^{A315T}$ V5 were excised from Flag-pCMV Sport 6.1- TDP43 -V5 and Flag-pCMV Sport 6.1- TDP43$^{A315T}$-V5 using SalI/HindIII and cloned into the p-NeuroD-IRES-GFP plasmid obtained from Addgene using XhoI/PstI.

## In utero electroporation

The Institutional Animal Care and Use Committee of the City of Hamburg, Germany, approved all experiments (approval n0. 86/2020 acc. to the Animal Care Act, §8 from May 18, 2006). Time-pregnant C57BL/6J or floxed Pum2 mice were given a preoperative dose of buprenorphine (0.01 mg/kg body weight) by subcutaneous injections at least 30 min before surgery. Animals were then anesthetized using 2.5% isoflurane/O$_2$ inhalation. Oxygen was delivered with a flow rate of 0.65 l/min and together with isoflurane were applied via a vaporizer (Föhr Medical Instruments, Seeheim-Oberbeerbach, Germany). The uterine horns were exposed, and respective plasmids mixed with Fast Green (Sigma) were microinjected into the lateral ventricles of the embryos. Five current pulses (50 ms pulse, 950 ms interval; 32 mV or 35 mV, respectively, for E12/13 or E14 embryos) were delivered across the heads of the embryos. Post surgery, 2–3 drops of meloxicam (0.5 mg/kg body weight) were given orally through soft food for 96 hr. Brains were collected at P0, and 40-µm-thick sections were cut at the cryostat and treated for immunostaining. Sections were incubated with the blocking solution (10% goat serum and 0.3% Triton X-100) for 1 hr at RT, and were then incubated with primary antibodies overnight at 4°C. The primary antibodies used were chicken anti-GFP (dil 1:800, Abcam 13970), rat anti-Ctip2 (Bcl11b) (dil 1:300, Abcam ab18465), rabbit anti-Sox5 (1:200, Abcam ab94396), mouse anti-Rorβ (1:200, Perseus Proteomics PP-N7927-00), mouse anti-human monoclonal TDP-43 (1:100, Novus Biologicals, H000023435-M01), and rabbit anti-Pum2 (1:100, Bethyl A300-202A). Subsequent to three washes with PBS, the slides were incubated with donkey anti-chicken Alexa Fluor 488 secondary antibodies (1:300, Jackson by Dianova #703-545-155) and corresponding secondary antibody goat anti-rat, rabbit, or mouse Alexa Fluor 633 (1:300; Life Technologies) for 2 hr at RT. Sections were washed with PBS three times and incubated for 10 min in PBS with DAPI (1:1000) (Thermo Fisher) and mounted with ROTI Mount FluorCare (Roth).

## Imaging, counting, and statistical analysis

Images were acquired using an Olympus FluoView 1000 microscope, and similar acquisition settings for laser power, offset, and detector gain across conditions were used. Bright-field brain and low-magnification Nissl staining images were acquired using a Zeiss Stemi 2000-C binocular. Higher-magnification images of Nissl staining were acquired using a Zeiss Axiophot microscope.

For counting, images of P0 neocortices from coronal sections of the pS and F/M regions were divided into six equal bins. At P7, the radial surface of analyzed neocortices was divided into eight equal bins. Counting of single-labeled cells was normalized to the total number of DAPI cells in each bin. For DAPI cells and cholera toxin-labeled neurons, the counting was performed on cortical images

with a constant width of 600 µm. In general, cells were counted using ImageJ Fiji. The trainable Weka Segmentation plugin was used to distinguish signal from background, and the resulted images were counted for each bin using Analyze Particles option for ImageJ Fiji. For Bcl11b, only high-expressing neurons mainly located in layer V were counted by setting low Bcl11b expression as background. For Sox5 and Bcl11b colocalization analysis, single staining for Sox5 and Bcl11b was submitted to trainable Weka Segmentation using ImageJ Fiji, and the generated images were analyzed for colocalization using JaCoP plugin in individual bins. For IUE, GFP and marker-specific channels were submitted to trainable Weka Segmentation using ImageJ Fiji, and the generated images were analyzed for colocalization of GFP-labeled electroporated neurons with Sox5, Bcl11b, or Rorβ using JaCoP plugin. FISH, transfected primary neurons, and CTB-labeled images were manually counted using Photoshop. Cortical thickness, hemisphere length, and width were measured using the measure option of ImageJ Fiji after setting scale using scale bar. Hemisphere area was measured by selecting hemisphere and the measure option of ImageJ Fiji. Nuclei size was analyzed using the average nuclei size for DAPI signal ×60 magnification images after particle analysis for DAPI channel.

Data were statistically analyzed in GraphPad Prism or Microsoft Office Excel and graphically represented using GraphPad Prism. Error bars represent the standard error of the mean (SEM). In general, a two-tailed Student's $t$-test was used for the analysis of statistical significance (*p≤0.05, **p≤0.01, ***p≤0.001) between different groups. One-tailed $t$-tests were used for the analysis of statistical significance of polysome profiling experiments because we have a specific prediction about the direction of the difference. The Mann–Whitney $U$ test was used for analysis of statistical significance of CLIP-qRT-PCR experiments.

## Acknowledgements

We especially thank Eva Kronberg for mouse colony management. We also thank Denis Jabaudon (Geneva, Switzerland) for retrograde labeling experiments, many helpful scientific discussions, and the pNeuroD-Cre-IRES-GFP plasmid. We also thank Christoph Janiesch for helpful advice on methods and technical assistance with qRT-PCR assays, Robin Scharrenberg for helpful advice on automated counting and planning of mice for in utero electroporations, Birgit Schwanke, Saskia Siegel, Maike Voß, Laura Maria Marcelin, Ayob Aleko, and Stine Behrman for technical assistance, and Jane Visvader (WEHI, Melbourne, Australia) and Bennett Novitch (UCLA, Los Angeles, USA) for antibodies. We also thank Laura Frangeul and Sarah Homann for initial help with CTB experiments and *Pum2* KO mouse generation, respectively. The vector and ES cells used for *Pum2* targeting were generated by the trans-NIH Knock-Out Mouse Project (KOMP) and obtained from the KOMP Repository (https://www.komp.org). NIH grants to Velocigene at Regeneron Inc (U01HG004085) and the CSD Consortium (U01HG004080) funded the generation of gene-targeted ES cells for 8500 genes in the KOMP Program and archived and distributed by the KOMP Repository at UC Davis and CHORI (U42RR024244). FCdA was supported by Deutsche Forschungsgemeinschaft (CA 1495/4-1 and CA 1495/7-1), ERA-NET NEURON (BMBF, 01 EW1910 and 01 EW2108B), and JPND (BMBF, ED1806). This work was supported by an Alexander von Humboldt Foundation Postdoctoral Fellowship to KH, a DAAD Doctoral Fellowship to NN, and a grant from the Hamburg Landesforschungsförderung to KED. MH was partially funded by a scholarship (ID: seventh plan 2012–2017) from the Cultural Affairs and Missions Sector, Ministry of Higher Education of the Arab Republic of Egypt.

## Additional information

### Funding

| Funder | Grant reference number | Author |
| --- | --- | --- |
| Alexander von Humboldt-Stiftung | | Kawssar Harb |
| Deutscher Akademischer Austauschdienst | | Nagammal Neelagandan |

| Funder | Grant reference number | Author |
|--------|------------------------|--------|
| Hamburg Landesforschungsfoerderung | LFF-FV27b/P9 | Kent Duncan |
| Deutsche Forschungsgemeinschaft | CA 1495/4-1 and CA 1495/7-1 | Froylan Calderon de Anda |
| ERA-NET NEURON | 01 EW1910 and 01 EW2108B | Froylan Calderon de Anda |
| JPND | BMBF,ED1806 | Froylan Calderon de Anda |
| Cultural Affairs and Missions Sector, Ministry of Higher Education of the Arab Republic of Egypt | seventh plan 2012-2017 | Melad Henis |

The funders had no role in study design, data collection and interpretation, or the decision to submit the work for publication.

## Author contributions

Kawssar Harb, Conceptualization, Data curation, Formal analysis, Funding acquisition, Investigation, jointly conceived the project. KH planned and performed most experiments, analyzed, and interpreted data and wrote the manuscript with input from all authors who approved the final version., Methodology, Project administration, Resources, Software, Supervision, Validation, Visualization, Writing – original draft, Writing – review and editing; Melanie Richter, Data curation, Formal analysis, wrote the license application for IUEs and also planned and performed IUEs, Methodology, Writing – original draft, Writing – review and editing; Nagammal Neelagandan, Data curation, Formal analysis, Methodology, planned, performed, analyzed, and interpreted data for CLIP-qRT-PCR and polysomes for TDP43A315T, Writing – review and editing; Elia Magrinelli, planned and performed CTB retrograde labeling, Methodology, Writing – review and editing; Hend Harfoush, Data curation, Formal analysis, performed part of immunostaining and qPCR experiments and quantified FISH images, Methodology, Visualization; Katrin Kuechler, Data curation, generated plasmids for IUE, Methodology, Visualization; Melad Henis, Data curation, Formal analysis, planned and performed polysome profiling experiments with puromycin., Methodology; Irm Hermanns-Borgmeyer, generated Pum2 mouse lines, Methodology, Writing – review and editing; Froylan Calderon de Anda, Conceptualization, Data curation, helped in planning and performing IUE experiments, coordinated the study, planned experiments, analyzed, and interpreted data., Formal analysis, Funding acquisition, Investigation, Methodology, Project administration, Resources, Supervision, Validation, Visualization, Writing – original draft, Writing – review and editing; Kent Duncan, Conceptualization, Data curation, Formal analysis, Funding acquisition, Investigation, conceived the project,coordinated the study, planned experiments, analyzed, and interpreted data and wrote the manuscript with input from all above authors, Methodology, Project administration, Resources, Software, Supervision, Validation, Visualization, Writing – original draft, Writing – review and editing

## Author ORCIDs

Kawssar Harb http://orcid.org/0000-0002-3112-3084
Elia Magrinelli http://orcid.org/0000-0002-5225-5713
Froylan Calderon de Anda http://orcid.org/0000-0003-2743-7774
Kent Duncan http://orcid.org/0000-0003-4290-2670

## Ethics

All animal care and experimental procedures were performed according to institutional guidelines of the UKE Ethical approvals or University of Geneva and relevant national law. In Hamburg, guidelines were those of the UKE Animal Research Facility (FTH) and conformed to the requirements of the German Animal Welfare Act. Ethical approvals were obtained from the State Authority of Hamburg, Germany (G10/107_Pumilio, G14/003_Zucht Neuro, ORG_520, and ORG_765). The Institutional Animal Care and Use Committee of the City of Hamburg, Germany approved all in utero electroporation experiments (Approval N086/2020 acc. to the Animal Care Act, §8 from May 18th, 2006).

## Decision letter and Author response

Decision letter https://doi.org/10.7554/eLife.55199.sa1

Author response https://doi.org/10.7554/eLife.55199.sa2

## Additional files

### Supplementary files

• Source data 1. Quantification of layer II–VI molecular determinants in *Pum2* cKO mutants. Quantification of results from n = 3 mice of controls and Pum2 mutants in the prospective somatosensory cortex (pS) for Sox5, Bcl11b, Rorβ, and DAPI in single bins (*Figure 1a*) and total (*Figure 1—figure supplement 5a*) and Tbr1, Cux1 in single bins (*Figure 1—figure supplement 6a*). All markers are normalized to DAPI cells. Distribution of cells across six equal-sized bins is shown. For Bcl11b, only high-expressing neurons were counted. Data are shown as means ± standard error of the mean (SEM), n = 3 for each genotype. *p≤0.05, **p≤0.01, ***p≤0.001, two-tailed *t*-test. *Pum2* cKO: *Pum2^fl/fl^; Emx1^Cre^*; II–IV, V, VI: layers II–IV, V, and VI.

• Source data 2. Quantification of layer II–VI molecular determinants in TDP43^A315T^ mutants. Quantification of results from n = 3 mice of controls and TDP43^A315T^ in the prospective somatosensory cortex (pS) for Sox5, Bcl11b, Rorβ, and DAPI in single bins (*Figure 1b*) and total (or layer V for Bcl11b) (*Figure 1—figure supplement 5b*) and Tbr1, Cux1 in single bins (*Figure 1—figure supplement 6b*). All markers are normalized to DAPI cells. Distribution of cells across six equal-sized bins is shown. For Bcl11b, only high-expressing neurons were counted. Data are shown as means ± standard error of the mean (SEM), n = 3 for each genotype. *p≤0.05, **p≤0.01, ***p≤0.001, two-tailed *t*-test. *Pum2* cKO: *Pum2^fl/fl^; Emx1^Cre^*; II–IV, V, VI: layers II–IV, V, and VI.

• Source data 3. Quantification of layer V and VI molecular determinants in the frontal/motor (F/M) cortex of Pum2 and TDP-43 mutants. Quantification of results from n = 3 mice of Pum2 and TDP-43 mutants and their control littermates in the F/M for Sox5, Bcl11b, and DAPI in single bins (*Figure 2b*) and total (*Figure 1—figure supplement 5b*) and Tbr1 in single bins (*Figure 1—figure supplement 6b*). All markers are normalized to DAPI cells. Distribution of cells across six equal-sized bins is shown. For Bcl11b, only high-expressing neurons were counted. Data are shown as means ± standard error of the mean (SEM), n = 3 for each genotype. *p≤0.05, **p≤0.01, ***p≤0.001, two-tailed *t*-test. *Pum2* cKO: *Pum2^fl/fl^; Emx1^Cre^*; II–IV, V, VI: layers II–IV, V, and VI.

• Source data 4. Validation of *Pum2* cKO mutants by qRT-PCR. qRT-PCR of E14.5 cortical RNA from controls (*Ctrl*) vs. *Pum2* cKO using primers to the floxed exons. The fold change in expression levels of *Pum2* mRNA normalized to *GAPDH* mRNA in the *Pum2* cKO is shown relative to the *Cre^-^* control (*Ctrl*) in *Figure 1—figure supplement 1c*. Data are shown as means ± standard error of the mean (SEM), n = 3 for each genotype. * p≤0.05, two-tailed *t*-test.

• Source data 5. Quantification of general cortical developmental features in Pum2 and TDP-43 mutants. Quantification of the brain anatomy including hemisphere length, width, and area (*Figure 1—figure supplement 2a*), cortical thickness (*Figure 1—figure supplement 2b*), and nuclei size (*Figure 1—figure supplement 2c*) in Pum2 and TDP-43 mutants. n = 3–6 samples of each genotype. *p≤0.05, two-tailed *t*-test. *Pum2* cKO: *Pum2^fl/fl^; Emx1^Cre^*.

• Source data 6. Quantification of Sox5 expression in the prospective somatosensory cortex (pS) of *Pum2* KO mice. Quantification of results from n = 3 mice of controls and *Pum2 KO* mice in the pS for Sox5 normalized to DAPI in single bins and total (*Figure 1—figure supplement 7*). Data are represented as means ± standard error of the mean (SEM). *p≤0.05, **p≤0.01 by two-tailed *t*-test. *Ctrl*: controls; *Pum2 KO*: *Pum2* constitutive knockout.

• Source data 7. Quantification of TDP-43 overexpression. Quantification of fold changes in protein levels of human TDP-43 (hTDP-43) or both mouse and human (m+h) TDP-43 normalized to total protein in nuclear or cytoplasmic fractions from three mice (n1–3) of each genotype (*Ctrl*, TDP43, or TDP43^A315T^) (*Figure 1—figure supplement 8c*). Data are shown as means ± SEM, n = 3 for each genotype. *p≤0.05, **p≤0.01, ***p≤0.001 by one-tailed *t*-test.

• Source data 8. Quantification of layer IV/V molecular determinants in hTDP-43 mice. Quantification of results from n = 3 animals of controls mice (*Ctrl*) or mice from a transgenic line expressing *Prnp-TARDBP* (TDP43) shown in six equal-sized bins and the total number of Sox5- or Rorβ- or Bcl11b or DAPI-positive cells (*Figure 1—figure supplement 9b*). Only high-expressing Bcl11b+ neurons were counted. Data are shown as means ± SEM, n = 3 for each genotype. *p≤0.05, **p≤0.01, ***p≤0.001 by two-tailed *t*-test. IV, V, VI: layers IV, V, and VI.

• Source data 9. Quantification of subcerebral projection neuron (SCPN) in Pum2 and TDP-43 mutants. Quantification of retrogradely labeled SCPNs in equal-sized bins for the three genotypes.

Analysis of bins 3 and 4 is shown separately and combined (*Figure 3c*). Data are shown as means ± standard error of the mean (SEM), n = 3 for each genotype. \*\*p≤0.01, \*\*\*p≤0.001, two-tailed *t*-test. *Pum2* cKO: *Pum2^fl/fl^; Emx1^Cre^*.

• Source data 10. Quantification of Sox5/Bcl11b colocalization in Pum2 and TDP-43 mutants. Quantification of results from n = 3 brains of controls (*Ctrl*), *Pum2* cKO, or *hTARDBP^A315T^* (TDP43^A315T^) in the prospective somatosensory area (pS) for Sox5 and Bcl11b colocalization across six equal-sized bins (*Figure 4a*). Data are shown as means ± standard error of the mean (SEM), n = 3 for each genotype. \*p≤0.05, \*\*p≤0.01, two-tailed *t*-test. *Pum2* cKO: *Pum2^fl/fl^; Emx1^Cre^*.

• Source data 11. Analysis of frontal motor (F/M) and prospective somatosensory (pS) areas identities. Quantification of results from n = 3 animals from controls (*Ctrl*), *Pum2* cKO, and TDP43^A315T^ for Lmo4 and Bhlhb5 in F/M and pS areas in single bins and total. Results of F/M and pS for both markers are compared between mutants and their controls and between F/M and pS of each genotype. A summary of total cells only is shown independently comparing F/M and pS in each genotype (*Figure 5a*). Quantification of the number of barrels per section (*Figure 5b*) from n = 3 brains of controls (*Ctrl*), *Pum2* cKO, or *hTARDBP^A315T^* (TDP43^A315T^). Data are shown as means ± standard error of the mean (SEM). \*p≤0.05, \*\*p≤0.01, \*\*\*p≤0.001, two-tailed *t*-test. *Pum2* cKO: *Pum2^fl/fl^; Emx1^Cre^*.

• Source data 12. Analysis of TDP-43 gain-of-function effect in vitro on layer IV/V molecular determinants. Quantification of the fraction of Sox5^+^, Bcl11b^+^, or Rorβ^+^ neurons among all transfected neurons with plasmids encoding either control GFP, TDP43, or TDP43^A315T^. At least 50 cells were counted for each replicate of every transfection. Data are shown as means ± standard error of the mean (SEM), n = 3 for each transfection. \*p≤0.05, \*\*p≤0.01, \*\*\*p≤0.001, two-tailed *t*-test.

• Source data 13. Analysis of post-mitotic effect of Pum2 loss-of-function and TDP-43 gain-of-function in vivo on layer IV/V molecular determinants. Quantification of results from *Pum2^fl/fl^* or WT brains at P0 electroporated at E13,5 with pNeuroD-IRES-GFP as control, or with p-NeuroD-IRES-Cre-GFP to ablate Pum2 expression (*Figure 7a*) or p-NeuroD-TDP43-IRES-GFP or p-NeuroD-TDP43^A315T^-IRES-GFP to overexpress hTDP-43 alleles (*Figure 7b*) only in post-mitotic neurons. The fraction of Sox5^+^, Bcl11b^+^, or Rorβ^+^ neurons among all electroporated cells was quantified. Data are shown as means ± standard error of the mean (SEM), n = 3 for each electroporation. Both p-NeuroD-IRES-Cre-GFP and hTDP-43 alleles were co-electroporated with T-dimer (red) to distinguish them from littermate control brains electroporated only with pNeuroD-IRES-GFP. For both hTDP-43 alleles, the respective control littermates for each variant were combined to a total of n = 6 for pNeuroD-IRES-GFP electroporations. \*\*p≤0.01, \*\*\*p≤0.001, two-tailed *t*-test.

• Source data 14. Quantification of mRNA levels of layer IV/V neuronal identity determinants in *Pum2* cKO or TDP43^A315T^ mutants. qRT-PCR of RNA derived from P0 somatosensory area-enriched cortical lysates for *Pum2* cKO (*Figure 8a*) or TDP43^A315T^ (*Figure 8b*). The fold change for *Sox5, Bcl11b, Rorb,* and *Fezf2* mRNAs normalized to *GAPDH* mRNA is shown for mutants relative to respective control samples (*Ctrl*). Data are displayed as means ± standard error of the mean (SEM) for at least n = 4 of each genotype.

• Source data 15. Quantification of mRNA levels of layer IV/V neuronal identity determinants in *Pum2* cKO or TDP43^A315T^ mutants. Quantification of results from single-molecule fluorescent in situ hybridization (smFISH) for *Sox5, Bcl11b, Rorb,* and *Fezf2* mRNAs on coronal sections from the prospective somatosensory area (pS) of controls (*Ctrl*), *Pum2* cKO, and TDP43^A315T^ mice at P0. Distribution of cells across six equal-sized bins (*Figure 8d*). The number of RNA dots in the bins where they are mostly expressed is normalized to the total number of cell nuclei (DAPI) within that bin. Data are shown as means ± standard error of the mean (SEM), at least n = 3 for each genotype. \*p≤0.05 by two-tailed *t*-test. *Pum2* cKO: *Pum2^fl/fl^; Emx1^Cre^*.

• Source data 16. Translational control of layer IV/V neuronal identity determinants by TDP-43 in developing neocortex. Quantification of results from n = 3 experiments of polysome profiling on TDP43^A315T^ cortices at E14.5 (*Figure 8c*). Histograms depict the distribution of the *Sox5, Bcl11b, Rorb,* and *Fezf2* mRNAs across the gradient fractions for TDP43^A315T^ relative to corresponding controls (*Ctrl*). Samples in heavier gradient fractions were virtually pooled at analysis to simplify visualization in the case of the *Bcl11b B1* primer. Levels of specific mRNAs in each fraction were analyzed by qRT-PCR with normalization to an RLuc mRNA spike-in control, which was added in an equal amount to the fractions prior to RNA preparation. Data are shown as means ± standard error of the mean (SEM), n = 3 for each genotype. \*p≤0.05, \*\*p≤0.01, one-tailed *t*-test.

• Source data 17. Translational control of layer IV/V neuronal identity determinants by Pum2 in

developing neocortex. Quantification of results from n = 3 experiments of polysome profiling on *Pum2 cKO* prospective somatosensory area (pS)-enriched cortices at P0 (*Figure 8d*). Histograms depict the distribution of the *Sox5, Bcl11b, Rorb,* and *Fezf2* mRNAs across the gradient fractions for *Pum2* cKO relative to corresponding controls (*Ctrl*). Samples in heavier gradient fractions were virtually pooled at analysis to simplify visualization. Levels of specific mRNAs in each fraction were analyzed by qRT-PCR with normalization to an RLuc mRNA spike-in control, which was added in an equal amount to the fractions prior to RNA preparation. Data are shown as means ± standard error of the mean (SEM), n = 3 for each genotype. *p≤0.05, **p≤0.01, one-tailed *t*-test. *Pum2* cKO*: Pum2*<sup>fl/fl</sup>*; Emx1*<sup>Cre</sup>.

- Source data 18. Expression of *Sox5* splicing isoforms in Pum2 and TDP-43 mutant neocortices. Quantification of expression of *Sox5* splicing mRNA isoforms normalized to *GAPDH* mRNA in P0 somatosensory area-enriched cortical lysates of *Pum2* cKO (*Figure 9—figure supplement 1a*) and TDP43<sup>A315T</sup> (*Figure 9—figure supplement 1b*) mutants and their respective control samples (*Ctrl*). For *Sox5,* 7 protein-coding isoforms were annotated. We designed primers recognizing three of them, and it was not possible to design specific qPCR primers to distinguish the other four isoforms for which we used a primer called Sox5 diff to detect the four of them simultaneously. Data are shown as means ± standard error of the mean (SEM) for at least n = 4 of each genotype. *Pum2* cKO*: Pum2*<sup>fl/fl</sup>*; Emx1*<sup>Cre</sup>. Two-tailed *t*-test.

- Source data 19. Expression of *Bcl11b* and *Rorb* splicing isoforms in Pum2 and TDP-43 mutant neocortices. Quantification of expression of *Bcl11b* and *Rorb* splicing mRNA isoforms normalized to *GAPDH* mRNA is shown in P0 somatosensory area enriched cortical lysates of *Pum2* cKO (*Figure 9—figure supplement 1a*) and TDP43<sup>A315T</sup> (*Figure 9—figure supplement 1b*) mutants and their respective control samples (*Ctrl*). Data are shown as means ± standard error of the mean (SEM) for at least n = 4 of each genotype. *Pum2* cKO*: Pum2*<sup>fl/fl</sup>*; Emx1*<sup>Cre</sup>. Two-tailed *t*-test.

- Source data 20. Expression of *Sox5* 3′UTR isoforms in Pum2 and TDP-43 mutant neocortices. Quantification of expression of *Sox5* 3′UTR mRNA isoforms normalized to *GAPDH* mRNA in P0 somatosensory area-enriched cortical lysates of *Pum2* cKO (*Figure 9—figure supplement 2a*) and TDP43<sup>A315T</sup> (*Figure 9—figure supplement 2b*) mutants and their respective control samples (*Ctrl*). Data are shown as means ± standard error of the mean (SEM) for at least n = 4 of each genotype. *Pum2* cKO*: Pum2*<sup>fl/fl</sup>*; Emx1*<sup>Cre</sup>. Two-tailed *t*-test.

- Source data 21. Expression of *Bcl11b* and *Rorb* 3′UTR isoforms in Pum2 and TDP-43 mutant neocortices. Quantification of expression of *Bcl11b* and *Rorb* 3′UTR mRNA isoforms normalized to *GAPDH* mRNA is shown in P0 somatosensory area-enriched cortical lysates of *Pum2* cKO (*Figure 9—figure supplement 2a*) and TDP43<sup>A315T</sup> (*Figure 9—figure supplement 2b*) mutants and their respective control samples (*Ctrl*). Data are shown as means ± standard error of the mean (SEM) for at least n = 4 of each genotype. *Pum2* cKO*: Pum2*<sup>fl/fl</sup>*; Emx1*<sup>Cre</sup>. Two-tailed *t*-test.

- Source data 22. Analysis of general translation in Pum2 and TDP-43 mutant cortices. Quantification of polysome/monosome (P/M) ratio from polysome profiles of E14.5 neocortices for controls (Ctrl), *Pum2* cKO, and TDP43<sup>A315T</sup> for n = 3 of each genotype (*Figure 9—figure supplement 3a*). Two-tailed *t*-test.

- Source data 23. Translational control of layer V neuronal identity determinants by Pum2 in developing E13.5 neocortex. Quantification of polysome profiling from E13.5 neocortices of *Pum2* cKO (*Figure 9—figure supplement 3b*). Histograms showing the distribution of the *Sox5* and *Bcl11b* mRNAs at E13.5 across polysome gradient fractions for *Pum2* cKO relative to controls. E13.5 is the peak time of birth for layer V neurons when no layer IV Rorβ+ neurons are born yet. Values were normalized to an RLuc mRNA spike-in control, which was added in an equal amount to the fractions prior to RNA preparation. Data are represented as means ± standard error of the mean (SEM). *p≤0.05 by two-tailed *t*-test.

- Source data 24. Translational control of layer V neuronal identity determinants by Pum2 in developing E14.5 neocortex. Quantification of polysome profiling from E14.5 neocortices of *Pum2* cKO (*Figure 9—figure supplement 3b*). Histograms showing the distribution of the *Sox5, Bcl11b,* and *Rorb* mRNAs at E14.5 across polysome gradient fractions for *Pum2* cKO relative to controls. Values were normalized to an RLuc mRNA spike-in control, which was added in an equal amount to the fractions prior to RNA preparation. Data are represented as means ± standard error of the mean (SEM). **p≤0.01 by two-tailed *t*-test.

- Source data 25. Translational control of layer V neuronal identity determinants by Pum2 in developing E18.5 neocortex. Quantification of polysome profiling from E18.5 neocortices of *Pum2* cKO (*Figure 9—figure supplement 3b*). Histograms showing the distribution of the *Sox5, Bcl11b,*

and *Rorb* mRNAs at E18.5 across polysome gradient fractions for *Pum2* cKO relative to controls. Values were normalized to an RLuc mRNA spike-in control, which was added in an equal amount to the fractions prior to RNA preparation. Data are represented as means ± standard error of the mean (SEM). Two-tailed *t*-test.

• Source data 26. Analysis of Pum2 and TDP-43 interaction with mRNAs encoding key regulators of layer IV/V neuronal identity in developing neocortex. Quantification of results from UV cross-linking immunoprecipitation (UV-CLIP) from E18.5 cortices (*Figure 10c*). Dissociated cells were either cross-linked with UV light or left untreated as a control. Lysates were used for immunoprecipitations with antibodies against TDP-43, Pum2, or control nonspecific IgG. RNA in the input and IP eluate were analyzed by qRT-PCR for *Sox5*, *Bcl11b*, *Rorb*, *Fezf2*, *Cux1*, *Pum2*, Tdp43, and *18S* mRNAs. After verifying enrichment relative to IgG controls for UV-treated samples, histograms were generated that represent the fraction of input mRNA co-immunoprecipitated with either Pum2 or TDP-43 in the presence or absence of UV cross-linking. Statistically significant enrichment was evaluated relative to 18S rRNA, which is not known to interact significantly with either protein. Reduced signal in the absence of UV-cross-linking implies an interaction is cross-linking-dependent, that is, direct. Data are represented as means ± standard error of the mean (SEM) from n = 3–6 samples. Raw values and data normalized to 18S of each replicate are shown independently in different sheets, and a summary of consolidated results from six replicates is in the last Excel sheet. *p≤0.05, ** p≤0.01, Mann–Whitney *U* test.

• Source data 27. mRNA expression pattern of *Emx1*, *Sox6*, and *Unc5C*. Quantification of the fold change for *Emx1* mRNA normalized to *GAPDH* mRNA is shown for P0 somatosensory area-enriched cortical lysates of *Pum2* cKO relative to respective control samples (reviewers *Figure 1a*). Quantification of the fold change for *Sox6* and *Unc5C* mRNA normalized to *GAPDH* mRNA is shown for P0 somatosensory area-enriched cortical lysates of *Pum2* cKO and TDP43[A315T] (reviewers *Figure 2a and b*) relative to respective control samples (*Ctrl*). Data are shown as means ± standard error of the mean (SEM) for n = 4-6 animals of each genotype. *p≤0.05 by two-tailed *t*-test.

• Source data 28. Emx1 protein expression in Pum2 mutants. Analysis of results of Western blot performed on nuclear fractions from three mice (N1–3) of *Ctrl* and *Pum2* cKO for Emx1 protein. Quantification of corresponding fold changes in Emx1 protein levels normalized to total protein is shown below. Data are shown as means ± standard error of the mean (SEM), n = 3 of each genotype. two-tailed *t*-test.

• Source data 29. Analysis of *Sox5*, *Bcl11b*, and *Rorb* mRNAs across polysome gradient fractions after puromycin treatment. Quantification of results of polysome profiling from P0 WT somatosensory area-enriched cortices neocortices for controls (Ctrl) and puromycin-treated samples (reviewers *Figure 3*). Histograms showing the distribution of the *Sox5*, *Bcl11b*, *Rorb*, *Fezf2*, *GAPDH*, and *18S* mRNAs across polysome gradient fractions for puromycin-treated samples relative to controls. Values were normalized to an RLuc mRNA spike-in control, which was added in an equal amount to the fractions prior to RNA preparation. Data are represented as means ± standard error of the mean (SEM). *p≤0.05 by two-tailed *t*-test.

• Source data 30. Source data for Western blots. A zipped folder containing original and labeled bands photos for Western blots of Pum2 and tubulin as control (*Figure 1—figure supplement 1e*), human and mouse TDP-43 and total protein stain as control (*Figure 1—figure supplement 8c*), and Emx1 and total protein stain as control (reviewers *Figure 1b*).

• Transparent reporting form

### Data availability

All data generated or analysed during this study are included in the manuscript and supporting files.

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
