## [Editor Report]

All of the reviewers and editors agree that your deep, rigorous, and multidimensional study addresses novel and significant questions, that it will be an important addition to the literature, and that it presents experiments and data that will motivate further study in the field. You have convincingly demonstrated translational modulation of Sox5, Bcl11b, and *Rorb* by the RNA-binding proteins (RBPs) Pum2 and TDP-43, with thoughtful experiments supporting cell-autonomous effects of the RBPs on the translational regulation of Bcl11b, Sox5, and *Rorb*. Your in vivo gain and loss-of-function data in a range of genetically manipulated mouse lines, coupled with in vitro neuronal culture experiments, provide strong evidence for the control that Pum2 and TDP43 exert on these key regulators of neuronal diversification during corticogenesis. Thank you for completing such a deeply informative body of work and for discussing the complexities involved so thoughtfully within your paper.

---

## [Decision Letter]

**Decision letter after peer review:**

Thank you for submitting your article "Pum2 and TDP-43 area-specifically modulate neuronal identity in mouse neocortex via bi-directional translational control" for consideration by *eLife*. Your manuscript has been reviewed by three peer reviewers, including Jeffrey Macklis as the Reviewing Editor and Reviewer #1, and the evaluation has been overseen by a Senior Editor.

The reviewers have discussed the reviews with one another and the Reviewing Editor has drafted this decision to help you prepare a revised submission.

Summary:

The authors investigate the role of post-transcriptional regulation by RNA binding proteins Pum2 and TDP-43 in defining area-specific cytoarchitecture and in specifying neuronal subtype composition within the primary somatosensory cortex (S1) and the frontal/motor cortex (FM). Specifically, the authors characterize the phenotypes of mice with Pum2 knockout in forebrain progenitors (Pum2 cKO) or those with over-expression of wild-type or ALS-causative human TDP-43 (hTDP-43 or hTDP-43 A315T). By combining genetics, immunostaining, and biochemical approaches, this interesting study proposes that a bidirectional mRNA translational control switch, driven by Pum2 and TDP-43, is important to control binary molecular programs that define the specialized neuronal identity and connectivity of cortical region S1.

Either Emx1-driven conditional Pum2 KO or human TDP-43 over-expression is reported to result in "motorization" of Layers V+VI of primary somatosensory cortex. While S1 in these mice retains features of normal arealization such as proper thalamic input, the authors find alteration of its laminar organization: expansion of layer V neurons with protein expression of subcerebral- and broader corticofugal-related transcription factors Sox5 and Ctip2, increased subcerebral projection, and reduction of layer IV neurons expressing Rorβ. The effect is area-specific, since FM remains normal. These results suggest that Pum2 and TDP-43 are necessary for establishment of the cytoarchitecture, but not area identity, of S1. Interestingly, the authors report unchanged Sox5, Ctip2, and paradoxically increased Rorβ mRNA expression in either PUM2 cKO or hTDP-43A315T mice, suggesting that Pum2 and TDP-43 might regulate Sox5, Ctip2, and Rorβ via translation control.

The authors also provide evidence that this effect may be due to direct Pum2 and TDP-43 translational regulation of target mRNAs Sox5, Ctip2, and *Rorb*. Polysome profiling on whole brain or S1-containing cortex shows a monosome-to-heavy-polysome shift for Sox5 and Ctip2 transcripts as evidence for their enhanced translation, as well as the opposite trend for Rorβ, in both E14.5 hTDP-43 A315T mice and P0 Pum2 cKO mice. The authors also report that Pum2 and TDP-43 interact physically with the three mRNA species. With these lines of evidence, the authors suggest that Pum2 and TDP-43 act as two opposing parts of a switch to regulate translation of subtype-specific regulators of S1 cortex downstream of arealization.

While this is an interesting study, and relatively novel in its question, there are several important points that would need to be addressed to support the conclusions of this study, which are at times not supported by the results, and often lacking statistical clarity.

Essential revisions:

The concept that both RNA binding proteins regulate cytoarchitecture specifically in the somatosensory cortex via translational regulation is quite novel and interesting. However, the data presented are insufficient to rule out other mechanisms in addition to and/or beside translational regulation by Pum2 and TDP-43, and there are a number of questions about data acquisition, data analysis, additional undiscussed phenotypes, selection of model mouse lines, among others. As such, the current submission is not suitable for publication without substantially modifying the claims and interpretations and/or undertaking substantial further experiments to investigate, assess, confirm, or rule out the role of transcription and/or splicing in the specific neurons under consideration, and to consider broader phenotypes seemingly not considered.

In addition, a more comprehensive introductory discussion of why these two RNA binding proteins were chosen for study and why the human over-expression mouse lines were employed instead of alternative options would be very helpful for readers, and important to understand the context and whether this work has conceptual specificity to these RBPs vs. broader implications of the reported mechanism(s)- either could be fine, but enabling the reader to understand would help.

In light of the COVID-19 pandemic and the shutdown of the majority of laboratories, all reviewers support extending revision time per the new *eLife* editorial policy during COVID-19 lab closures and limitations. We find that there might potentially be much of interest, but there is need for very substantial revision with new experiments, analyses, framing, and very likely reduced claims. That said, we see potential for seriously undertaking these major changes toward successful revision.

1. Do the expression patterns in M1/S1 of endogenous Pum2 and TDP-43 match the predicted model for a simple 'switch'? Are these two RBPs expressed in an area-specific manner that would support the overall model?

2. Why was human (as opposed to mouse) TDP-43 used in the over-expression studies? Although the data are supplemented by use of a WT allele in culture, justification for using a mutated human allele of TDP-43 in the first place was lacking. In order to evaluate the physiological role of TDP43 in regulating neuronal identity in specific areas of the cortex, would other models be more appropriate? Recently, several knock-in mouse models have been generated (Ebstein et al., 2019) which allow to studying disease-related mutations on TDP-43 function and the dose-dependent effect of mutant TDP-43. This line of experiments needs clarification and stronger motivation, and seems weaker than other lines of investigation, since it represents a human mutant protein-in-mouse over-expression experiment without any clear justification.

3. When over-expressing a human protein several issues require consideration. Constitutive over-expression (under the PRNP promoter) of a human protein could affect the downstream machinery of protein regulation, complicating the analysis on layer V subcerebral neurons. Also, a potential competition with the endogenous murine protein can occur. The authors have shown that one of the TDP43wt lines does not have a sustained expression of the human protein, and therefore is not suitable for their analysis. It appears that at least the other human wild type TDP43 overexpression L2 model, which shows instead a small, yet significant, difference in the number of Sox5 and Ctip2 in pS, should be used throughout as a reference to estimate the function of the wild-type human protein and the defects induced by the specific mutation. This is missing in most of the analyses of the manuscript. The manuscript should be restructured including this control line from the beginning, and clearly state what are the functional effects of over-expressed hTDP43 on layer V development, and what are the effects induced by the mutant ALS-related form. Connectivity effects in the L2 model should also be addressed.

4. An important point is that Emx1 is itself involved in neocortical area patterning (deletion of Emx1 can lead to motor area expansion). No stock number is given for the Emx-Cre line used, but a reference to Iwasato et al., 2000 would suggest this is the KI Emx1-Cre (Emx1 disruption allele). If so, the cKO mice in this study would also be missing one copy of functional Emx1. Can the authors show that the Emx-Cre transgene does not affect Emx1 levels? While Figure 1's supplement 5 partially addresses the concern about the Cre line, the data are limited (can neural connectivity by retrograde tracing data be shown for the constitutive Pum2 KO? This line of investigation is further motivated below.) and the effects of the Cre line itself on Emx1 mRNA/protein expression levels would be critical to know.

5. Since the determination of neuronal identity and connectivity are assessed by imaging-based measures, it would be important to use some form of unbiased stereological counting of the reported markers, as well as normalization to some unchanged cell marker to account for differences in staining/tissue quality between biological replicates. This concern applies more broadly across methods employed in multiple data analyses.

Manually Photoshop quantifications are extremely inaccurate when it comes to discriminating packed cell nuclei in particularly dense regions of the brain, such as upper layers of the somatosensory cortex in sections. An alternative, more suitable, ideally automated, method should be chosen (from the many available, free resources) for any quantification present in the manuscript, and consistency of the measurements should be assured when analyzing the different molecular markers in all the models and their controls.

6. Pum2 and TDP-43 have been shown to regulate thousands of mRNAs across many studies. Although this study reports that Pum2 and TDP-43 both bind directly to Sox5 and *Rorb*, these two genes themselves may not be the ones directly responsible for the proposed 'switch.'

7. Connected to this issue is another major concern, the seemingly over-simplistic suggested mechanism of action of these two RBPs. Upregulation of a few layer specific markers (individually estimated) upon over-expression of a mutant human protein (hTDP43 mutant) seems unlikely to explain a physiological modulation of a binary fate decision in neurons. Strong statements should therefore be toned down. It would probably be more faithful to the data to discuss the regulation of specific layer markers by RBPs, unless further analysis on the hTDP43 L2 or event murine ko/kI can support those statements.

On the other hand, to better define the time window of action, it would be really interesting to develop a strategy to revert the effect of Pum2 deletion and bring back the "normal" number of layer V Ctip2 and Sox5 and *Rorb*, and ideally their connectivity.

8. Related to the previous two points, can the authors provide any evidence that genes that are in turn regulated by Sox5 and Ctip2 are altered in expression by non-imaging-based measures? This would be a good addition to the imaging-heavy data and support the overall hypothesis.

9-25. Throughout the figures and related text, there are confusing aspects. The authors should improve the presentation of data in the figures, and clarify the numerous analysis, quantification, statistical, and other issues raised consistently by all three reviewers.

9. Figure 1: It is not clear why quantitation is to DAPI as opposed to a neuronal marker (e.g. NeuN+ cells), since it is in theory possible that glial cell numbers are changed. Also, not clear why Ctip2+ cell quantification is different across panels 1a and 1b. And is it true that only ~10% of DAPI+ cells would express Ctip2 in the F/M region as the quantitation (but not presented image) suggests in 1b?

10. The analysis of the different subtype and layer markers is not adequately organized. Why have different measurement criteria been used to quantify cell percentage in the layers for each marker in Figure 1 (% TF positive cells/dapi /bin vs cells FC increase/ ctrl)? From the images, it appears that there is an upregulation/ectopic expression of CTIP2 (as shown for Sox5 in bin 5) also in the upper layers of hTDP43A315T. Showing the data as is done for Sox5 would help clarify this. Tbr1 staining (shown in supplementary Figure1-4a) shows a small, albeit significant, increase in hTDP43A315T, which should be reported in a similar way. These data together, if confirmed by blinded / automated analysis as discussed above, might suggest an increased number of deep layer identity neurons with a concomitant decrease of RobB expressing neurons in P0 pS1 in both Pum2cKO and hTDP43A315T mutants.

11. Further raising question is that the statistical methods are not clear: How many litters have been analyzed? The legend states n=3 for each genotype. Are they all coming from one litter? How are the t-Test performed? Are the animals littermates? It would be useful to clarify to which comparison the asterisks refer in the graph. Again here, the proper control should be reported (ideally both hTDP43wt and NT).

12. Co-expression of Ctip2 and Sox5, at least in pS1, also would be interesting to understand the extent of increased number of layer V neurons in both models.

13. Also, the sections shown in Figure 1 (both pS and F/M) display a strong difference in cortical thickness in the mutants, especially in the hTDP43 A315T line. The quantification of the DAPI positive nuclei does not seem to reflect such evident differences, not even in bin 5 and 6 in pS for the hTDP43A315T line, where the differences appear to be striking. The higher cell density in the ULs is also clearly visible in Figure 2, where the increased thickness of hTDP43 A315T cortex also appears. Further raising question is that nuclei in both mutants also seem to be smaller.

14. Are the overall brains larger in experimentals (it appears so)? Gross morphological characterizations of the different models should be reported, ideally with histological staining (e.g. Nissl), both at birth. Cortical thickness quantification (which can be performed on DAPI staining too) would allow for a basic understanding of the overall cortical architecture, which is critical to fully interpret the results.

15. One major concern of the whole papers regards the controls. There are several independent mouse lines compared throughout the study (i.e. Pum2 conditional, Pum2 KO, hTDP43 A315T, hTDP43 wt, NT). It is important that littermates are compared, then, averaged data cross-compared with the other lines for developmental studies. It is unclear both in the figures and in the text what is considered control in each location. In the Methods, Pum floxed mice and NT are described as control, but in the figures (i.e. Figure 1) only one control is shown. How was this selected? The authors should clarify this in each case, and, when possible, add the proper internal control to each experiment.

16. Overall Figures 1-3 and associated supplemental figures: The characterization of area identity, cytoarchitecture, and axon projection phenotypes in experimental mice is qualitatively fairly convincing. The figures appear to show that Pum2 cKO or hTDP-43, wildtype or mutant, overexpression exhibit S1-specific laminar change: increase in Sox5+/Ctip2+ cells in layer V, decrease in Rorß+ cells in layer IV, and increase in subcerebral projections to the pons. In addition, S1 areal identity seems likely preserved, as the expression pattern of *Lmo4* and Bhlhb5, "motor" and "somatosensory" area marker respectively, do not appear from the images shown to change, and neither does the stereotypical "barrel" pattern of thalamocortical innervation.

That said, in Figure 2, it would be helpful to combine retrograde labeling of SCPN with staining for the markers of interest, Sox5 and Ctip2, to test whether the increased retrogradely labeled neurons in S1 directly correspond with the increased number of Sox5+/Ctip2+ neurons.

17. However, area-specific layer markers such as *Lmo4* and Bhlhb5, and even barrel analysis, are only qualitatively reported in Figure 3. Robust quantifications, as per Figure 1, are required with the appropriate controls to draw such a central conclusion for the overall story. It is also confusing that, while in the lower magnification panel a clear layer of Bhlhb5 positive cells appears to be present in Pum2 cKO F/M, in the magnified image, the TDP43A315T cortex instead shows Bhlhb5 ectopic expression in the deep layers.

18. There are no bidirectional data for either Pum2's or TDP-43's effects. To show genetic necessity and sufficiency, Pum2 over-expression and TDP-43 cKO experiments would be needed as well. Figure 4 demonstrates that overexpression of wild-type TDP-43 is sufficient to drive an increase in Layer V Sox5+/Ctip2+ neurons and a decrease in Layer IV Rorβ+ neurons in S1, and immediate transfection of wild-type or mutant hTDP-43 into E18.5 primary neuron cultures is sufficient to cause similar expression changes. When lab access allows, it would be interesting to directly test the in vivo sufficiency of TDP-43 over-expression to induce subtype change, as well as extending this assessment to Pum2 knockout. One could perform in utero electroporation (IUE) of either hTDP-43 or Cre (into a PUM2fl/fl background) to test whether the electroporated cells also misexpress Sox5, Ctip2, and Rorβ, and aberrantly project subcerebrally. Since the authors have positive results in E18.5 primary culture with hTDP-43 overexpression, and find evidence for Pum2 post-mitotic mode of action, this IUE experiment at E12.5 to hit both layer V and IV progenitors, or even E14.5 to test upper layer progenitors, would seem feasible and informative, and quick once labs are accessible. While not absolutely necessary for the scope of this study, such experiments would strengthen the interpretations, and the Discussion section should at least discuss these limits of interpretation.

19. Figure 4—figure supplement 1-2 show evidence for gain-of-function over-expression of TDP-43 in hTDP-43 transgenic lines. The authors should discuss the apparent expression pattern of hTDP-43 transgenes in the cortex in more depth: compared to hTDP-43 (line 1) or wild-type TDP-43, "hTDP-43 L2" and hTDP-43A315T seem to be expressed more highly in superficial layer neurons. Why is this the case, and why does this not cause Sox5, Ctip2, Rorβ expression in superficial layer neurons? In addition, the western blots show increased TDP-43 protein level in the nucleus but not in the cytoplasm, for both hTPD-43 A315T and hTDP-43 L2. The authors should discuss how these predominantly nuclear changes in TDP-43 expression affect Sox5, Ctip2, and Rorβ expression through translational control. Since global cytoplasmic TDP-43 levels are not statistically different, it is difficult to reconcile these results with a purely cytoplasmic (translational) mechanism. In this regard, it would be advised to substantially "soften" the title and text to acknowledge something like "at least partially via translational control", once new experiments are completed, and assuming that they confirm this.

20. In addition, these supplemental figures appear to be out of order and are quite confusing. Figure 4S2 would seem to better go before Figure 4S1, because it is mentioned first in the text. In particular, the immunostainings in Figure 4S2 should come first, as they provide the proper context for interpreting the rest of Figure 4. In addition, the associated text ("TDP-43 gain-of-function.… cell autonomously" result section) is confusing because the first hTDP-43 line doesn't have a distinct name. Perhaps better to list together all the names of the transgenic lines near the paragraph's beginning before phenotype description: "hTDP43-L1", "hTDP43-L2", and "hTDP-43A315T".

21. Figure 5 and Figure 5—figure supplement 1 examine steady-state mRNA levels of Sox5, Ctip2, Rorβ, and Fezf2 with either smFISH in P0 S1 or qRTPCR in E14.5 cortical lysates. The data currently do not convincingly rule out the possibility of mRNA level changes of these transcripts (another of multiple reasons identified by all three reviewers to soften the interpretation, text, and title). Although not statistically significant, there is a trend toward higher Sox5 and Ctip2 signal. In addition, smFISH is likely not the most accurate method to quantify mRNA levels. One option for a more quantitative experiment that is area and layer specific would be to use at least relatively layer V or IV-specific Cre-driver (such as Rbp4-Cre for layer V1), micro-dissect S1, sort labeled neurons, then examine expression in them via qRT-PCR. Further investigation of potential mRNA expression changes of these genes in the appropriate neurons is critical because an alternative hypothesis explaining the change in mRNA association with heavy polysomes seen in Figure 6 is that there are simply changes in the number of neurons expressing the genes, rather than the translational efficiency of the mRNAs in S1. This alternative would essentially negate/substantially reduce the central claim of the manuscript, so more deeply investigating that alternative would seem to be critical, not an incremental "bell or whistle". All reviewers concur that substantial experiments need to be performed to confirm and/or refute aspects of the interpretations and conclusions presented.

22. Figure 6 and Figure Supplements infer the translational status of Sox5, Ctip2, and Rorβ mRNA of interest by testing the association with heavy polysomes. They show increased association for Sox5 and Ctip2, and decreased association for Rorβ, in both E14.5 hTDP-43A315T whole cortex and P0 Pum2 cKO micro-dissected S1 cortex. Interestingly, no changes were seen in E14.5 Pum2 cKO cortical lysates. Overall, the effects seen are quite weak, and likely represent only modest changes in the global translational output from these mRNAs. In addition, there are several concerns over the design of this experiment. First, as a bulk assay, it does not address whether translational regulation of the transcripts specifically occurs in the neuron population of interest. Second, there is circular logic regarding Sox5 and Ctip2: the change in the laminar composition of the cortex might result in increased association with heavy polysomes without any translational regulatory mechanism simply because there are more cells expressing these genes. For Rorβ, the paradoxical increase in mRNA and decrease in heavy polysome association is a more likely case of translational control. Third, there is a possibility that some transcripts found in heavy polysome fractions do not actually associate with translating ribosomes, but co-sediment because of association with other ribonucleoprotein complexes (a valid concern given Pum2 and TDP-43 function as RNA-binding proteins that possibly form large RNA-protein granules). It would be optimal to add a control in the experiment to ensure that Sox5, Ctip2, and Rorβ are truly engaged by ribosomes. Adding puromycin as a polysome disruptor prior to profiling will shift bona-fide translated transcripts toward lighter ribosome fractions. This is likely to be possible in the coming months.

How do the authors explain the paradoxical effect of *Rorb* RNA and protein levels? Why do they exclude that the Pum2 and TDP43 could have a role in regulating the amount of *Rorb* RNA available in the neurons? In addition, despite not being significant, both Sox5 and CTIP2 appear to show a trend of increase. How many replicates were analyzed, and how many litters? It does not seem conclusive, and additional points should be added to finalize the quantification and investigate RNA level involvement.

23. A more direct test of translational regulation, likely beyond the scope of this paper, would be to perform the PUM2 cKO or hTDP-43 overexpression experiments in a "RiboTag" (RPL22-HAfl/fl) background. One could express tagged ribosomes in either layer V or layer IV through specific Cre drivers, immunoprecipitate the tagged ribosomes, then compare ribosome association with the mRNAs of interest between experimental mice. This could be done or at least discussed.

24. A key limitation and missed opportunity of the manuscript is the lack of attention given to alternative splicing and isoform-specific translational regulation. Figure 6 – Supplement 1 shows the importance of this consideration. The figure explores the expression of various 3' UTR variants of Sox5, Ctip2, and Rorβ, finding multiple isoforms expressed at significantly different levels in the wildtype cortex. Surprisingly, considering TDP-43 is reported to be a key splicing regulator (2,3) and TDP-43 binding sites are found on the transcripts of interest, there is no analysis of possible alternative splicing in TDP-43 over-expression in this manuscript. It is possible that differential isoform usage of subtype identity regulators might be the/a mechanism underlying the expansion of layer V/shrinkage of layer IV. Related to this, the qPCR experiment performed on different polysome fractions to determine the translational status of mRNAs frequently contains results from only one isoform-specific primer set (Figure 6c, d). In Sox5's case, "S4" primers capture the longest- but also the least abundant- isoform. Hence, it is possible that the shift to heavy polysome found in Sox5 and Ctip2 is only valid for one isoform, and not the global transcript population. It is also entirely possible that translational control exists, but acts in an isoform-specific manner. However, the current manuscript does not explore this important topic at all, nor seem to really acknowledge or engage it.

25. Figure 7 demonstrates that both TDP-43 and PUM2 proteins localize to the cytoplasm along with the mRNAs of interest in the cortex, and that these proteins specifically associate with the mRNAs of interest in cortical, cytoplasmic lysates. RNA IP experiments are notoriously noisy, and while the authors controlled for enrichment over IgG and no UV conditions, the most appropriate control would be to establish a baseline of IP in PUM2 cKO cortices or wild-type mouse cortices not expressing hTDP-43. Perhaps more importantly, some discussion of prior literature on PUM2 and TDP-43 interactions with these mRNAs of interest (especially relevant CLiP experiments (2-4)) would be a helpful addition. These articles are cited, but their results are not discussed in comparison to the present study's UV-RIP experiments.

26. All reviewers identified apparent oversights or inadequacy in citation of several clearly relevant papers on related topics that set a context and foundation for elements of this work. Some are listed above when discussing related issues, and others are commented on below. These should be corrected:

26a. The manuscript should mention some previous papers that investigate area-restricted neuronal subtype specification; the manuscript now reads as if this has not been encountered previously, nor seemingly even considered. For example the transcription factor Bcl11a/Ctip1 regulates area-specific composition/proportions of neuronal subtypes: cortical Bcl11a/Ctip1 KO causes an increase in SCPN in sensory and visual cortex, but not in the motor cortex (5, 6). Cederquist et al., 2013 similarly addresses this issue re: *Lmo4* control in rostral motor cortex (7). Discussing / incorporating these papers of course would not take away from the novelty of the current work, which focuses on post-transcriptional effectors of specification downstream of molecular-genetic (particularly transcriptional) control.

26b. Curiously, the authors omit Molyneaux et al., 2005 (8), the first report re: Fezf2 (then Fezl) and its control over Ctip2 and subcerebral identity/fate when they cite two later papers on p.11, line 24.

26c. The authors should provide more depth on the motivation for studying TDP-43 and PUM2 in arealization and cortical development specifically. Although these are "classic RNA binding proteins", the rationale for such a detailed look at these RNA binding proteins in particular is not fully explained in the Introduction and Discussion. One might assume that it is because of their connections with motor neuron disease/ALS, but this and/or other reasons should be made clear and explicit early in the manuscript. Also, the observation that both have similar reported cortical organization phenotypes, and both regulate the genes of interest, requires additional discussion regarding potential mechanistic overlap.

26d. In the Intro, the authors should acknowledge that there have been reports on the contribution of RNA-binding proteins in cortical cytoarchitecture such as FMRP (Altered cortical Cytoarchitecture in the Fmr1 knockout mouse, 2019, Frankie H. F. Lee, Terence K. Y. Lai, Ping Su and Fang Liu).

1. Glickfeld, L.L., Andermann, M.L., Bonin, V. and Reid, R.C. Cortico-cortical projections in mouse visual cortex are functionally target specific. Nature Neuroscience 16, 219-226 (2013).

2. Polymenidou, M. et al. Long pre-mRNA depletion and RNA missplicing contribute to neuronal vulnerability from loss of TDP-43. Nature neuroscience 14, 459-468 (2011).

3. Arnold, E.S. et al. ALS-linked TDP-43 mutations produce aberrant RNA splicing and adult-onset motor neuron disease without aggregation or loss of nuclear TDP-43. Proceedings of the National Academy of Sciences of the United States of America 110, E736-E745 (2013).

4. Tollervey, J.R. et al. Characterizing the RNA targets and position-dependent splicing regulation by TDP-43. Nature neuroscience 14, 452-458 (2011).

5. Greig, L.C., Woodworth, M.B., Greppi, C. and Macklis, J.D. Ctip1 Controls Acquisition of Sensory Area Identity and Establishment of Sensory Input Fields in the Developing Neocortex. Neuron 90, 261-277 (2016).

6. Woodworth, M.B., Greig, L.C., Liu, K.X., Ippolito, G.C., Tucker, H.O., and Macklis, J.D. (2016). Ctip1 Regulates the Balance between Specification of Distinct Projection Neuron Subtypes in Deep Cortical Layers. Cell Rep. 15, 999-1012.

7. Cederquist GY, Azim E, Shnider SJ, Padmanabhan H, Macklis JD. (2013). *Lmo4* establishes rostral motor cortex projection neuron subtype diversity. J Neurosci. 2013; 33(15): 6321-6332.

8. Molyneaux BJ, Arlotta P, Hirata T, Hibi M, Macklis JD. Fezl is required for the birth and specification of corticospinal motor neurons. Neuron 2005; 47: 817-831.

[Editors' note: further revisions were suggested prior to acceptance, as described below.]

Thank you for resubmitting your work entitled "Pum2 and TDP-43 area-specifically modulate neuronal identity in developing mouse neocortex via post-transcriptional and post-mitotic mechanisms" for further consideration by *eLife*. Your revised article has been reviewed by 3 peer reviewers, one of whom is a member of our Board of Reviewing Editors, and the evaluation has been overseen by Catherine Dulac as the Senior Editor.

The manuscript has been improved but there are some remaining issues that need to be addressed, as outlined below:

After open discussion between the three reviewers, taking into consideration both (1) the balance of the importance of the work presented vs. its relatively less critical remaining shortcomings and (2) the experimental complexities of the pandemic, the reviewers all agree that this paper should be published with modifications to the text in appropriate sections to address those remaining shortcomings. The reviewers all agree that the authors performed the majority of requested experiments, and provided a highly thoughtful, comprehensive, and insightful response to the initial reviews, addressing most things well. The authors provided a substantial amount of new data in the revised manuscript, despite the difficult times, and these new data address most of the concerns initially raised by all three reviewers. Overall, all reviewers agree that this study addresses novel and significant questions, and presents experiments and data that will motivate further study in the field.

However, the reviewers also agree that the authors should soften some statements (as pointed out in reviewer comments), especially about (1) neuronal identity definition, (2) direct control over neuronal diversification, (3) whether there might also be transcriptional control occurring, in addition to translational control. That said, we agree that the polysome profiling and smFISH experiments in Figure 8 do address the transcriptional vs. translational contribution question to a reasonable degree. The reviewers all request that the authors temper their language regarding their interpretations and conclusions, and include consideration of some possible alternatives/complementary possibilities regarding their findings. This softening and "toning down" of the absoluteness of these and a few other claims would provide the gentle re-framing necessary to be admirably publishable in *eLife*.

The authors should also address the other issues raised in the reviews, whether in one case about the author list by direct communication to the handling editor, or within the manuscript itself.

The authors' additional work and revision substantially solidified an already novel and interesting project, and will be an important addition to the literature. While some aspects still remain partially open to further experimentation and solidification, all reviewers agree that, overall, the work presented will enable and inspire other interesting studies in the field.

Essential revisions:

The reviewers also agree that the authors should soften some statements (as pointed out in reviewer comments), especially about:

1) Neuronal identity definition;

2) Direct control over neuronal diversification;

3) Whether there might also be transcriptional control occurring, in addition to translational control.

The reviewers also all request that the authors

4) Temper their language regarding their interpretations and conclusions, and include consideration of some possible alternatives/complementary possibilities regarding their findings.

*Reviewer #1:*

This is a substantially revised manuscript, with major effort evident to address the limitations and reviewer criticisms raised in initial review. The authors performed the majority of requested experiments, and provided a highly thoughtful, comprehensive, and insightful response to the reviewers.

However, while the authors have convincingly demonstrated translational modulation of Sox5, Bcl11b, *Rorb* by Pum2 and TDP-43, they did not rule out transcriptional regulation as the or a root cause – in particular, they did not perform the suggested experiments to investigate whether there might be transcriptional changes that might drive regulation of these proteins' abundance in developing layer IV/V.

This is a serious oversight, as it potentially undermines the entire claim of "post-transcriptional" regulation. While there is not sufficient evidence to claim Pum2 and TDP-43 function in appropriate S1-specific laminar organization via post-transcriptional control instead of regulation of steady state levels of their target mRNAs, the post-transcriptional effect per se is novel, and will be worthy of reporting once the possibility of transcriptional regulation is properly investigated. For this pivotal reason that could potentially undermine the central conclusion, this manuscript is not currently publishable in *eLife*.

Specific Comments:

Authorship: Denis Jabaudon was listed as an author on the first submission, but not this revision, yet author contributions from "DJ" are listed in this revision. There is no other author with initials DJ. Is Denis Jabaudon no longer an author on the paper intentionally, but included in the author contributions? What is the explanation? Do we know that he requested to be removed as an author? If so, why? If not, why was he removed? Are there any disagreements among the initial author list in terms of interpretations of the data or approach to this revision? Since authorship is conventionally "earned", it is of note that an authorship has been revoked or deleted for any reason following initial submission.

Line 182: Possible typo: "hippocampal significant staining", should this be e.g., "significant hippocampal staining"?

Figure 4: Satisfies the request for co-localization of retrolabeled SCPN with new Sox5+ and Bcl11b+ cells in layer IV. Would be best to also perform a quantification of Sox5+ SCPN-label+ and Bcl11b+ SCPN-label+ double positive cells per bin as done for Sox5+ Bcl11b+ in Figure 4a.

Figure 5 description in Results section "Somatosensory area identity.… being "motorized": A more precise description of *Lmo4* and Bhlhb5 expression patterns in experimental mice is needed, since the patterns do not seem to be "fully wildtype". By acknowledging up-front subtle differences, then highlighting specific evidence showing distinct and unmixed pS and F/M areal identities, the authors could put readers' minds at ease and prevent them from getting distracted from the main argument. The authors' response to reviewer comment 17 would be well suited here, thus could be considered for incorporation into the text.

Figure 7: IUE of either Cre-GFP in a Pum2fl/fl background, or hTDP43-GFP constructs under NeuroD promoters, at E13.5 demonstrate increased Sox5+ or Bcl11b+ cells among the electroporated neurons (mostly layer IV or upper layer neurons in WT or Cre- conditions), and a decrease in *Rorb*+ neurons. This is consistent with the previous observations using Pum2 cKO or TDP-43 transgenic lines, and a direct test of the model that these RNA binding proteins regulate the relative proportions of cortical lamina. We understand that the authors were unable to perform E12.5 IUE successfully; this is a difficult experiment, and E13.5 IUE seems acceptable given the developmental timing of layer IV differentiation.

Is there is a change in axonal connectivity toward subcerebral projection of the electroporated population that is consistent with an increase in the number of cells expressing Sox5 and Bcl11b, and consistent with the retrograde labeling result? Such an "optional" analysis could make the manuscript more complete, and could be done relatively easily (perhaps especially so if the authors have saved extra samples for tissue processing and microscopy).

Figure 8: Initial review raised concerns that the smFISH method used to quantify Sox5 and Bcl11b mRNA expression in Pum2 cKO or TDP43A315T lines is not accurate. The current figure quantifies expression using the metric "mRNA dots/DAPI cells". However, the FISH signal does not appear to be especially dot-like, and the numbers imply multiple dots per cell, when it looks like the signal largely fills most of the cell. Might this quantification be more appropriate by defining cell positions, integrating fluorescence intensity within a cell, and then comparing the distributions of intensities?

Figure 9 and supplements: The authors' investigation of potential differences in Sox5, Bcl11b, *Rorb* splicing and 3' UTR usage is admirable. That said, the suggested investigation of isoform-specific translational regulation would not require a "tour de force of multi-omic data integration"; the authors could simply repeat their qPCR analysis of their existing polysome profiles with their isoform-specific qPCR primers, and test if any isoforms show changes in the % of each isoform in the gradients, as is done in Figure 9 without isoform-specific primers. The authors' inclusion of a detailed explanation of polysome profiling analysis and quantification is also a positive; this is very helpful to the diverse audience for this paper, and for interpreting the translational changes.

However, the authors did not perform suggested and critical experiments focusing on quantifying Sox5, Bcl11b, and *Rorb* mRNA levels in layer IV/V, citing difficulties obtaining layer-specific Cre-driver lines, e.g. Rbp4-Cre for layer V or *Rorb*-Cre for layer IV, as well as difficulties obtaining sorter facility access due to COVID restrictions.

Although unfortunate and understandable that this might require a longer revision period, this is a serious limitation. Initial review was very clear that these sets of experiments are crucial for full interpretation of the hypothesis of post-transcriptional regulation by Pum2 and TDP-43. It is both very conceivable and very possible that these proteins actually act primarily by regulating the mRNA abundance of the relevant mRNAs in the subtypes of interest, and the observed translational effects are merely secondary. The only cell population-specific evidence the authors present to argue against transcriptional regulation is the smFISH experiments in Figure 8, which are improved, but remain semi-quantitative at best, and thus insufficient to rule out transcriptional effects.

If Cre-lines experiments remain overly challenging, the following experiment could be performed: layer V and layer IV neurons have different "birthdates", and can be labeled by BrdU incorporation at different developmental times. It is likely that in the Pum2 cKO, and hTDP-43 lines, there is a shift toward increased layer V specification at the times that typically yield layer IV neurons. The authors could perform a BrdU labeling experiment at E13.5-E14.5, and (1) look to see whether there is an increase in BrdU^+^Sox5+ or Bcl11b+ at P0, and decrease in BrdU^+^*Rorb*+ cells by immunofluorescence in Pum2 cKO and/or hTDP-43 lines, and (2) FACS-purify the BrdU^+^ cells, and perform qRT-PCR for Sox5, Bcl11b, and *Rorb*. This experiment should take only a few weeks to complete, requires no Cre driver lines to be imported, and requires no specialized procedures.

In summary: the polysome profiling experiments demonstrate translational regulation of Sox5, Bcl11b, and *Rorb* in the developing cortex in Pum2 cKO or hTDP-43 overexpression mice compared to wild-type. However, this could be primarily due to transcriptional regulation, and only secondarily with translation effects. To be able to claim "bona-fide" post-transcriptional regulation, the authors would need to rigorously test the alternative hypothesis that Pum2 and TDP-43 regulate mRNA abundance of the genes of interest in the relevant cells. The evidence presented (smFISH) fails to rigorously test this hypothesis. The central conclusion of manuscript relies on this, and would fall apart if the alternative were the case, so the current set of experiments is incomplete without such rigorous tests of the alternative hypothesis.

*Reviewer #2:*

The authors have adequately addressed all of my points, and I support publication of this revised manuscript.

*Reviewer #3:*

The revised manuscript by Harb and colleagues has greatly improved and few key new experiments in support of the cell autonomous effect of the RBPs on the translational regulation of Bcl11b, Sox5 and *Rorb*. In particular, the IUE data as well as the colabeling analysis of Sox5 and Bcl11b, coupled with the in vitro neuronal culture, provide strong evidence for the control that Pum2 and TDP43 exert on these key players of neuronal diversification during corticogenesis.

Nevertheless, some important points raised in the first review have not been fully addressed and leave the reader still puzzled by the interpretations of some analyses.

In particular, the authors' choice of using the TDP43 mutant line is still problematic: while determining whether early developmental defects might contribute to the aetiology of neurodegenerative diseases is a compelling and very timely question – as numerous studies have recently been published along these lines – it is still unclear to me the link between translational regulation in area identity acquisition and the disease-associated mutations. This becomes even more puzzling when considering the chosen line does not develop ALS symptoms and therefore does not represent a true "disease model". Moreover, as previously requested in the reviews, the most suitable control line for the experiments involving the mutant line would have been the TDP43 wild type overexpression mouse model. If the goal was to address the effect of the disease-associated mutation any effect of the mutant line should have been properly assessed and 'normalised' to the wild type line, which – as stated by the authors in the revised manuscript – shows a milder alteration; if, on the other hand, the aim was to investigate the role of the control of TDP43 RPB on neuronal/area identity acquisition in the gain-of-function setting, the most appropriate line to be used should be the wild type line and not the mutant line, independently of the extent of the phenotype observed. The decision of the authors of not carrying along in *all*L the analysis the is therefore arguable and leaves the reader confused on the specific goals of the study.

Connectivity data: While the IUE experiments undoubtedly contributes to support a direct involvement of RBPs in the phenotypes observed by the authors and convincingly determine their control over canonical markers of neuronal subtypes, the lack of connectivity analysis in Pum2 ko as well as in the TDP43 wild type lines limit the finding to the cellular phenotypes. While convincing the data on the cKO and the mutant TDP43 lines, it might be risky to assume similar connectivity defects in the other contexts.

Rescue analysis: The rescue experiment in the Pum2 cKO or PumKO is not addressed at all, and according to the authors is beyond the scope of the study. We respectfully disagree with the authors about this point. Providing the rescue experiments, or at least attempting it with techniques that have been presented in the revised version of the manuscript like IUE, would have provided direct evidence for Pum2 to be sufficient for the expression of layer-specific markers in vivo, highlighting its physiological relevance in area-specific neuronal identity.

One last point still remains problematic in this reviewer's opinion and it concerns the statistical power of the majority of the analysis in the manuscript (a point already raised in the previous review and that according to the point-to-point response the authors claimed to have addressed). Most of the data (including new experiments and analysis) shown in Figure 1-7, 9-10 as well as in the supplementary figures have been performed on "3 replicates per genotype", and in some rare cases even two dot points are shown in the bar plots. There is no reference in the text about the number of litters or sex of the animals and in some experiments – like IUE – this choice of analysis and data collection might dangerously fall below standards and impact the significance of the results.

Ribosome profiling: the text related to these experiments has become significantly more clear and the logics of the different analysis can be easily followed in the description. However, it is unfortunate that no attempt to resolve the ribosome profiling at the population level (or at least at layer level, as already shown for RNA datasets in multiple publications) has been made by the authors in this revision. This would have provided stronger evidence to the mechanisms underlying the protein alteration and brought an additional level of novelty to the work that the bulk profiling analysis is currently lacking.

In addition, although greatly improved in the flow, the manuscript will still benefit from a more rigorous analysis and quantitative approach to better support the general claims. More specifically:

– The quality of the NeuN images shown in Suppl Figure 2 are strongly divergent and do not look quite comparable. Has there any technical problem occurred that could motivate these differences?

– The nissl staining analysis as presented in Supplementary Figure 3 does not bring relevant information about the cytoarchitecture of the different models, as originally motivated in text, as no quantitative morphometric analyses have been performed, remaining merely qualitative. The overall figure will benefit for additional and improved imaging; indeed, multiple matching sections need to be considered to address overall brain architecture at comparable anatomical levels; higher quality images (the sections seem damaged at the pia level, and it is hard to discriminate the canonical tissue features of the cerebral cortex) coupled by punctual analysis of the higher magnification will help determine whether the evident impairment of the hippocampus observed in Pum cko – not claimed by the authors – is confirmed. Given the area phenotype observed, a more detailed analysis of the internal capsule and the somatic morphology of subcerebral PNs in different areas would have been extremely relevant and is currently missing. As presented, the current figure does not bring definitive support to the interpretations reported in the text and for the phenotype described is key to confirm the overall cytoarchitecture of the cerebral cortex: in several panels, indeed, the cortical thickness of the images shown is not comparable.

In suppl. figure 7 there seem to be large differences in the overall cortical thickness where Pum2ko, ko and hz all show significant smaller cortices compare to the control.

Is this a matching problem, an unfortunate selection of the images or this line presents abnormalities in the cortical thickness? it is hard to conlcude such results from the data. It needs to be toned down.

– In the data reported about nuclear size, what cell types/layer is considered? no information are provided about where are those images shown in Figure suppl 2c are taken, neither if they represent any specific area of the cortex.

– *Rorb* staining in Figure 1 shows a great level of variability among the controls of each mouse lines, which is puzzling considering that the same antibody has been used and an automatic counting method has been used. Do the authors have any explanation for this discrepancy? it is important to assess how reliable is the difference observed in this marker expression. Moreover, in this case the TDP43 control becomes key to use as a reference for the mutant line.

---

## [Author Response]

Essential revisions:The concept that both RNA binding proteins regulate cytoarchitecture specifically in the somatosensory cortex via translational regulation is quite novel and interesting. However, the data presented are insufficient to rule out other mechanisms in addition to and/or beside translational regulation by Pum2 and TDP-43, and there are a number of questions about data acquisition, data analysis, additional undiscussed phenotypes, selection of model mouse lines, among others. As such, the current submission is not suitable for publication without substantially modifying the claims and interpretations and/or undertaking substantial further experiments to investigate, assess, confirm, or rule out the role of transcription and/or splicing in the specific neurons under consideration, and to consider broader phenotypes seemingly not considered.In addition, a more comprehensive introductory discussion of why these two RNA binding proteins were chosen for study and why the human over-expression mouse lines were employed instead of alternative options would be very helpful for readers, and important to understand the context and whether this work has conceptual specificity to these RBPs vs. broader implications of the reported mechanism(s)- either could be fine, but enabling the reader to understand would help.In light of the COVID-19 pandemic and the shutdown of the majority of laboratories, all reviewers support extending revision time per the new eLife editorial policy during COVID-19 lab closures and limitations. We find that there might potentially be much of interest, but there is need for very substantial revision with new experiments, analyses, framing, and very likely reduced claims. That said, we see potential for seriously undertaking these major changes toward successful revision.

We would like to start by profusely thanking the reviewers, both for their general interest in and overall appreciation for the novelty and significance of our work, as well as for their extremely thorough reviewing of our submission and the many well-considered points they raised. We are happy to report that in the extended time for revision that was granted, we were able to address essentially all points, in the main via new experiments where this was relevant. There is certainly no doubt from our side that this manuscript has been greatly improved by all of their time and input!

The issue about needing a better explanation of the decision to focus on Pum2 and TDP-43 and selection of the specific lines also arises several times in different forms under the specific points. We added new text in the Introduction to indicate more clearly why we chose to focus on these RBPs and what broader implications the work could have. We hope the reviewers find the new text helpful in resolving this issue.

1. Do the expression patterns in M1/S1 of endogenous Pum2 and TDP-43 match the predicted model for a simple 'switch'? Are these two RBPs expressed in an area-specific manner that would support the overall model?

In both, the original and revised manuscript, we included data showing that these proteins are not expressed in an area-specific manner (Figure 1—figure supplement 8, Figure 10—figure supplement1). However, we do not see this as a prediction of a simple “switch”. We think it would be one possible simple explanation for a switch mechanism. Alternatives include e.g. area-specific post-translational regulation. Determining the exact mechanism will be challenging, as it is not due to simple differential expression. We see this as interesting future work well beyond the scope of this manuscript.

In addition, since the claim of a “simple switch” seemed to be a bigger general concern, we followed the suggestion to tone down this aspect. Specifically, we changed how we present the “translational switch” in the abstract, introduction, results, and discussion. We now present it as one interesting possibility that is consistent with our data, rather than as a central conclusion.

2. Why was human (as opposed to mouse) TDP-43 used in the over-expression studies? Although the data are supplemented by use of a WT allele in culture, justification for using a mutated human allele of TDP-43 in the first place was lacking. In order to evaluate the physiological role of TDP43 in regulating neuronal identity in specific areas of the cortex, would other models be more appropriate? Recently, several knock-in mouse models have been generated (Ebstein et al., 2019) which allow to studying disease-related mutations on TDP-43 function and the dose-dependent effect of mutant TDP-43. This line of experiments needs clarification and stronger motivation, and seems weaker than other lines of investigation, since it represents a human mutant protein-in-mouse over-expression experiment without any clear justification.

We understand all of the issues raised. It is important to appreciate that we did not choose human TDP-43 over mouse *per se* for our studies. We were interested in the idea that early developmental defects might contribute to etiology of neurodegenerative disease. Thus, our primary goal was to analyze this aspect of cortical development in an established model of the neurodegenerative disease ALS driven by a patient mutant allele of TDP-43. The specific lines used are established ALS models or control lines available from JAX. We already had them in the lab for other projects (e.g. Marques et al., 2020; Neelagandan et al., 2019), so we used them for this project as well and obtained the interesting results reported here.

While we appreciate that arguably more suitable models may have recently become available, we think repeating everything with new mouse models is beyond the scope of this manuscript. Presumably the main concern is that the results might be model-specific artifacts. In that case, we address this by explaining our controls more thoroughly and also with new experimental approaches (e.g. IUE – see completely new Figure 7 with these data).

We modified the text to clarify our choices and motivate this aspect better.

3. When over-expressing a human protein several issues require consideration. Constitutive over-expression (under the PRNP promoter) of a human protein could affect the downstream machinery of protein regulation, complicating the analysis on layer V subcerebral neurons.

This is obviously a possibility and therefore a caveat for all of the numerous published studies that use this approach, including those describing the ALS models and controls that we have used here.

On the other hand, we do not use this approach with *Pum2*, where we have a conditional knockout line and no overexpression of a protein from mouse or human. Thus, this caveat does not apply generally to the data that we present here. In our view, this suggests it is not likely to be the simplest explanation for the phenotypes.

Moreover, we found essentially the same phenotype of a Layer V SCPN molecular determinants increase and layer IV decrease in our IUE experiments. In these new experiments, we over-express hTDP-43 WT and mutant under the control of the pNeuroD promoter. This promoter is highly regulated and is specifically expressed in post-mitotic neurons. Thus, constitutive expression (such as with the Prnp promoter) is not required for the phenotypes with TDP-43 either. Altogether, we do not think the issue raised underlies the phenotypes.

Also, a potential competition with the endogenous murine protein can occur. The authors have shown that one of the TDP43wt lines does not have a sustained expression of the human protein, and therefore is not suitable for their analysis. It appears that at least the other human wild type TDP43 overexpression L2 model, which shows instead a small, yet significant, difference in the number of Sox5 and Ctip2 in pS, should be used throughout as a reference to estimate the function of the wild type human protein and the defects induced by the specific mutation. This is missing in most of the analyses of the manuscript. The manuscript should be restructured including this control line from the beginning, and clearly state what are the functional effects of over-expressed hTDP43 on layer V development, and what are the effects induced by the mutant ALS-related form. Connectivity effects in the L2 model should also be addressed.

We completely agree that this restructuring makes sense and have implemented it in the revised manuscript. Our apologies for any confusion.

We removed all sections regarding the hTDP-43 L1 line and kept only a single WT hTDP-43 line (hTDP-43 L2) which we moved to the beginning. Higher hTDP-43 cytoplasmic overexpression and major phenotypic aspects led us to choose this hTDP-43 over-expressing line in addition to the hTDP-43 A315T. Since primary neurons and in utero electroporations show similar phenotypes with both the WT and mutant allele, we chose to focus our additional analyses (regarding MA, connectivity, arealization, gross morphological features, RNA analysis, polysome profiling) using the hTDP-43 line alone to simplify the experiments and the presentation.

Unfortunately, new connectivity experiments were not possible. Due to the pandemic animal transfers are not as we expected. Moreover, to get the approval in house is not possible right now due to the slowdown of the bureaucratic paperwork, again due to the pandemic. Collectively, these factors made new assays with the Jabaudon lab impossible.

We explored alternatives and were initially excited that a colleague at the ZMNH had some expertise with CTB injection. Unfortunately, they were not familiar with injecting into sub-cerebral regions and lacked the ultrasound device typically used to guide these injections. Moreover, they would also need to obtain approval for these animal experiments, a process which has been proceeding even slower than usual due to the CoViD-19 pandemic.

In summary, while we see the added scientific value of performing these assays, they were logistically impossible even in the extended time-frame of this revision due to the CoViD-19 pandemic. On balance, we think we nevertheless were able to address most of the other key issues raised and hope that this specific aspect will not be considered essential for publication. We try to be clear in the revised text about what we have shown vis-à-vis connectivity.

4. An important point is that Emx1 is itself involved in neocortical area patterning (deletion of Emx1 can lead to motor area expansion). No stock number is given for the Emx-Cre line used, but a reference to Iwasato et al., 2000 would suggest this is the KI Emx1-Cre (Emx1 disruption allele). If so, the cKO mice in this study would also be missing one copy of functional Emx1. Can the authors show that the Emx-Cre transgene does not affect Emx1 levels? While Figure 1's supplement 5 partially addresses the concern about the Cre line, the data are limited (can neural connectivity by retrograde tracing data be shown for the constitutive Pum2 KO? This line of investigation is further motivated below.) and the effects of the Cre line itself on Emx1 mRNA/protein expression levels would be critical to know.

We thank the reviewers for reminding us that the *Emx1::Cre* line described in (Iwasato et al., 2000) is a disruptive knock-in at the *Emx1* locus (we were aware). This raises the possibility that phenotypes in the *Pum2* cKO line (but not hTDP-43 lines) might involve genetic interaction between full loss of *Pum2* and the half genetic dose of *Emx1* in these mice.

However, as the reviewers mention, we addressed this concern already in Figure 1, Figure Supplement 5 of our original manuscript (now Figure 1—figure supplement 7). There we showed – at least qualitatively – that *Emx1::Cre*; *Pum2 ^fl/+^* mice (i.e. *Pum2* cKO heterozygotes) have *no* phenotype. Conversely, we also showed there that the Pum2 constitutive KO (with two WT *Emx1* loci and no Cre transgene present) also shows the phenotypes seen with the Pum2-cKO line. In our view, these observations from our control experiments in our original manuscript are already reasonable evidence that Pum2-cKO phenotypes are not due to reduced dose of the *Emx1* gene.

Consistent with these results, we did not detect a clear reduction in *Emx1* mRNA or Emx1 protein levels in the *Emx1::Cre* line relative to littermate controls (Author response image 1).

**Author response image 1. sa2fig1:** 

Moreover, we also addressed this issue by performing IUE experiments, as suggested under point 18 below. These results are presented in a completely new Figure 7. Briefly, introducing pNeuroD-Cre into the pS of Pum2fl/fl mice via IUE recapitulates the phenotypic effects on layer neuron identity seen in the cKO line. Obviously, these mice have two fully WT copies of the *Emx1* locus.Together with the control experiments described above, we believe that these new data clearly demonstrate that reduced levels of Emx1 expression due to using this *Emx1::Cre* line are not an important factor for the phenotypes seen in Pum2-cKO mice.

Unfortunately, new connectivity experiments were not possible, for reasons described above. Accordingly, in the revised manuscript, we try to be careful about making general conclusions about effects on connectivity, since we were only able to show this with our “core genotypes” and not in every additional analysis. We hope reviewers will find our new presentation of these data to be fair, with the conclusions justified by the underlying data.

5. Since the determination of neuronal identity and connectivity are assessed by imaging-based measures, it would be important to use some form of unbiased stereological counting of the reported markers, as well as normalization to some unchanged cell marker to account for differences in staining/tissue quality between biological replicates. This concern applies more broadly across methods employed in multiple data analyses.Manually Photoshop quantifications are extremely inaccurate when it comes to discriminating packed cell nuclei in particularly dense regions of the brain, such as upper layers of the somatosensory cortex in sections. An alternative, more suitable, ideally automated, method should be chosen (from the many available, free resources) for any quantification present in the manuscript, and consistency of the measurements should be assured when analyzing the different molecular markers in all the models and their controls.

We agree that automated counting is preferable in principle for many reasons. To address this issue, we first identified an automated counting workflow that seemed able to give proper discrimination, as mentioned by reviewers. We then reanalyzed most of our earlier imaging data using automated counting with an Image J-based workflow. Gratifyingly, the new, automated counting results are largely similar to those generated with manual counting and fully support all of our original conclusions. For the revised manuscript, we have updated many figures (Figures 1, 2, 4, 5a and 7 and Figure 1—figure supplement 2, 5, 6 and 9) with the new automated imaging quantification results and associated statistical analyses. We also include new text, particularly in the methods section describing our automated counting procedure in detail. Because the results with automated and manual counting were so similar in cases where we did both, and because packed cell nuclei problems do not apply for CTB labeled neurons, primary neurons in vitro or barrels number and because our 20x FISH mRNA dots could not be counted with our automated image J approach, we retained the manual counting results for certain figures (Figure 3, 5b, 6 and 8 and Figure 1—figure supplement 7). We make it clear in the methods which procedure was followed in each specific case.

6. Pum2 and TDP-43 have been shown to regulate thousands of mRNAs across many studies. Although this study reports that Pum2 and TDP-43 both bind directly to Sox5 and Rorb, these two genes themselves may not be the ones directly responsible for the proposed 'switch.'

It is certainly true that both proteins have been shown to bind to thousands of mRNAs. However, it has also been observed that only a much smaller subset of these mRNAs appears to be detectably regulated when the proteins are depleted or mutant versions are expressed. In any case, it could very well be that other mRNAs are involved in the regulation that we see. In the revised manuscript, we explicitly acknowledge this in the relevant sections.

That said, we think it is important to bear in mind that the examined mRNAs encode proteins that are themselves known to play important roles as transcription factors in driving fate changes – this is particularly true for Sox5 and Ctip2(Chen et al., 2008; Kwan et al., 2008; Lai et al., 2008). Overexpression of Rorβ alone is also sufficient to drive a Layer IV neuronal fate (Jabaudon et al., 2012; Nakagawa and O'Leary, 2003). Thus, if TDP-43 and Pum2 bind them directly and affect their translation in the cortex, this is consistent with a direct regulatory effect on mRNAs encoding proteins known to drive layer neuron fate. Moreover, we examined directly one possible alternative: an effect on the upstream regulator, FEZF2, and obtained data that show TDP-43 and Pum2 do not operate through this regulator. As mentioned already, in the revised manuscript, we now present the “simple switch” driven by translational control of these mRNAs as merely one interpretation consistent with our data.

7. Connected to this issue is another major concern, the seemingly over-simplistic suggested mechanism of action of these two RBPs. Upregulation of a few layer specific markers (individually estimated) upon over-expression of a mutant human protein (hTDP43 mutant) seems unlikely to explain a physiological modulation of a binary fate decision in neurons. Strong statements should therefore be toned down. It would probably be more faithful to the data to discuss the regulation of specific layer markers by RBPs, unless further analysis on the hTDP43 L2 or event murine ko/kI can support those statements.

We apologize for giving the impression that all conclusions were based on overexpressing a human mutant protein, but we also see this effect with WT hTDP-43 protein, consistent with the common belief in the field that the hTDP-43^A315T^ mutant retains almost all WT function. In addition, we see these phenotypes with conditional loss of murine *Pum2* (*Pum2* cKO), which is the same strategy used by countless papers in the field. Moreover, we also see these effects cell-autonomously both in vivo by IUE (new figure 7) and, as reported in the original manuscript, in vitro with transfection of primary SA neurons (Figure 6).

We think a key issue is not how many other regulatory targets there might be, but could deregulation of these specific mRNAs be sufficient to explain the observed effects? As already mentioned above under point 6, the specific proteins deregulated are not merely “individually estimated markers”, but are themselves previously characterized to be important regulators of cortical neuronal identity in the relevant layers.

Nevertheless, we endeavored in the revised manuscript both to “tone down strong statements” and to incorporate the notion that other factors may also contribute. We hope the reviewers find the new presentation to be a fully accurate representation of the underlying data.

On the other hand, to better define the time window of action, it would be really interesting to develop a strategy to revert the effect of Pum2 deletion and bring back the "normal" number of layer V Ctip2 and Sox5 and Rorb, and ideally their connectivity.

We agree that these experiments would be really interesting, but find them beyond the scope of this manuscript.

8. Related to the previous two points, can the authors provide any evidence that genes that are in turn regulated by Sox5 and Ctip2 are altered in expression by non-imaging-based measures? This would be a good addition to the imaging-heavy data and support the overall hypothesis.

We performed new experiments with non-imaging methods to evaluate the levels of some downstream targets which are now presented in Author response image 2. Sox5 is known to repress Fezf2 expression until all layer VI neurons are born (Kwan et al., 2008; Lai et al., 2008). Our qRT-PCR of both Fezf2 and Ctip2 on both mutants doesn’t show any effect on their mRNAs (Figure 8-Figure supplement 1), suggesting a translational regulation of Ctip2 mRNA rather than transcriptional control through Fezf2 or Sox5. Sox5 and Sox6 are known to be cross-repressive (Azim et al., 2009) in cortical progenitors. We analyzed Sox6 mRNA by qRT-PCR in prospective somatosensory region of both Pum2 and TDP-43 mutants and found that Sox6 is significantly down-regulated in Pum2 cKO (consistent with increased Sox5 expression) but not in hTDP-43^A315T^ (Author response image 2) suggesting possibly different timing actions and cell compartments (progenitors/post-mitotic neurons) for both RNA binding proteins. In addition, Ctip2 and Satb2, are known to negatively regulate Unc5C and DCC respectively to regulate sub-cerebral vs callosal axonal projections (Srivatsa et al., 2014). However our qRT-PCR analysis for Unc5C showed no significant changes in both mutants probably due to earlier regulation timing of Unc5C by Ctip2. These new data are now included as Author response image 2.

However, as the reviewers themselves noted below, the non-imaging, biochemical approaches lack resolution and can potentially lead to false-negative results. Single-cell sequencing of sorted neurons would probably be an appropriate method to address this point, but we hope the reviewers will agree that this is beyond the scope of our current manuscript.

9-25. Throughout the figures and related text, there are confusing aspects. The authors should improve the presentation of data in the figures, and clarify the numerous analysis, quantification, statistical, and other issues raised consistently by all three reviewers.

We apologize to the reviewers for any confusing aspects. In the revised manuscript we thoroughly re-worked all figures and adjusted the text accordingly with the goal to present our data in a manner that is easier to follow and to clarify all issues raised by the reviewers.

9. Figure 1: It is not clear why quantitation is to DAPI as opposed to a neuronal marker (e.g. NeuN+ cells), since it is in theory possible that glial cell numbers are changed. Also, not clear why Ctip2+ cell quantification is different across panels 1a and 1b. And is it true that only ~10% of DAPI+ cells would express Ctip2 in the F/M region as the quantitation (but not presented image) suggests in 1b?

We agree with reviewers that normalizing to a neuronal marker is better than normalizing to DAPI cells. However, since normalizing to DAPI is much more practical than normalizing to NeuN given that NeuN staining is not only nuclear but cytoplasmic as well. Moreover, NeuN staining is not possible in many cases due to antibody species, it is very difficult for us to repeat stainings normalizing to NeuN. Based on our understanding of cortical development, we reasoned that there would be very few astrocytes present in the cortex at the stages analyzed (astrogenesis begins at E18.5-P0). It is also true that microglia infiltration is already taking place in utero; although, those cells are restricted to proliferative places and not in the cortical plate (CP) (e.g, Garcia-Marques and Lopez-Mascaraque, 2013; Ge et al., 2012). Therefore, we reasoned that most cells in the imaged field (CP) would be neurons. In theory, this might be different in our mutants as reviewers mentioned. To address this concern directly with new experiments, we performed new staining with GFAP as a glial marker and NeuN as a neuronal marker to examine whether we have a change in the % of neurons relative to glial cells at P0 in both Pum2 and TDP-43 mutants. As expected, we do not see much glial staining at this stage at baseline, and this is not altered in our mutant lines. This is not a technical issue, since we were able to detect robust glial staining in the hippocampus stained in parallel. Moreover, NeuN neuronal staining was colocalizing with the majority of cells in cortex of controls and mutants. We conclude that glia cells are unlikely to be a significant cell population in neocortical layers at this point in development and our mutant lines do not affect this. These new results are presented in Figure 1, Figure Supplement 2d. We think they support the validity of DAPI normalization.

For Ctip2, we only consider Ctip2 high-expressing neurons which are mainly in layer Vb in S1 and the thick layer V in M1. The % of high-expressing Ctip2^+^ cells to total number of DAPI cells in all layers is around 17% in M1 with our new automated counting. The problem with S1 is that the number is lower and is distributed in bin 3 and 4, and this distribution changes when cortical thickness changes along the tangential axis of the cortex. For this reason, we previously grouped bins 3 and 4 into layer V and normalized only to layer V neurons to better show the difference of a small population normalized to very high number of cells. In M1, both quantifications show no differences and we now present the results of the single bins. We understand the concern of the reviewers and to reduce variability in presenting our data between different markers, we changed our figures and represented all markers counting in different bins in main figure 1 and grouped in layer V or total in Figure 1—figure supplement 5. Importantly, all presentation methods shown at the end support our results and conclusions. (single bins, layer specific or total cells) and our primary source data are now submitted with the manuscript.

10. The analysis of the different subtype and layer markers is not adequately organized. Why have different measurement criteria been used to quantify cell percentage in the layers for each marker in Figure 1 (% TF positive cells/dapi /bin vs cells FC increase/ ctrl)? From the images, it appears that there is an upregulation/ectopic expression of CTIP2 (as shown for Sox5 in bin 5) also in the upper layers of hTDP43A315T. Showing the data as is done for Sox5 would help clarify this. Tbr1 staining (shown in supplementary Figure1-4a) shows a small, albeit significant, increase in hTDP43A315T, which should be reported in a similar way. These data together, if confirmed by blinded / automated analysis as discussed above, might suggest an increased number of deep layer identity neurons with a concomitant decrease of RobB expressing neurons in P0 pS1 in both Pum2cKO and hTDP43A315T mutants.

We apologize again for any confusing aspects of our analyses and their presentation. We always performed our analysis in 6 single bins, but as mentioned earlier, in the specific case of Ctip2 we previously grouped bin 3 and 4 to show clearly the effect on layer V, high-Ctip2 expressing neurons. We followed a similar approach for Tbr1 in layer VI (where we grouped bins 1, 2 and 3) and Cux1 for upper layers (bins 4, 5 and 6) because the expression of these genes is mainly restricted to these layers and grouping them may show a phenotype which cannot be observed in different bins due to variability of the binning system in sections with variable thickness across the cortical tangential axis.

We showed the Ctip2 increase as a fold-change compared to controls to better appreciate the differences in a small cell population normalized to all DAPI cells. We understand that this might be confusing and in our revised manuscript we always present the results in 6 bins for all markers (Figure 1,2 Figure 1-Figure Supplement 6), and present additionally either the total number of cells or layer-restricted analyses in supplementary figures (Figure 1-Figure supplement 5). Importantly, all presentation methods shown at the end support our results and conclusions.

Regarding Tbr1, in our original manual counting, we had a slight increase in the case of hTDP-43^A315T^ mutants only when we grouped bins 1, 2 and 3 to present layer VI but not in single bins. In our repeated automated counting, we did not observe a significant change of Tbr1 in either mutant, irrespective of whether we grouped in single bins (as shown in Figure 1-Figure supplement 6) or as total or layer VI-specific (see original counting Tables submitted with the manuscript). We hope this adequately addresses this issue.

11. Further raising question is that the statistical methods are not clear: How many litters have been analyzed? The legend states n=3 for each genotype. Are they all coming from one litter? How are the t-Test performed? Are the animals littermates? It would be useful to clarify to which comparison the asterisks refer in the graph. Again here, the proper control should be reported (ideally both hTDP43wt and NT).

We regret that this crucial issue was not properly addressed in our initial submission. Of course we appreciate how important it is to use littermate controls for the comparisons and this is precisely what we did. In the revised manuscript, we now show a direct comparison of the mutant mice to their corresponding littermate controls with mainly 3 independent different litters in all figures. In some cases, we used more than 3 animals, in this case the number of animals can be seen in figures and we state “at least 3 or 4 or 5” in the legends. For both hTDP-43 lines we used non-transgenic littermates as controls. In the case of *Pum2* cKO, we used *Pum* fl/fl, Cre-negative littermates as controls.

We perform our t-tests in Excel software according to the procedure indicated in our methods section. It is a standard two-tailed test apart from polysome profiling, where we did a 1-tailed T-test, since we already had a directional hypothesis, as mentioned in the methods. Our analyses can be reviewed in our primary data and analysis submission with the revised manuscript. We hope the asterisks in all graphs clearly indicate the relevant comparisons now.

12. Co-expression of Ctip2 and Sox5, at least in pS1, also would be interesting to understand the extent of increased number of layer V neurons in both models.

This is a very important point that we have effectively addressed with new experimental data. As shown in our new Figure 4a, we consistently observed statistically significant increase in the number of cells with co-expression of Ctip2 and Sox5 in all genotypes examined.

To further confirm the identity and connectivity of these neurons, we also performed either Sox5 or Ctip2 co-expression analysis together on the original CTB-labelling in the Pons for retrograde tracing of sub-cerebral projection neuron identity. This confirmed co-expression of all labelled neurons with Ctip2 or Sox5 in controls and in the case of both mutants, all retrogradely labelled neurons expressed Sox5 or Ctip2 as well and also the ones with altered connectivity in upper layer V, exactly as expected suggesting that the increase of Sox5 and Ctip2 colocalize with the increase of SCPN. These new experimental data address point 16 below as well and are presented in a new Figure 4b.

13. Also, the sections shown in Figure 1 (both pS and F/M) display a strong difference in cortical thickness in the mutants, especially in the hTDP43 A315T line. The quantification of the DAPI positive nuclei does not seem to reflect such evident differences, not even in bin 5 and 6 in pS for the hTDP43A315T line, where the differences appear to be striking. The higher cell density in the ULs is also clearly visible in Figure 2, where the increased thickness of hTDP43 A315T cortex also appears. Further raising question is that nuclei in both mutants also seem to be smaller.

We performed a number of additional new experiments to address these issues. Specifically, we measured cortical thickness in the different genotypes in somatosensory cortex (Figure 1-Figure supplement 2c). This did not reveal any significant differences between the experimental genotypes and their respective littermate controls. Analyzing the average thickness in at least 6 images of somatosensory cortex in at least 3 animals of each genotype showed no final significant change between different genotypes. We also increased the number of animals analyzed for DAPI staining in hTDP43 A315T mutants where we had big variability in our original data. Repeating these analysis with automated counting and higher animal numbers did not show any significant change in the number of DAPI cells in single bins or in total in the different genotypes (Figure 1, 2 and Figure 1-Figure supplement 5). We also checked the nuclei size with 60X high magnification images in high number of cells with Fiji and it showed a slight increase in the nuclei size in the case of Pum2 CKO but not A315T mutants. These data now appear in Figure 1-Figure supplement 2b.

14. Are the overall brains larger in experimentals (it appears so)? Gross morphological characterizations of the different models should be reported, ideally with histological staining (e.g. Nissl), both at birth. Cortical thickness quantification (which can be performed on DAPI staining too) would allow for a basic understanding of the overall cortical architecture, which is critical to fully interpret the results.

To address this issue we acquired bright-field images of full brains from controls and mutant animals and analyzed hemisphere length, width and area. This did not reveal any significant differences between the experimental genotypes and their respective littermate controls. These additional new data appear in Supplemental figure 1-Figure supplement 2a.

We also performed Nissl staining. This did not reveal any major morphological changes as far as we could tell. These additional new data now appear in Figure 1-Figure Supplement 3. Collectively, these new results presented together in new Figure 1—figure supplement 2 and 3 support our conclusion that *overall cortical architecture is not altered* in any of the mutant lines relative to littermate controls.

15. One major concern of the whole papers regards the controls. There are several independent mouse lines compared throughout the study (i.e. Pum2 conditional, Pum2 KO, hTDP43 A315T, hTDP43 wt, NT). It is important that littermates are compared, then, averaged data cross-compared with the other lines for developmental studies. It is unclear both in the figures and in the text what is considered control in each location. In the Methods, Pum floxed mice and NT are described as control, but in the figures (i.e. Figure 1) only one control is shown. How was this selected? The authors should clarify this in each case, and, when possible, add the proper internal control to each experiment.

This is related to point 11 above. Once again, we regret that this fundamental issue was not properly presented in our initial submission. Of course, it is crucial to use littermate controls for the comparisons and this is precisely what we did. For the revised manuscript, we now show a direct comparison of the mutant mice to their corresponding littermate controls in the quantifications of all figures.

Moreover, we have endeavored to make it *absolutely* crystal clear for every single figure in the legend and associated text *exactly* what animals were selected for controls in every single case. In multi-animal, qualitative images, we show only one representative control image to simplify presentation – not because we did not process the littermate controls in each individual case. For all quantifications in the revised manuscript, we have included the comparisons to the littermate controls in all analyses. We trust that this crucial issue has now been adequately and convincingly addressed by these changes.

16. Overall Figures 1-3 and associated supplemental figures: The characterization of area identity, cytoarchitecture, and axon projection phenotypes in experimental mice is qualitatively fairly convincing. The figures appear to show that Pum2 cKO or hTDP-43, wildtype or mutant, overexpression exhibit S1-specific laminar change: increase in Sox5+/Ctip2+ cells in layer V, decrease in Rorß+ cells in layer IV, and increase in subcerebral projections to the pons. In addition, S1 areal identity seems likely preserved, as the expression pattern of Lmo4 and Bhlhb5, "motor" and "somatosensory" area marker respectively, do not appear from the images shown to change, and neither does the stereotypical "barrel" pattern of thalamocortical innervation.That said, in Figure 2, it would be helpful to combine retrograde labeling of SCPN with staining for the markers of interest, Sox5 and Ctip2, to test whether the increased retrogradely labeled neurons in S1 directly correspond with the increased number of Sox5+/Ctip2+ neurons.

We performed new experiments that we think effectively address this concern (see related point 12). Co-staining for Sox5 and Ctip2 co-expression together with CTB-labelling confirmed co-expression of Ctip2 and Sox5 in the neurons with altered connectivity, exactly as expected. These new experimental data are presented in a completely new Figure 4b.

17. However, area-specific layer markers such as Lmo4 and Bhlhb5, and even barrel analysis, are only qualitatively reported in Figure 3. Robust quantifications, as per Figure 1, are required with the appropriate controls to draw such a central conclusion for the overall story. It is also confusing that, while in the lower magnification panel a clear layer of Bhlhb5 positive cells appears to be present in Pum2 cKO F/M, in the magnified image, the TDP43A315T cortex instead shows Bhlhb5 ectopic expression in the deep layers.

Robust quantifications as per Figure 1of *Lmo4*/Bhlhb5 (using automated counting) are now included in the revised manuscript in Figure 5. Quantitative analysis of total number of *Lmo4* and Bhlhb5 cells normalized to DAPI showed major significant differences between motor and somatosensory cortex in all genotypes, as expected, suggesting that the pS maintains its areal identity and doesn’t show an F/M identity.

Comparison of *Lmo4* and Bhlhb5 analysis between controls and mutants in different bins and in total is not presented in figures, but is provided in the primary data counting table associated with manuscript. These showed no significant changes in *Lmo4* and Bhlhb5 in the pS of *Pum2* cKo compared to controls, but a significant increase of *Lmo4* in bin1 and decrease of Bhlhb5 in bins 3 and 4 in hTDP-43 A315T. In the case of the motor cortex, an increase in Bhlhb5 has been observed in bin 6 of hTDP-43 A315T while *Lmo4* has been unaltered in all bins. Instead Pum2 cKO didn’t show any change for Bhlhb5 but *Lmo4* expression is decreased in bins 1 and 4. These differences do not affect our conclusion regarding the unchanged areal identity of pS or F/M.

Moreover, we performed a manual counting of barrels across different P7 sagittal sections of controls and mutants and showed no significant changes in the number of barrels. Conservation of the barrels in the somatosensory cortex both qualitatively and quantitatively further confirms our conclusion about the conserved identity of this area in the lines examined.

18. There are no bidirectional data for either Pum2's or TDP-43's effects. To show genetic necessity and sufficiency , Pum2 over-expression and TDP-43 cKO experiments would be needed as well. Figure 4 demonstrates that overexpression of wild-type TDP-43 is sufficient to drive an increase in Layer V Sox5+/Ctip2+ neurons and a decrease in Layer IV Rorβ+ neurons in S1, and immediate transfection of wild-type or mutant hTDP-43 into E18.5 primary neuron cultures is sufficient to cause similar expression changes. When lab access allows, it would be interesting to directly test the in vivo sufficiency of TDP-43 over-expression to induce subtype change, as well as extending this assessment to Pum2 knockout. One could perform in utero electroporation (IUE) of either hTDP-43 or Cre (into a PUM2fl/fl background) to test whether the electroporated cells also misexpress Sox5, Ctip2, and Rorβ, and aberrantly project subcerebrally. Since the authors have positive results in E18.5 primary culture with hTDP-43 over-expression, and find evidence for Pum2 post-mitotic mode of action, this IUE experiment at E12.5 to hit both layer V and IV progenitors, or even E14.5 to test upper layer progenitors, would seem feasible and informative, and quick once labs are accessible. While not absolutely necessary for the scope of this study, such experiments would strengthen the interpretations, and the Discussion section should at least discuss these limits of interpretation.

We separately address the two concerns raised, bidirectionality and IUE.

Bidirectionality: It is true that we have not addressed bi-directionality in the genetic sense – by overexpressing and knocking out both TDP-43 and Pum2 and seeing opposite phenotypes. However, we do not claim to have done so or interpret our data as if we did. We agree that these experiments would be informative, but do not view their absence as a major flaw.

It occurred to us that this issue might have arisen because we referred to “bidirectional translational control” in the original title and text. For R1, we have modified the title and text to remove any references to bi-directionality and found a different way to describe the fact that we find evidence that Pum2 and TDP-43 can both repress and activate translation of specific mRNAs, a point that we consider important and worth highlighting.

IUE assays: To address this point we performed IUE experiments in collaboration with the group of Dr. Froylan Calderon de Anda at the ZMNH. The specific IUE injections performed and results are briefly described here. We started by electroporating wild type mice at E12. as requested either with pNeuroD-GFP or with pNeuroD-hTDP-43 or pNeuro-hTDP-43^A315T^ coupled with T-dimer. However, we didn’t have any success with these electoporations since we didn’t see any GFP positive brain from this stage. Even when we co-electroporated with Tdimer, we didn’t detect any Red fluorescent brains apart in very few cases where electroporated red cells didn’t show as well any green expression. This suggest either that the hTDP-43 over-expression at this stage is toxic and is leading to cell death, or that pNeuroD expression is almost undetectable when electroporated at E12.5 or both. For this reason, we made our IUE at early E13,5, the peak time of birth of layer V neurons (see new Figure 7) versus IUE at late E14,5 to hit mainly upper layer progenitors (Figure 7—figure supplement 2) which both worked very well. Also, since our phenotype is on layers IV and V, we think E13.5 is a suitable stage for our analysis. Our in utero electroporations at E13,5 yielded in vivo data for both hTDP-43 and hTDP-43^A315T^ electroporation that were strikingly congruent with those obtained via in vitro transfection of primary neurons. Moreover, by introducing Cre recombinase into the *Pum2 fl/fl* background under the control of the pNeuroD promoter at E13.5, we also found upregulation of Sox5 and CTIP2, as well as downregulation of Rorβ (See new Figure 7). These data are completely consistent with the data obtained in Pum2-cKO and KO mice. These new data are presented in a new main figure 7 and Figure 7—figure supplement 1 and 2 and their implications are discussed in corresponding text sections.

We find it worth emphasizing here that these experiments were actually considered “optional” by the reviewers. Moreover, while they were certainly feasible and informative, they were definitely not “quick”. We first had to apply for animal experiment approval from the relevant authorities in order to be allowed do these experiments at all. This is normally a time-consuming process under the best of circumstances, but it proved even more challenging during the pandemic: it took >6 months to get approval.

Challenges notwithstanding, we are glad we decided to delay resubmission for these assays, since we believe results from IUE experiments significantly strengthen our conclusions in a number of important ways. First, they demonstrate direct cell-autonomous fate-switching in pS driven by either the absence of Pum2 or the overexpression of TDP-43 (WT or the A315T patient mutant). In addition, the genes expressed in these assays were under control of a post-mitotically activated promoter in pNeuroD, which should not be active in progenitors (Guerrier et al., 2009). Thus, these experiments strongly suggest that these proteins act cell autonomously and post-mitotically in newly born neurons to control cell-fate in pS. This is a completely new observation for both proteins, whose roles have previously only been examined and interpreted in the context of effects in neural progenitors.

Finally, as a completely orthogonal assay performed in WT mice, our IUE results also address many of the specific concerns raised above about whether phenotypes might potentially be artifacts related to the specific mouse lines used. We clearly see the same phenotypes with IUE in a WT background, so they are certainly not line-dependent artifacts.

When we performed IUE of either pNeuroD cre in Pum2 flox or hTDP-43 or hTDP-43 ^A315T^ in wt animals, none of the mentioned experiments resulted in a change of fate in upper layer neurons. This is consistent with our in-vivo analysis of Pum2 CKO and hTDP-43 mutants. None of them showed a phenotype in upper layer neurons suggesting both, an area specificity and layer specificity of the function of our RBPs. Even though they are expressed ubiquitaryly in most cortical cells, they act specifically on the interface between layers IV and V of the somatosensory cortex, leading to the fine adjustment of the thickness of layers IV/V in the somatosensory cortex with respect to its function. One interpretation could be that both, Pum2 and TDP-43 functions, are controlled by an early upstream regulator of area and layer identity.

19. Figure 4—figure supplement 1-2 show evidence for gain-of-function over-expression of TDP-43 in hTDP-43 transgenic lines. The authors should discuss the apparent expression pattern of hTDP-43 transgenes in the cortex in more depth: compared to hTDP-43 (line 1) or wild-type TDP-43, "hTDP-43 L2" and hTDP-43A315T seem to be expressed more highly in superficial layer neurons. Why is this the case, and why does this not cause Sox5, Ctip2, Rorβ expression in superficial layer neurons? In addition, the western blots show increased TDP-43 protein level in the nucleus but not in the cytoplasm, for both hTPD-43 A315T and hTDP-43 L2. The authors should discuss how these predominantly nuclear changes in TDP-43 expression affect Sox5, Ctip2, and Rorβ expression through translational control. Since global cytoplasmic TDP-43 levels are not statistically different, it is difficult to reconcile these results with a purely cytoplasmic (translational) mechanism. In this regard, it would be advised to substantially "soften" the title and text to acknowledge something like "at least partially via translational control", once new experiments are completed, and assuming that they confirm this.

Two separate issues are raised. One relates to upper layer expression and lack of phenotypes there. The other centers on the apparent absence of increased steady-state levels of TDP-43 in the cytoplasm and how this can be compatible with translational effects. We address each below, beginning with the issue of cytoplasmic overexpression.

We now observe increased TDP-43 in the cytoplasm, consistent with effects on translation

We repeated these immunoblots to enable a direct comparison of controls to hTDP-43 L2 and hTDP-43 A315T and removed hTDP-43 L1. Our new WB experiment presented now in Figure 1-Figure supplement 8 shows a significant increase in both cytoplasmic and nuclear expression of total (m+h) TDP-43. Overexpression of TDP-43 in the cytoplasm of both transgenic lines is consistent with regulatory effects in this compartment (e.g. on translation).

Transgene expression in upper layers – why no phenotype there?

We adjusted the text to better describe the apparent expression pattern of the transgenes. This point (regarding no phenotype in UL) is discussed just above and confirmed by our IUE at E14.5.

20. In addition, these supplemental figures appear to be out of order and are quite confusing. Figure 4S2 would seem to better go before Figure 4S1, because it is mentioned first in the text. In particular, the immunostainings in Figure 4S2 should come first, as they provide the proper context for interpreting the rest of Figure 4. In addition, the associated text ("TDP-43 gain-of-function.… cell autonomously" result section) is confusing because the first hTDP-43 line doesn't have a distinct name. Perhaps better to list together all the names of the transgenic lines near the paragraph's beginning before phenotype description: "hTDP43-L1", "hTDP43-L2", and "hTDP-43A315T".

We grouped both Figure 4 S1 and S2 into a single Figure 1-Figure Supplement 8 and presented the staining data before the western blot as requested.

We also decided to remove the L1 line from the manuscript entirely, since it became superfluous with new L2 data and new data from IUE experiments. This should simplify things and reduce confusion.

21. Figure 5 and Figure 5—figure supplement 1 examine steady-state mRNA levels of Sox5, Ctip2, Rorβ, and Fezf2 with either smFISH in P0 S1 or qRT-PCR in E14.5 cortical lysates. The data currently do not convincingly rule out the possibility of mRNA level changes of these transcripts (another of multiple reasons identified by all three reviewers to soften the interpretation, text, and title). Although not statistically significant, there is a trend toward higher Sox5 and Ctip2 signal.

To obtain more convincing data addressing this important point, we performed additional qRT-PCR assays with a new cohort of mutant mice and littermate controls in pS at P0 and increased the number of animals used (n = 4-6, Figure 8a,b). In our updated figure with these additional data, we find no evidence for a trend toward mRNA-level changes (presented now in Figure 8). We hope the reviewers will agree that these new data fully support our conclusion that there is no significant difference in the mRNA levels for *Sox5*, *Ctip2* or *Rorβ* in the samples examined.

In addition, smFISH is likely not the most accurate method to quantify mRNA levels.

Despite its limitations, other investigators have used this approach in published studies (e.g. Zahr et al., 2018). Even if smFISH is not the most accurate method on its own, we did not see any other straightforward way to examine mRNA levels with the requisite spatial resolution. For the revised manuscript, we also increased the number of images/animal and the number of animals analyzed (n=4 for most of them) in the smFISH assays to improve reliability (Figure 8c).

Taken together with the clear absence of tissue-wide effects in the bulk qPCR data, we are convinced that this completely independent method provides sufficient data supporting our conclusion that mRNA level changes do not underlie the protein-level changes we see.

One option for a more quantitative experiment that is area and layer specific would be to use at least relatively layer V or IV-specific Cre-driver (such as Rbp4-Cre for layer V1), microdissect S1, sort labeled neurons, then examine expression in them via qRT-PCR. Further investigation of potential mRNA expression changes of these genes in the appropriate neurons is critical because an alternative hypothesis explaining the change in mRNA association with heavy polysomes seen in Figure 6 is that there are simply changes in the number of neurons expressing the genes, rather than the translational efficiency of the mRNAs in S1. This alternative would essentially negate/substantially reduce the central claim of the manuscript, so more deeply investigating that alternative would seem to be critical, not an incremental "bell or whistle". All reviewers concur that substantial experiments need to be performed to confirm and/or refute aspects of the interpretations and conclusions presented.

We appreciate the elegance and potential added value of an experiment like the one proposed. In theory it could overcome some of the caveats of the two methods that we have used already to address potential mRNA level changes. However, due to work limitations in the pandemic period, we were unable to import new mouse lines (considering we need to do embryo transfer and our animal facility is not running at full capacity). Moreover, sorting, and sequencing facilities are devoted to COVID-related projects. Therefore, the waiting list is huge. We regret this, but hope that reviewers will find our interpretation of the data to be appropriate.

Further investigation of potential mRNA expression changes of these genes in the appropriate neurons is critical because an alternative hypothesis explaining the change in mRNA association with heavy polysomes seen in Figure 6 is that there are simply changes in the number of neurons expressing the genes, rather than the translational efficiency of the mRNAs in S1.

While we greatly appreciate the reviewers’ helpful feedback and find their thoroughness admirable, we do not think this particular caveat is logically correct. We think the issue probably arises from insufficient explanation of how our sucrose gradient polysome assay works and what it can reveal. In fact, a big advantage of the polysome gradient approach is that it reveals the *percentage* of total mRNA signal in each of the different fractions. An mRNA will only show an altered distribution across the gradient if it is associating with heavy complexes (e.g. ribosomes) to a greater or lesser extent. There is no theoretical reason to believe that total mRNA level changes- even if occurring – would affect the *percentage* of the mRNA in one fraction or another. A 10-fold increase (or decrease) in levels of mRNA X in the absence of translational regulation would therefore be expected to lead to no change whatsoever in the percentage distribution of the mRNA across the gradient. This will be true whether the increase in mRNA levels is within the same set of cells or results from more cells in the analyzed population expressing that gene. More cells or mRNA ≠ mRNA deeper in gradient!

Countless publications and our own experience over the years provides empirical support for the fact that the polysome gradient assay is independent of mRNA level measurements. Indeed, it can even reveal translational regulation that is occurring either in the presence or absence of mRNA level changes and might even be “paradoxical” (i.e. opposite direction to mRNA level changes). One example appears to be Rorβ (see point 22 below): mRNA levels go up, but the mRNA shifts out of the deeper fractions of the gradient, consistent with reduced translation of more mRNA, and the observed reduction in protein levels.

Bottom line: We disagree that a change in the number of cells expressing *Sox5*, *Ctip2* or any gene is an alternative explanation for an altered distribution of that mRNA in the polysome gradient. If this is indeed the major concern of the reviewers, then we trust that our explanation of how the polysome assay is not affected by mRNA level changes will alleviate their concerns.

In the revised manuscript, we added new text in Results and Discussion to improve our explanation of how sucrose-gradient polysome assays work and why they are mRNA-level independent.

22. Figure 6 and Figure Supplements infer the translational status of Sox5, Ctip2, and Rorβ mRNA of interest by testing the association with heavy polysomes. They show increased association for Sox5 and Ctip2, and decreased association for Rorβ, in both E14.5 hTDP-43A315T whole cortex and P0 Pum2 cKO microdissected S1 cortex. Interestingly, no changes were seen in E14.5 Pum2 cKO cortical lysates. Overall, the effects seen are quite weak, and likely represent only modest changes in the global translational output from these mRNAs. In addition, there are several concerns over the design of this experiment. First, as a bulk assay, it does not address whether translational regulation of the transcripts specifically occurs in the neuron population of interest.

In the revised manuscript, we now explicitly acknowledge the limitation of the polysome gradient approach as a bulk assay in the Discussion. However, because the assay can provide a window on translational regulation we think it still has significant value. In our view, the key issue with bulk assays would actually be *false negative* results due to lack of sensitivity. On the other hand, modest *positive* effects in the global translational output are exactly what one might expect in such a bulk assay for regulation that is occurring mainly in a sub-population of cells. In that sense, it is arguably impressive that we are able to detect significant effects at all. This is especially true for *Sox5* mRNA, which is also expressed strongly in neurons in Layer VI, where we do not see any phenotype.

We also note now in the discussion that the effect on Rorβ mRNA is actually not so weak: ~50% of this mRNA shifts from a fraction corresponding to ~7 ribosomes/mRNA to a fraction corresponding to ~1 ribosome/mRNA (Figure 9c and d). We think such a change in ribosome density could readily account for a biologically meaningful reduction in protein synthesis and steady-state levels of the encoded Rorβ protein – exactly as we observe.

We agree that having a quantitative, cell-specific assay for translation rates for each of these mRNAs would be very useful. However, as explained earlier for point 21, introduction of neuron-specific cre lines or FACS sorting of specific neuronal cohorts is beyond the scope of the revision of this manuscript. We hope that this crucial issue has been adequately addressed by changes we included in figures and text.

Second, there is circular logic regarding Sox5 and Ctip2: the change in the laminar composition of the cortex might result in increased association with heavy polysomes without any translational regulatory mechanism simply because there are more cells expressing these genes. For Rorβ, the paradoxical increase in mRNA and decrease in heavy polysome association is a more likely case of translational control.

We thought hard about this concern, trying to understand it. As far as we can tell, it is exactly the same issue as point 21, namely that more cells expressing a gene might affect the % of the encoded mRNA found in specific fractions of polysome gradients.

We do not see any theoretical reason why having more cells expressing a particular mRNA would increase the *percentage* of that mRNA in the deeper fractions of a polysome gradient (that is what is plotted in all polysome gradient figures). Moreover, we know empirically from countless published studies (e.g. Blair et al., 2017; Floor and Doudna, 2015, 2016) and all of our own work with this assay over the past decade (e.g. Neelagandan et al., 2019), that changes in the mRNA level in either direction do not per se have any impact on the distribution of that mRNA in the gradient

Thus, we do not understand this concern and we do not believe that there is any “circular logic” here for *Sox5*, *Ctip2*, or any other mRNA/protein pair examined. We think this concern arose from our need to better explain how the polysome gradient assays work. As mentioned under point 21, we added new text in Results and Discussion to help all readers better understand how sucrose-density gradient polysome assays work. There, we emphasize how their design enables us to provide insights into translation independent of any changes to mRNA levels.

Third, there is a possibility that some transcripts found in heavy polysome fractions do not actually associate with translating ribosomes, but co-sediment because of association with other ribonucleoprotein complexes (a valid concern given Pum2 and TDP-43 function as RNA-binding proteins that possibly form large RNA-protein granules). It would be optimal to add a control in the experiment to ensure that Sox5, Ctip2, and Rorβ are truly engaged by ribosomes. Adding puromycin as a polysome disruptor prior to profiling will shift bona-fide translated transcripts toward lighter ribosome fractions. This is likely to be possible in the coming months.

It is true that mRNA shifts in a polysome gradient might be due to association with large complexes other than ribosomes. However, here the changes in the gradients correlate with protein level changes that are not explained by mRNA level changes. We think this makes the interpretation that they reflect ribosome density changes more likely.

Nonetheless, we agree it would be optimal to add a control to ensure that *Sox5*, *Ctip2*, and *Rorβ* mRNAs are truly engaged by ribosomes. We tried a *post lysis* puromycin treatment. In this case, we generated lysates in the absence of cycloheximide then raised lysate temperature briefly to 37 degrees in the presence of 2mM puromycin. The idea was that this would allow for a very brief resumption of translation on the mRNAs in the lysates, so puromycin would have time to act selectively on these elongating ribosomes. As can be seen in our Author response image 3, we were not able to observe any difference in polysome association of any mRNAs in the tissue under the conditions we tested. One potential explanation is that translation did not actually resume under the cell-free polysome buffer conditions. These buffer conditions are optimized to *stabilize* polysomes and not for in vitro translation elongation post-lysis. Since we also have extensive experience in the group with cell-free translation assays, we know all too well that slight changes in buffer components can massively affect translational activity in cell lysates. Regardless of the underlying cause for puromycin not working under the conditions we tested, this disappointing result suggested to us that it might take quite some time to identify robust conditions enabling efficient and selective disruption of actively translating ribosome-mRNA complexes in lysates from mouse neocortex.

**Author response image 3. sa2fig3:** 

The problem is that puromycin is not really a “polysome disruptor” per se. Puromycin can only function as desired when the ribosomes are indeed actively in the act of translating. This mechanism of action makes it ideal for disrupting translation elongation taking place under physiological conditions in living cells. However, it raises issues for designing a good “polysome disruption” experiment with puromycin for lysed tissue samples from mouse cortex.When exactly do you add puromycin “prior to profiling”? And importantly: at what temperature? To stabilize polysomes in their physiological state one chills the tissue prior to lysis and keeps everything cold thereafter. This makes sense: translation elongation, which we want to stall in a physiological state, involves GTP hydrolysis and is absolutely temperature dependent. Ergo, if you just add puromycin to polysome lysates made from chilled tissue and then stored on ice for stability, it should not actually do anything helpful: under these conditions there is no active translation taking place for it to disrupt.

For cultured cells you can simply add puromycin directly to the cells in the dish and return them to the incubator briefly prior to lysis (e.g. Nottrott et al., 2006). In this case, it is also feasible to add cycloheximide either just before or during lysis as a polysome stabilizer. But how do you do this in an informative way with dissected mouse neocortical tissue? Do you disrupt cells first and then puromycin treat ex vivo at 37 degrees like for cultured cells prior to lysis? Or should you perfuse the mice with puromycin before dissecting the tissue? Or do you try to treat the lysates themselves? Our attempts at the latter did not succeed and led us to believe that this is a tricky control to properly implement with dissected tissue.

We sincerely regret that we were unable to address this particular concern by performing the suggested control experiment. In the revised manuscript and we now explicitly acknowledge the implications of not performing this control in the Discussion.

How do the authors explain the paradoxical effect of Rorb RNA and protein levels?

We were not completely sure what type of explanation reviewers were seeking here, so we do our best to cover several potential issues.

Presumably the reviewers appreciate that one simple explanation for paradoxical changes between mRNA and protein levels is translational regulation. Examining the percentage distribution of an mRNA in a polysome gradient can help support this notion. For example, if the percent distribution of *Rorb* mRNA would shift from heavy polysomes to lighter ones, this would be consistent with decreased ribosome engagement with this mRNA, leading to reduced protein synthesis. Such a result would support reduced translation of the increased amount of mRNA being responsible for paradoxically reduced levels of Rorβ protein. This is exactly what we find. In the revised manuscript, we show it in the new Figure 9

An alternative explanation to explain increased *Rorβ* mRNA with decreased protein would be that there are mitigating effects on protein stability. This is not mutually exclusive with translational regulation. In the text, we acknowledge this could also be occurring. However, we propose that protein stability effects are less likely to be the main driver for two reasons: (1) we have direct evidence (via the polysome assays) for regulatory effects on translation and (2) it seems easier to imagine how mRNA-binding proteins regulate translation of mRNAs they bind vs. stability of the proteins encoded by those mRNAs. We reworked the relevant sections of text to make these points clearer.

Probably, these issues are clear and the reviewers’ question is more about why there is an increase in Rorβ mRNA at all? Obviously, it could be due to effects on transcription and/or stability and could be either direct or indirect. One possibility could be activation of a compensatory transcriptional pathway that senses the decreased protein levels and tries to restore balance via upregulating mRNA levels. Based on all we know, a direct effect of Pum2 on Rorβ transcription seems very unlikely, but indirect effects are possible. It is also possible that both proteins directly or indirectly affect stability of the mRNA.

One possible indirect mechanism for effects on Rorβ mRNA stability would actually be consistent with a primary effect on translation (as evidenced from the polysome gradient data) that indirectly affects the stability of the mRNA. The field tends to think in either/or terms, but there is actually a lot of extant data implying that the translational status of an mRNA can impact on the mRNA’s stability. Actually, testing this here would require assays that would be challenging to do in vivo in a complex tissue and are certainly beyond the scope of this manuscript.

In the revised manuscript, we expanded discussion of these issues.

Why do they exclude that the Pum2 and TDP43 could have a role in regulating the amount of Rorb RNA available in the neurons?In addition, despite not being significant, both Sox5 and CTIP2 appear to show a trend of increase. How many replicates were analyzed, and how many litters? It does not seem conclusive, and additional points should be added to finalize the quantification and investigate RNA level involvement.

As explained above (point 21), we have addressed it with new experiments by performing additional replicates with additional litters, exactly as suggested. These new data show *no trend* of increase for Sox5 or Ctip2 (New Figure 8). These observations from additional new experiments fully support our original conclusion that there is no significant difference in the levels of these mRNAs in the cortical regions analyzed.

23. A more direct test of translational regulation, likely beyond the scope of this paper, would be to perform the PUM2 cKO or hTDP-43 overexpression experiments in a "RiboTag" (RPL22-HAfl/fl) background. One could express tagged ribosomes in either layer V or layer IV through specific Cre drivers, immunoprecipitate the tagged ribosomes, then compare ribosome association with the mRNAs of interest between experimental mice. This could be done or at least discussed.

We respectfully disagree that using the Ribotag would be a “more direct test of translational regulation”. In fact, a major caveat of this approach is that it *cannot* by itself distinguish changes in translation from changes in mRNA levels in the cells examined. Changes in mRNA levels in the absence of any translational control will be reflected by corresponding changes in the ribosome-associated mRNA population. For example, if you observe a 10x change in the amount of ribosome-associated mRNA X in your mutant vs. control RiboTag pulldown, this could be either due to changes in that mRNA’s level or its translation or both. There is no way to distinguish whether observed changes reflect translational or transcriptional effects based on RiboTag data alone. Importantly, you cannot reliably reference the input material here to make the call, since this includes confounding signal from other cells.

We believe that this is precisely the caveat raised above for our polysome gradient assays. We reiterate here that this caveat does *not* apply to polysome gradients because the *percentage* of all mRNA signal is distributed across the gradient. Changes in an mRNA’s polysome distribution can be observed whether or not there are changes in mRNA levels and do not depend on the mRNA levels in any obvious manner.

A separate, equally important issue is that tagged ribosome IP approaches (RiboTag or TRAP) are inherently insensitive to changes in ribosome density (the *number* of ribosomes engaged with an mRNA). Theoretically, an mRNA would be expected to be found in the ribosome-associated pool after pulldown whether there are 2, 4 or 10 ribosomes translating it and we have verified in our lab that this is indeed the case (Marques, Stenzler, and Duncan, unpublished). Why does this matter? Because we know that translational control frequently involves changes in the *number* of ribosomes engaged with an mRNA*,* rather than whether *any* ribosomes at all are engaged with it. Such changes can be observed in many published analyses using polysome gradients (e.g. Barbieri et al., 2017; Blair et al., 2017; Floor and Doudna, 2016; Neelagandan et al., 2019) and we also see them here in Figure 9.

We are not saying that RiboTag and related approaches are not useful. They offer the major advantage of cell-type specific analysis of ribosome-associated mRNAs. Moreover, if properly used, they can also help to overcome the caveat that mRNAs may shift in a polysome gradient independently of association with ribosomes. However, they definitely do not by themselves constitute a “more direct test of translational regulation”. They actually provide a less direct and less sensitive assay for translational regulation than the one that we have used here: classical sucrose-density gradient polysome profiling from developing neocortex.

In the revised manuscript, we briefly discuss these aspects in the Discussion, emphasizing the limitations of the Ribotag method and the advantages of the classical polysome approach that we have used here for our specific application.

24. A key limitation and missed opportunity of the manuscript is the lack of attention given to alternative splicing and isoform-specific translational regulation. Figure 6 – Supplement 1 shows the importance of this consideration. The figure explores the expression of various 3' UTR variants of Sox5, Ctip2, and Rorβ, finding multiple isoforms expressed at significantly different levels in the wildtype cortex. Surprisingly, considering TDP-43 is reported to be a key splicing regulator (2,3) and TDP-43 binding sites are found on the transcripts of interest, there is no analysis of possible alternative splicing in TDP-43 over-expression in this manuscript. It is possible that differential isoform usage of subtype identity regulators might be the/a mechanism underlying the expansion of layer V/shrinkage of layer IV. Related to this, the qPCR experiment performed on different polysome fractions to determine the translational status of mRNAs frequently contains results from only one isoform-specific primer set (Figure 6c, d). In Sox5's case, "S4" primers capture the longest – but also the least abundant- isoform. Hence, it is possible that the shift to heavy polysome found in Sox5 and Ctip2 is only valid for one isoform, and not the global transcript population. It is also entirely possible that translational control exists, but acts in an isoform-specific manner. However, the current manuscript does not explore this important topic at all, nor seem to really acknowledge or engage it.

We agree that isoform-specific translational regulation is an interesting aspect that could potentially have been developed more in our manuscript.

For the revised manuscript, we addressed this point by performing additional experimental analyses with new cortical RNA samples from a higher number of replicates in pS of controls and both mutants. We examined potential effects on both 3’UTR diversity and the splicing of the mRNAs identified. As shown in Figure 9-Figure supplement 1 and 2, we observed no changes in levels of the 3’UTR isoforms examined in mutants relative to respective controls. Moreover, guided by annotations in ENSEMBL and the literature, we examined potential effects on the splicing of multiple introns using either custom qRT-PCR primers or previously published RT-PCR primers where available. The conclusion from all of these studies is that there is no apparent effect on the splicing of any of the mRNAs that we see deregulated at the translational and protein levels in developing neocortical tissue.

The revised text now includes explicitly acknowledges that we have considered the possibility of effects on other steps of gene expression and that our new data suggests they may not be the main drivers of the phenotypes we observe. We further acknowledge that there could be subtle isoform- or cell-type-specific effects that we may have missed.

We also revised the text to more explicitly address isoform-specific translational control and the issue of how this might relate to the phenotypes that we observe, as well as how future studies could explore this in more detail, perhaps on a genome-wide level.

Finally, we completely agree that a comprehensive examination of the potential interplay between alternative splicing and polyadenylation and the impact this has on translation during cortical development would be fascinating. However, this vast undertaking would require a tour-de-force of multi-omic data integration and follow-up analyses. As such, it seems far beyond the scope of our manuscript and entirely appropriate for a separate future study.

25. Figure 7 demonstrates that both TDP-43 and PUM2 proteins localize to the cytoplasm along with the mRNAs of interest in the cortex, and that these proteins specifically associate with the mRNAs of interest in cortical, cytoplasmic lysates. RNA IP experiments are notoriously noisy, and while the authors controlled for enrichment over IgG and no UV conditions, the most appropriate control would be to establish a baseline of IP in PUM2 cKO cortices or wild-type mouse cortices not expressing hTDP-43. Perhaps more importantly, some discussion of prior literature on PUM2 and TDP-43 interactions with these mRNAs of interest (especially relevant CLiP experiments (2-4)) would be a helpful addition. These articles are cited, but their results are not discussed in comparison to the present study's UV-RIP experiments.

Two points are raised, one about additional controls and the other about discussing previously published datasets in the context of our CLIP assays.

1. Are the two current controls of control IgG and no-UV crosslinking sufficient?

For Pum2, we agree that the additional suggested control for Pum2-KO material is appropriate and arguably superior to IgG control. However, we also wonder how much additional value it would truly add, given that we already included two standard specificity controls routinely used in numerous published studies. In the end, we decided to focus limited resources in the revision period on other issues that we found more pressing.

For TDP-43, we are analyzing interactions of *endogenous mouse TDP-43* with mRNA in these assays. Therefore, we believe that we cannot think of a better control than the two that we have already used, rather than to establish a baseline in the absence of hTDP-43 (as suggested by the Referees). We reworked the relevant text to emphasize this point in order to minimize potential confusion about this aspect.

2. Discussion of the cited previously published interaction studies

We searched for *Sox5*, *Ctip2*, and *Rorβ* mRNAs in previously published CLIP and RIP datasets. Interactions with Pum2 or TDP-43 were not detected in most studies, but some were found in studies that looked at P0 mouse brains. To confirm and extend these observations, we looked ourselves directly in the developing neocortex. In the revised manuscript we now summarize these previous results in the relevant section of Results.

26. All reviewers identified apparent oversights or inadequacy in citation of several clearly relevant papers on related topics that set a context and foundation for elements of this work. Some are listed above when discussing related issues, and others are commented on below. These should be corrected:

We appreciate the reviewers’ attention to detail and scholarly accuracy, which we also consider important. For the revised manuscript, we incorporated all suggested changes.

26a. The manuscript should mention some previous papers that investigate area-restricted neuronal subtype specification; the manuscript now reads as if this has not been encountered previously, nor seemingly even considered. For example the transcription factor Bcl11a/Ctip1 regulates area-specific composition/proportions of neuronal subtypes: cortical Bcl11a/Ctip1 KO causes an increase in SCPN in sensory and visual cortex, but not in the motor cortex (5, 6). Cederquist et al., 2013 similarly addresses this issue re: Lmo4 control in rostral motor cortex (7). Discussing / incorporating these papers of course would not take away from the novelty of the current work, which focuses on post-transcriptional effectors of specification downstream of molecular-genetic (particularly transcriptional) control.

We now cover this important issue in the Discussion and cite the publications mentioned.

26b. Curiously, the authors omit Molyneaux et al., 2005 (8), the first report re: Fezf2 (then Fezl) and its control over Ctip2 and subcerebral identity/fate when they cite two later papers on p.11, line 24.

We now cite Molyneaux et al., 2005 (8) in addition to the two later papers. We appreciate the reviewers highlighting this important reference to include here.

26c. The authors should provide more depth on the motivation for studying TDP-43 and PUM2 in arealization and cortical development specifically. Although these are "classic RNA binding proteins", the rationale for such a detailed look at these RNA binding proteins in particular is not fully explained in the Introduction and Discussion. One might assume that it is because of their connections with motor neuron disease/ALS, but this and/or other reasons should be made clear and explicit early in the manuscript. Also, the observation that both have similar reported cortical organization phenotypes, and both regulate the genes of interest, requires additional discussion regarding potential mechanistic overlap.

Two points are raised:

(26c.1) motivation for studying Pum2 and TDP-43 in this context

This was raised several times previously. We addressed it by adding text in the Introduction.

(26c.2) Additional discussion regarding potential mechanistic overlap

As suggested, for R1 we added additional text in the Discussion covering the implications of similar phenotypes for the two RBPs and potential mechanistic overlap.

26d. In the Intro, the authors should acknowledge that there have been reports on the contribution of RNA-binding proteins in cortical cytoarchitecture such as FMRP (Altered cortical Cytoarchitecture in the Fmr1 knockout mouse, 2019, Frankie H. F. Lee, Terence K. Y. Lai, Ping Su and Fang Liu).

This is acknowledged in the intro and we now cite the publication mentioned.

References:

Azim, E., Jabaudon, D., Fame, R. M., and Macklis, J. D. (2009). SOX6 controls dorsal progenitor identity and interneuron diversity during neocortical development. Nat Neurosci, 12(10), 1238-1247. https://doi.org/10.1038/nn.2387

Barbieri, I., Tzelepis, K., Pandolfini, L., Shi, J., Millan-Zambrano, G., Robson, S. C., Aspris, D., Migliori, V., Bannister, A. J., Han, N., De Braekeleer, E., Ponstingl, H., Hendrick, A., Vakoc, C. R., Vassiliou, G. S., and Kouzarides, T. (2017). Promoter-bound METTL3 maintains myeloid leukaemia by m(6)A-dependent translation control. Nature, 552(7683), 126-131. https://doi.org/10.1038/nature24678

Blair, J. D., Hockemeyer, D., Doudna, J. A., Bateup, H. S., and Floor, S. N. (2017). Widespread Translational Remodeling during Human Neuronal Differentiation. Cell Rep, 21(7), 2005-2016. https://doi.org/10.1016/j.celrep.2017.10.095

Chen, B., Wang, S. S., Hattox, A. M., Rayburn, H., Nelson, S. B., and McConnell, S. K. (2008). The Fezf2-Ctip2 genetic pathway regulates the fate choice of subcortical projection neurons in the developing cerebral cortex. Proc Natl Acad Sci U S A, 105(32), 11382-11387. https://doi.org/10.1073/pnas.0804918105

Floor, S. N., and Doudna, J. A. (2015). Get in LINE: Competition for Newly Minted Retrotransposon Proteins at the Ribosome. Mol Cell, 60(5), 712-714. https://doi.org/10.1016/j.molcel.2015.11.014

Floor, S. N., and Doudna, J. A. (2016). Tunable protein synthesis by transcript isoforms in human cells. *eLife*, 5. https://doi.org/10.7554/*eLife*.10921

Garcia-Marques, J., and Lopez-Mascaraque, L. (2013). Clonal identity determines astrocyte cortical heterogeneity. Cereb Cortex, 23(6), 1463-1472. https://doi.org/10.1093/cercor/bhs134

Ge, W. P., Miyawaki, A., Gage, F. H., Jan, Y. N., and Jan, L. Y. (2012). Local generation of glia is a major astrocyte source in postnatal cortex. Nature, 484(7394), 376-380. https://doi.org/10.1038/nature10959

Guerrier, S., Coutinho-Budd, J., Sassa, T., Gresset, A., Jordan, N. V., Chen, K., Jin, W. L., Frost, A., and Polleux, F. (2009). The F-BAR domain of srGAP2 induces membrane protrusions required for neuronal migration and morphogenesis. Cell, 138(5), 990-1004. https://doi.org/10.1016/j.cell.2009.06.047

Iwasato, T., Datwani, A., Wolf, A. M., Nishiyama, H., Taguchi, Y., Tonegawa, S., Knopfel, T., Erzurumlu, R. S., and Itohara, S. (2000). Cortex-restricted disruption of NMDAR1 impairs neuronal patterns in the barrel cortex. Nature, 406(6797), 726-731. https://doi.org/10.1038/35021059

Jabaudon, D., Shnider, S. J., Tischfield, D. J., Galazo, M. J., and Macklis, J. D. (2012). RORbeta induces barrel-like neuronal clusters in the developing neocortex. Cereb Cortex, 22(5), 996-1006. https://doi.org/10.1093/cercor/bhr182

Kwan, K. Y., Lam, M. M., Krsnik, Z., Kawasawa, Y. I., Lefebvre, V., and Sestan, N. (2008). SOX5 postmitotically regulates migration, postmigratory differentiation, and projections of subplate and deep-layer neocortical neurons. Proc Natl Acad Sci U S A, 105(41), 16021-16026. https://doi.org/10.1073/pnas.0806791105

Lai, T., Jabaudon, D., Molyneaux, B. J., Azim, E., Arlotta, P., Menezes, J. R., and Macklis, J. D. (2008). SOX5 controls the sequential generation of distinct corticofugal neuron subtypes. Neuron, 57(2), 232-247. https://doi.org/10.1016/j.neuron.2007.12.023

Marques, R. F., Engler, J. B., Kuchler, K., Jones, R. A., Lingner, T., Salinas, G., Gillingwater, T. H., Friese, M. A., and Duncan, K. E. (2020). Motor neuron translatome reveals deregulation of SYNGR4 and PLEKHB1 in mutant TDP-43 amyotrophic lateral sclerosis models. Hum Mol Genet, 29(16), 2647-2661. https://doi.org/10.1093/hmg/ddaa140

Nakagawa, Y., and O'Leary, D. D. (2003). Dynamic patterned expression of orphan nuclear receptor genes RORalpha and RORbeta in developing mouse forebrain. Dev Neurosci, 25(2-4), 234-244. https://doi.org/10.1159/000072271

Neelagandan, N., Gonnella, G., Dang, S., Janiesch, P. C., Miller, K. K., Kuchler, K., Marques, R. F., Indenbirken, D., Alawi, M., Grundhoff, A., Kurtz, S., and Duncan, K. E. (2019). TDP-43 enhances translation of specific mRNAs linked to neurodegenerative disease. Nucleic Acids Research, 47(1), 341-361. https://doi.org/10.1093/nar/gky972

Nottrott, S., Simard, M. J., and Richter, J. D. (2006). Human let-7a miRNA blocks protein production on actively translating polyribosomes. Nat Struct Mol Biol, 13(12), 1108-1114. https://doi.org/10.1038/nsmb1173

Srivatsa, S., Parthasarathy, S., Britanova, O., Bormuth, I., Donahoo, A. L., Ackerman, S. L., Richards, L. J., and Tarabykin, V. (2014). Unc5C and DCC act downstream of Ctip2 and Satb2 and contribute to corpus callosum formation. Nat Commun, 5, 3708. https://doi.org/10.1038/ncomms4708

Zahr, S. K., Yang, G., Kazan, H., Borrett, M. J., Yuzwa, S. A., Voronova, A., Kaplan, D. R., and Miller, F. D. (2018). A Translational Repression Complex in Developing Mammalian Neural Stem Cells that Regulates Neuronal Specification. Neuron, 97(3), 520-537 e526. https://doi.org/10.1016/j.neuron.2017.12.045

[Editors' note: further revisions were suggested prior to acceptance, as described below.]

Essential revisions:The reviewers also agree that the authors should soften some statements (as pointed out in reviewer comments), especially about:1) Neuronal identity definition;2) Direct control over neuronal diversification;3) Whether there might also be transcriptional control occurring, in addition to translational control.The reviewers also all request that the authors4) Temper their language regarding their interpretations and conclusions, and include consideration of some possible alternatives/complementary possibilities regarding their findings.

We thank the reviewers for their overall positive view and willingness to publish our manuscript after the requested text revisions have been implemented

We softened our statements and conclusions regarding all above-mentioned points throughout the text, including the title and abstract. We also updated the text to include a consideration of possible alternative interpretations. Specific cases where we toned down or added additional consideration of possible alternative interpretations are highlighted in R2. Finally, we also address the specific additional points raised by reviewers 1 and 3 below and modified the text where requested.

Reviewer #1:This is a substantially revised manuscript, with major effort evident to address the limitations and reviewer criticisms raised in initial review. The authors performed the majority of requested experiments, and provided a highly thoughtful, comprehensive, and insightful response to the reviewers.However, while the authors have convincingly demonstrated translational modulation of Sox5, Bcl11b, Rorb by Pum2 and TDP-43, they did not rule out transcriptional regulation as the or a root cause – in particular, they did not perform the suggested experiments to investigate whether there might be transcriptional changes that might drive regulation of these proteins' abundance in developing layer IV/V.This is a serious oversight, as it potentially undermines the entire claim of "post-transcriptional" regulation. While there is not sufficient evidence to claim Pum2 and TDP-43 function in appropriate S1-specific laminar organization via post-transcriptional control instead of regulation of steady state levels of their target mRNAs, the post-transcriptional effect per se is novel, and will be worthy of reporting once the possibility of transcriptional regulation is properly investigated. For this pivotal reason that could potentially undermine the central conclusion, this manuscript is not currently publishable in eLife.

We appreciate the potential added value of the proposed experiments and strongly considered performing them. However, it required us to import new Cre lines and cross-breed them prior to performing the experiments. During the pandemic there was tremendous institutional pressure to *reduce* mouse lines, rather than expand them and resources for embryo transfer – normally a bottleneck anyway – were extremely limited. There was simply no way for us to do these experiments even in the extended revision period. We’re glad that the collective discussion appears to have led reviewers to decide not to demand this experiment after all and to accept the results from the bulk qRT-PCR and smFISH as reasonable evidence that there are not major steady-state mRNA level changes. For R2 we have also tried to temper the strength of our claims here as requested and acknowledge caveats of the assays performed, as well as the potential for mRNA level changes within specific cell types.

However, even if we were to detect mRNA level changes using the proposed experiment, that would not in and of itself be evidence for *transcriptional* changes, since mRNA levels could be *post-transcriptionally* regulated at the level of mRNA stability. In that sense, while the proposed experiment is certainly worthwhile, we think it is important to acknowledge that it would also not be definitive for distinguishing transcriptional vs. post-transcriptional effects.

Specific Comments:Authorship: Denis Jabaudon was listed as an author on the first submission, but not this revision, yet author contributions from "DJ" are listed in this revision. There is no other author with initials DJ. Is Denis Jabaudon no longer an author on the paper intentionally, but included in the author contributions? What is the explanation? Do we know that he requested to be removed as an author? If so, why? If not, why was he removed? Are there any disagreements among the initial author list in terms of interpretations of the data or approach to this revision? Since authorship is conventionally "earned", it is of note that an authorship has been revoked or deleted for any reason following initial submission.

We appreciate the potential added value of the proposed experiments and strongly considered performing them. However, it required us to import new Cre lines and cross-breed them prior to performing the experiments. During the pandemic there was tremendous institutional pressure to *reduce* mouse lines, rather than expand them and resources for embryo transfer – normally a bottleneck anyway – were extremely limited. There was simply no way for us to do these experiments even in the extended revision period. We’re glad that the collective discussion appears to have led reviewers to decide not to demand this experiment after all and to accept the results from the bulk qRT-PCR and smFISH as reasonable evidence that there are not major steady-state mRNA level changes. For R2 we have also tried to temper the strength of our claims here as requested and acknowledge caveats of the assays performed, as well as the potential for mRNA level changes within specific cell types.

However, even if we were to detect mRNA level changes using the proposed experiment, that would not in and of itself be evidence for *transcriptional* changes, since mRNA levels could be *post-transcriptionally* regulated at the level of mRNA stability. In that sense, while the proposed experiment is certainly worthwhile, we think it is important to acknowledge that it would also not be definitive for distinguishing transcriptional vs. post-transcriptional effects.

Line 182: Possible typo: "hippocampal significant staining", should this be e.g., "significant hippocampal staining"?

We thank the reviewer for catching this typo, and have now corrected it in R2.

Figure 4: Satisfies the request for co-localization of retrolabeled SCPN with new Sox5+ and Bcl11b+ cells in layer IV. Would be best to also perform a quantification of Sox5+ SCPN-label+ and Bcl11b+ SCPN-label+ double positive cells per bin as done for Sox5+ Bcl11b+ in Figure 4a.

According to our images analysis, and as it can be seen in Figure 4 b and c, we found that all labelled SCPN are Sox5+ and Ctip2+ (100%) as expected, while the opposite is not true since the CTB injection is variable between animals and does not always target all subcerebral axons. We focused on higher magnification images to zoom in on layer IV- and V- labelled neurons, instead of showing a low-level magnification (as in Figure 4a) with all bins included.

Figure 5 description in Results section "Somatosensory area identity.… being "motorized": A more precise description of Lmo4 and Bhlhb5 expression patterns in experimental mice is needed, since the patterns do not seem to be "fully wildtype". By acknowledging up-front subtle differences, then highlighting specific evidence showing distinct and unmixed pS and F/M areal identities, the authors could put readers' minds at ease and prevent them from getting distracted from the main argument. The authors' response to reviewer comment 17 would be well suited here, thus could be considered for incorporation into the text.

We changed this part and include our response to reviewer comment 17 from R1 in the manuscript text.

Figure 7: IUE of either Cre-GFP in a Pum2fl/fl background, or hTDP43-GFP constructs under NeuroD promoters, at E13.5 demonstrate increased Sox5+ or Bcl11b+ cells among the electroporated neurons (mostly layer IV or upper layer neurons in WT or Cre- conditions), and a decrease in Rorb+ neurons. This is consistent with the previous observations using Pum2 cKO or TDP-43 transgenic lines, and a direct test of the model that these RNA binding proteins regulate the relative proportions of cortical lamina. We understand that the authors were unable to perform E12.5 IUE successfully; this is a difficult experiment, and E13.5 IUE seems acceptable given the developmental timing of layer IV differentiation.Is there is a change in axonal connectivity toward subcerebral projection of the electroporated population that is consistent with an increase in the number of cells expressing Sox5 and Bcl11b, and consistent with the retrograde labeling result? Such an "optional" analysis could make the manuscript more complete, and could be done relatively easily (perhaps especially so if the authors have saved extra samples for tissue processing and microscopy).

We first aimed to include this connectivity analysis in our IUE experiments. The best way would have been if samples electroporated at E14.5 would change their connectivity. In that case we would have found GFP-labelled subcerebral projections of the cells over-expressing hTDP-43 alleles or ablating Pum2 compared to no subcerebral projections in the control group. We saved later samples at P7 and P21 to analyze that. However, we found that electroporating UL neurons at E14.5 does not change their identity (Figure 7—figure supplement 2).

Regarding our E13.5 experiments, we aimed to do the same, however many difficulties appeared:

1 – Most but not all our electroporated pups with Cre or hTDP43 ^A315T^ were either not delivered or eaten by the mother at P0. We could mostly save the ones that were picked immediately after delivery. Due to the high number of electroporations that we have done at different stages from E12.5 to E14.5 and our limited time, we didn’t have so many extra brains to save for later stages (P7, P21) or for the different optimal cut (sagittal) for assessment of connectivity. We kept our saved brains for cell identity analysis.

2 – The brains electroporated with Cre or hTDP-43 variants showed a lot of electroporation efficiency differences (rate and GFP intensity of GFP expression in electroporated neurons) between each other and with GFP controls. This variability would make it hard to reliably assess a change in GFP+ subcerebral projections among the relatively low number of brains we had. This intensity-based analysis couldn’t be normalized to the efficiency of electroporation as we have done in the cell identity analysis.

Figure 8: Initial review raised concerns that the smFISH method used to quantify Sox5 and Bcl11b mRNA expression in Pum2 cKO or TDP43A315T lines is not accurate. The current figure quantifies expression using the metric "mRNA dots/DAPI cells". However, the FISH signal does not appear to be especially dot-like, and the numbers imply multiple dots per cell, when it looks like the signal largely fills most of the cell. Might this quantification be more appropriate by defining cell positions, integrating fluorescence intensity within a cell, and then comparing the distributions of intensities?

In our FISH analysis, we used the ACD/Bio-Techne RNAscope kit. We had several discussions with their technical team about best practices for analyzing the data. They made it clear to us that this single RNA molecule labeling approach reflects 1 RNA molecule per dot. They also emphasized that the intensity of dots *does not* reflect higher RNA expression, as the method involves massive non-linear amplification of signal to enable sm detection (the dots). For this reason, an intensity-based analysis is not appropriate with these data.

However, the number of dots *does* reflect the number of RNA molecules in one bin. In this case, the issue is only whether there are a countable number of discrete single-molecule dots and similar hybridization efficiencies can be assumed. The technical team believes that hybridization should be saturated under our conditions and parallel processing of test and controls would internally control for potential variability here. Moreover, in the all-representative high-resolution images in the figure, there are a discrete, countable set of green dots reflecting single molecules of the mRNAs in question.

We normalized to the total number of cells to reduce variability. We understand Reviewer 3’s point that it would have been better to assess the number of dots in every cell and compare it between controls and mutants. Because our zoomed in image analyses revealed almost exclusively intracellular dots, we think it is OK to analyze the data without laborious masking of individual cells.

Figure 9 and supplements: The authors' investigation of potential differences in Sox5, Bcl11b, Rorb splicing and 3' UTR usage is admirable. That said, the suggested investigation of isoform-specific translational regulation would not require a "tour-de force of multi-omic data integration"; the authors could simply repeat their qPCR analysis of their existing polysome profiles with their isoform-specific qPCR primers, and test if any isoforms show changes in the % of each isoform in the gradients, as is done in Figure 9 without isoform-specific primers. The authors' inclusion of a detailed explanation of polysome profiling analysis and quantification is also a positive; this is very helpful to the diverse audience for this paper, and for interpreting the translational changes.

We thank this reviewer for the appreciative comments regarding these analyses. The point with the “tour-de-force of multi-omic data integration” was simply in consideration of what it would take to do a comprehensive analysis of this subject using genome-wide approaches. Of course, isoform-specific primers can be used on the polysome fractions.

In fact, we have performed ORF and 3’UTR qPCR analysis on all our polysome profiles and showed the significant results in Figure 9.

We did not test all isoform-specific primers with our polysomes samples because we obtained a limited supply of material from the fractions and used up the samples for screening all ORF candidates and 3’UTR isoforms, so we didn’t have samples left for additional analyses (e.g., with newly designed splicing isoforms primers). Repeating all the polysome experiments yet again would have been very difficult and time consuming in our limited time under challenging circumstances. We decided to focus on making progress on the other important experiments requested. Moreover, had we found evidence for isoform-specific effects, this would certainly have required additional non-polysome experimentation to understand the significance for development. All in all, we think these are fascinating areas to explore systematically in the future (perhaps using multi-omic approaches).

However, the authors did not perform suggested and critical experiments focusing on quantifying Sox5, Bcl11b, and Rorb mRNA levels in layer IV/V, citing difficulties obtaining layer-specific Cre-driver lines, e.g. Rbp4-Cre for layer V or Rorb-Cre for layer IV, as well as difficulties obtaining sorter facility access due to COVID restrictions.Although unfortunate and understandable that this might require a longer revision period, this is a serious limitation. Initial review was very clear that these sets of experiments are crucial for full interpretation of the hypothesis of post-transcriptional regulation by Pum2 and TDP-43. It is both very conceivable and very possible that these proteins actually act primarily by regulating the mRNA abundance of the relevant mRNAs in the subtypes of interest, and the observed translational effects are merely secondary. The only cell population-specific evidence the authors present to argue against transcriptional regulation is the smFISH experiments in Figure 8, which are improved, but remain semi-quantitative at best, and thus insufficient to rule out transcriptional effects.If Cre-lines experiments remain overly challenging, the following experiment could be performed: layer V and layer IV neurons have different "birthdates", and can be labeled by BrdU incorporation at different developmental times. It is likely that in the Pum2 cKO, and hTDP-43 lines, there is a shift toward increased layer V specification at the times that typically yield layer IV neurons. The authors could perform a BrdU labeling experiment at E13.5-E14.5, and (1) look to see whether there is an increase in BrdU^+^Sox5+ or Bcl11b+ at P0, and decrease in BrdU^+^Rorb+ cells by immunofluorescence in Pum2 cKO and/or hTDP-43 lines, and (2) FACS-purify the BrdU^+^ cells, and perform qRT-PCR for Sox5, Bcl11b, and Rorb. This experiment should take only a few weeks to complete, requires no Cre driver lines to be imported, and requires no specialized procedures.In summary: the polysome profiling experiments demonstrate translational regulation of Sox5, Bcl11b, and Rorb in the developing cortex in Pum2 cKO or hTDP-43 overexpression mice compared to wild-type. However, this could be primarily due to transcriptional regulation, and only secondarily with translation effects. To be able to claim "bona-fide" post-transcriptional regulation, the authors would need to rigorously test the alternative hypothesis that Pum2 and TDP-43 regulate mRNA abundance of the genes of interest in the relevant cells. The evidence presented (smFISH) fails to rigorously test this hypothesis. The central conclusion of manuscript relies on this, and would fall apart if the alternative were the case, so the current set of experiments is incomplete without such rigorous tests of the alternative hypothesis.

We understand the limitation of smFISH and the benefit of the mentioned experiments. We do not exclude potential effects on transcription and/or mRNA stability (which would be bona fide post-transcriptional regulation) in neuronal subpopulations that might be missed in our bulk assays. We acknowledge that smFISH may not be sufficiently quantitative to detect these effects in situ. In R2 we try now to be even more explicit about these issues.

Very important: additional animal experiments need to be submitted for approval by an ethics committee in the context of a detailed animal experiment application where the 3R rules will be carefully considered. There are very few animal experiments that we can legally do in “a few weeks” even if the experiment itself can be done that quickly. The suggested BrdU experiment would have to be carefully justified and would need to be reviewed prior to approval. That process typically takes several months after the application itself is finished. We think it is important for the reviewers to take this into consideration.

Reviewer #3:The revised manuscript by Harb and colleagues has greatly improved and few key new experiments in support of the cell autonomous effect of the RBPs on the translational regulation of Bcl11b, Sox5 and Rorb. In particular, the IUE data as well as the colabeling analysis of Sox5 and Bcl11b, coupled with the in vitro neuronal culture, provide strong evidence for the control that Pum2 and TDP43 exert on these key players of neuronal diversification during corticogenesis.Nevertheless, some important points raised in the first review have not been fully addressed and leave the reader still puzzled by the interpretations of some analyses.In particular, the authors' choice of using the TDP43 mutant line is still problematic: while determining whether early developmental defects might contribute to the aetiology of neurodegenerative diseases is a compelling and very timely question – as numerous studies have recently been published along these lines – it is still unclear to me the link between translational regulation in area identity acquisition and the disease-associated mutations. This becomes even more puzzling when considering the chosen line does not develop ALS symptoms and therefore does not represent a true "disease model". Moreover, as previously requested in the reviews, the most suitable control line for the experiments involving the mutant line would have been the TDP43 wild type overexpression mouse model. If the goal was to address the effect of the disease-associated mutation any effect of the mutant line should have been properly assessed and 'normalised' to the wild type line, which – as stated by the authors in the revised manuscript – shows a milder alteration; if, on the other hand, the aim was to investigate the role of the control of TDP43 RPB on neuronal/area identity acquisition in the gain-of-function setting, the most appropriate line to be used should be the wild type line and not the mutant line, independently of the extent of the phenotype observed. The decision of the authors of not carrying along in ALL the analysis the is therefore arguable and leaves the reader confused on the specific goals of the study.

We understand this point and the unresolved issue about whether this early developmental defect is linked to disease-associated mutations. We have covered this issue in the Discussion. As shown in Figure 1—figure supplement 8, the hTDP-43^A315T^ line shows higher over-expression of cytoplasmic and nuclear TDP-43 which explains the milder but significant effect seen in the hTDP43 WT line which also suggests a dose-dependent effect of TDP-43 on cell identity. We aimed to re-do the retrograde labeling in this line as requested, but it was technically not possible to send these lines again to Geneva to perform new experiments. Repeating all our qRT-PCR, polysome profiling and FISH experiments again with this line was not possible in our revision time during the pandemic. Moreover, both our primary neuron transfection and IUE experiments with both alleles over-expressed in the same conditions showed similarly significant phenotypes.

Connectivity data: While the IUE experiments undoubtedly contributes to support a direct involvement of RBPs in the phenotypes observed by the authors and convincingly determine their control over canonical markers of neuronal subtypes, the lack of connectivity analysis in Pum2 ko as well as in the TDP43 wild type lines limit the finding to the cellular phenotypes. While convincing the data on the cKO and the mutant TDP43 lines, it might be risky to assume similar connectivity defects in the other contexts.

As mentioned in our R1 response, we were dependent on our collaborators (i.e., DJ) to perform these experiments. Ultimately, the complex logistics of mouse transfer during the CoViD pandemic, coupled with personnel changes in the Jabaudon lab, made it impossible to organize these experiments in the required time.

For R2, we have reviewed the sections about connectivity again and changed them to acknowledge the point that we have to be careful about assuming the connectivity phenotypes extrapolate to the lines/conditions not explicitly tested.

Rescue analysis: The rescue experiment in the Pum2 cKO or PumKO is not addressed at all, and according to the authors is beyond the scope of the study. We respectfully disagree with the authors about this point. Providing the rescue experiments, or at least attempting it with techniques that have been presented in the revised version of the manuscript like IUE, would have provided direct evidence for Pum2 to be sufficient for the expression of layer-specific markers in vivo, highlighting its physiological relevance in area-specific neuronal identity.

We agree that the experiments would have been useful and originally hoped to slot them in. Unfortunately, initial Pum2 constructs generated for IUE were problematic and we lacked the time and resources to generate and test new ones. We added a statement in the discussion indicating that it will be important to verify phenotypic rescue by Pum2.

One last point still remains problematic in this reviewer's opinion and it concerns the statistical power of the majority of the analysis in the manuscript (a point already raised in the previous review and that according to the point-to-point response the authors claimed to have addressed). Most of the data (including new experiments and analysis) shown in Figure 1-7, 9-10 as well as in the supplementary figures have been performed on "3 replicates per genotype", and in some rare cases even two dot points are shown in the bar plots. There is no reference in the text about the number of litters or sex of the animals and in some experiments – like IUE – this choice of analysis and data collection might dangerously fall below standards and impact the significance of the results.

We show in our legends and Methods section the use of 3 different litters per genotype. We do not use less than 3 animals, if only two dots might have appeared in any plot, it could be because 2 dots are very close or overlapping. We do not specify the sex of our animals because in our first analysis in mutant lines we haven’t noticed any difference in cell identity markers between males and females.

Ribosome profiling: the text related to this experiments has become significantly more clear and the logics of the different analysis can be easily followed in the description. However, it is unfortunate that no attempt to resolve the ribosome profiling at the population level (or at least at layer level, as already shown for RNA datasets in multiple publications) has been made by the authors in this revision. This would have provided stronger evidence to the mechanisms underlying the protein alteration and brought an additional level of novelty to the work that the bulk profiling analysis is currently lacking.

We agree that polysome profiling methods with enhanced cellular resolution would have been helpful for this manuscript. However, these methods are just emerging. Once robust workflows are established, we too are convinced they will greatly enable progress in the future – not only in the field of cortical development.

In addition, although greatly improved in the flow, the manuscript will still benefit from a more rigorous analysis and quantitative approach to better support the general claims. More specifically:– The quality of the NeuN images shown in Suppl Figure 2 are strongly divergent and do not look quite comparable. Has there any technical problem occurred that could motivate these differences?

We thank the reviewer for this comment. We have now shown better representative images that show consistent staining across genotypes.

– The nissl staining analysis as presented in Supplementary Figure 3 does not bring relevant information about the cytoarchitecture of the different models, as originally motivated in text, as no quantitative morphometric analyses have been performed, remaining merely qualitative. The overall figure will benefit for additional and improved imaging; indeed, multiple matching sections need to be considered to address overall brain architecture at comparable anatomical levels; higher quality images (the sections seem damaged at the pia level, and it is hard to discriminate the canonical tissue features of the cerebral cortex) coupled by punctual analysis of the higher magnification will help determine whether the evident impairment of the hippocampus observed in Pum cko – not claimed by the authors – is confirmed. Given the area phenotype observed, a more detailed analysis of the internal capsule and the somatic morphology of subcerebral PNs in different areas would have been extremely relevant and is currently missing. As presented, the current figure does not bring definitive support to the interpretations reported in the text and for the phenotype described is key to confirm the overall cytoarchitecture of the cerebral cortex: in several panels, indeed, the cortical thickness of the images shown is not comparable.

In our R2, we only claim cortical architecture is not impaired and we show better high magnification images for motor and prospective somatosensory cortex. We do not exclude any impairment in other brain structure especially the hippocampus in Pum2 mutants. Previous work has shown that Pum2 knockout affects hippocampal spine and synapse density. For R2, we have toned down our conclusion to say that cortical architecture is not impaired, which is related to our phenotype. Cortical thickness has been addressed with a high number of DAPI images with high number of animals in Figure 1—figure supplement 2b and it is shown not to be significantly impaired in either mutant.

In suppl. figure 7 there seem to be large differences in the overall cortical thickness where Pum2ko, ko and hz all show significant smaller cortices compare to the control.Is this a matching problem, an unfortunate selection of the images or this line presents abnormalities in the cortical thickness? it is hard to conlcude such results from the data. It needs to be toned down.

Our detailed analysis of cortical thickness in WT and *Pum2* cKO in Figure 1-Figure supplement 2b shows no differences in cortical thickness. We do not claim that cortical thickness is not impaired in hZ and Pum2KO since we haven’t analyzed this aspect there. However, in our view, Suppl figure 7 does not show huge differences in DAPI staining in cortical thickness (Layer VI to pial membrane) between the different genotypes.

– In the data reported about nuclear size, what cell types/layer is considered? no information are provided about where are those images shown in Figure suppl 2c are taken, neither if they represent any specific area of the cortex.

High magnification images at the level of somatosensory cortex were taken. They mostly cover layer IV and V since they mostly overlap with Ctip2 and *Rorb* stainings, and partly with layer VI and UL. For R2, we modified the text in the relevant figure legend to make this missing part clearer.

– Rorb staining in Figure 1 shows a great level of variability among the controls of each mouse lines, which is puzzling considering that the same antibody has been used and an automatic counting method has been used. Do the authors have any explanation for this discrepancy? it is important to assess how reliable is the difference observed in this marker expression. Moreover, in this case the TDP43 control becomes key to use as a reference for the mutant line.

The overall number of *Rorb* normalized to DAPI is almost 30% in both controls of both lines as shown in Figure 1-Figure supplement 5. The same antibody has been used in all experiments. However, an approximately 10% differences in lower cell number in single bins between both controls might arise from differences in binning system, subtle cortical thickness changes between pure wt and flox animals, different fixation, staining, imaging qualities. In general, the overall pattern is similar. This is not the case only for *Rorb*, but other markers. To reduce technical differences, we only use control littermates for each mutant line which are treated in the same way from the moment we take the pups until imaging.